# A Finite Sample Analysis of Distributional TD Learning with Linear Function Approximation

Yang Peng[*]    Kaicheng Jin[†]    Liangyu Zhang[‡]    Zhihua Zhang[§]

## Abstract

In this paper, we study the finite-sample statistical rates of distributional temporal difference (TD) learning with linear function approximation. The aim of distributional TD learning is to estimate the return distribution of a discounted Markov decision process for a given policy $\pi$. Previous works on statistical analysis of distributional TD learning mainly focus on the tabular case. In contrast, we first consider the linear function approximation setting and derive sharp finite-sample rates. Our theoretical results demonstrate that the sample complexity of linear distributional TD learning matches that of classic linear TD learning. This implies that, with linear function approximation, learning the full distribution of the return from streaming data is no more difficult than learning its expectation (value function). To derive tight sample complexity bounds, we conduct a fine-grained analysis of the linear-categorical Bellman equation and employ the exponential stability arguments for products of random matrices. Our results provide new insights into the statistical efficiency of distributional reinforcement learning algorithms.

## 1 Introduction

Distributional policy evaluation [Morimura et al., 2010, Bellemare et al., 2017, 2023], which aims to estimate the return distribution of a policy in an Markov decision process (MDP), is crucial for many uncertainty-aware or risk-sensitive tasks [Lim and Malik, 2022, Kastner et al., 2023]. Unlike the classic policy evaluation that only focuses on expected returns (value functions), distributional policy evaluation captures uncertainty and risk by considering the full distributional information. To solve a distributional policy evaluation problem, in the seminal work Bellemare et al. [2017] proposed distributional temporal difference (TD) learning, which can be viewed as an extension of classic TD learning [Sutton, 1988].

Although classic TD learning has been extensively studied [Bertsekas and Tsitsiklis, 1995, Tsitsiklis and Van Roy, 1996, Bhandari et al., 2018, Dalal et al., 2018, Patil et al., 2023, Li et al., 2024a,b, Chen et al., 2024, Samsonov et al., 2024a,b, Wu et al., 2024], the theoretical understanding of distributional TD learning, which is important for risk-sensitive tasks [Wang and Zhou, 2020, Wang et al., 2018, Moghimi and Ku, 2025, Ávila Pires et al., 2025, Qi et al., 2025], remains relatively underdeveloped. Recent works [Rowland et al., 2018, Böck and Heitzinger, 2022, Zhang et al., 2025, Rowland et al., 2024a,b, Peng et al., 2024] have analyzed distributional TD learning (or its model-based variants) in the tabular setting. Especially, Rowland et al. [2024b] and Peng et al. [2024] demonstrated that in the

---

[*]School of Mathematical Sciences, Peking University; email: `pengyang@pku.edu.cn`.

[†]School of Mathematical Sciences, Peking University; email: `kcjin@pku.edu.cn`.

[‡]School of Statistics and Data Science, Shanghai University of Finance and Economics; email: `zhangliangyu@sufe.edu.cn`.

[§]School of Mathematical Sciences, School of Computer Science, Peking University; Center for Intelligent Computing, Great Bay University; email: `zhzhang@math.pku.edu.cn`.

39th Conference on Neural Information Processing Systems (NeurIPS 2025).

tabular setting, learning the return distribution (in terms of the 1-Wasserstein distance[5]) is statistically as easy as learning its expectation. However, in practical scenarios, where the state space is extremely large or continuous, the function approximation [Dabney et al., 2018b,a, Rowland et al., 2019, Yang et al., 2019, Freirich et al., 2019, Yue et al., 2020, Nguyen-Tang et al., 2021, Zhou et al., 2021, Luo et al., 2022, Wenliang et al., 2024, Sun et al., 2024, Cho et al., 2024, Shen et al., 2025] becomes indispensable. This raises a new open question: *When function approximation is employed, does learning the return distribution remain as statistically efficient as learning its expectation?*

To answer this question, we consider the simplest form of function approximation, *i.e.*, linear function approximation, and investigate the finite-sample performance of linear distributional TD learning. In distributional TD learning, we need to represent the infinite-dimensional return distributions with some finite-dimensional parametrizations to make the algorithm tractable. Previous works [Bellemare et al., 2019, Lyle et al., 2019, Bellemare et al., 2023] have proposed various linear distributional TD learning algorithms under different parameterizations, namely categorical and quantile parametrizations. In this paper, we consider the categorical parametrization and propose an improved version of the linear-categorical TD learning algorithm (`Linear-CTD`). We then analyze the non-asymptotic convergence rate of `Linear-CTD`. Our analysis reveals that, with the Polyak-Ruppert tail averaging [Ruppert, 1988, Polyak and Juditsky, 1992] and a proper constant step size, the sample complexity of `Linear-CTD` matches that of classic linear TD learning (`Linear-TD`) [Li et al., 2024b, Samsonov et al., 2024b]. Thus, this confirms that learning the return distribution is statistically no more difficult than learning its expectation when the linear function approximation is employed.

**Notation.**    In the following parts of the paper, $(x)_+ := \max\{x, 0\}$ for any $x \in \mathbb{R}$. "$\lesssim$" (resp. "$\gtrsim$") means no larger (resp. smaller) than up to a multiplicative universal constant, and $a \simeq b$ means $a \lesssim b$ and $a \gtrsim b$ hold simultaneously. The asymptotic notation $f(\cdot) = \widetilde{\mathcal{O}}(g(\cdot))$ (resp. $\widetilde{\Omega}(g(\cdot))$) means that $f(\cdot)$ is order-wise no larger (resp. smaller) than $g(\cdot)$, ignoring logarithmic factors of polynomials of $(1-\gamma)^{-1}, \lambda_{\min}^{-1}, \alpha^{-1}, \varepsilon^{-1}, \delta^{-1}, K, \|\psi^\star\|_{\Sigma_\phi}, \|\theta^\star\|_{I_K \otimes \Sigma_\phi}$. We will explain the concrete meaning of the notation once we have encountered them for the first time.

We denote by $\delta_x$ the Dirac measure at $x \in \mathbb{R}$, $\mathbb{1}$ the indicator function, $\otimes$ the Kronecker product (see Appendix A), $\mathbf{1}_K \in \mathbb{R}^K$ the all-ones vector, $\mathbf{0}_K \in \mathbb{R}^K$ the all-zeros vector, $I_K \in \mathbb{R}^{K \times K}$ the identity matrix, $\|u\|$ the Euclidean norm of any vector $u$, $\|B\|$ the spectral norm of any matrix $B$, and $\|u\|_B := \sqrt{u^\top B u}$ when $B$ is positive semi-definite (PSD). $B_1 \preccurlyeq B_2$ stands for $B_2 - B_1$ is PSD for any symmetric matrices $B_1, B_2$. And $\prod_{k=1}^t B_k$ is defined as $B_t B_{t-1} \cdots B_1$ for any matrices $\{B_k\}_{k=1}^t$ with appropriate sizes. For any matrix $B = [b(1), \ldots, b(n)] \in \mathbb{R}^{m \times n}$, we define its vectorization as $\mathsf{vec}(B) = (b(1)^\top, \ldots, b(n)^\top)^\top \in \mathbb{R}^{mn}$. Given a set $A$, we denote by $\Delta(A)$ the set of all probability distributions over $A$. For simplicity, we abbreviate $\Delta([0, (1-\gamma)^{-1}])$ as $\mathscr{P}$.

**Contributions.**    Our contribution is two-fold: in algorithms and in theory. Algorithmically, we propose an improved version of the linear-categorical TD learning algorithm (`Linear-CTD`). Rather than using stochastic semi-gradient descent to update the parameter as in Bellemare et al. [2019], Lyle et al. [2019], Bellemare et al. [2023], we directly formulate the linear-categorical projected Bellman equation into a linear system and apply a linear stochastic approximation to solve it. The resulting `Linear-CTD` can be viewed as a preconditioned version [Chen, 2005, Li, 2017] of the vanilla linear categorical TD learning algorithm proposed in Bellemare et al. [2023, Section 9.6]. By introducing a preconditioner, our `Linear-CTD` achieves a finite-sample rate independent of the number of supports $K$ in the categorical parameterization, which the vanilla version cannot attain. We provide both theoretical and experimental evidence to demonstrate this advantage of our `Linear-CTD`.

Theoretically, we establish the first non-asymptotic guarantees for distributional TD learning with the linear function approximation. Specifically, we show that in the generative model setting, with the Polyak-Ruppert tail averaging and a constant step size, we need

$$T = \widetilde{\mathcal{O}}\left(\left(\varepsilon^{-2} + \lambda_{\min}^{-1}\right)(1-\gamma)^{-2}\lambda_{\min}^{-1}\left(K^{-1}(1-\gamma)^{-2}\|\theta^\star\|_{I_K \otimes \Sigma_\phi}^2 + 1\right)\right)$$

online interactions with the environment to ensure `Linear-CTD` yields a $\varepsilon$-accurate estimator with high probability, when the error is measured by the $\mu_\pi$-weighted 1-Wasserstein distance. We also

---

[5]Solving distributional policy evaluation $\varepsilon$-accurately in the 1-Wasserstein distance sense is harder than solving classic policy evaluation $\varepsilon$-accurately, as the absolute difference of value functions is always bounded by the 1-Wasserstein distance between return distributions.

extend the result to the Markovian setting. Our sample complexity bounds match those of the classic `Linear-TD` with a constant step size [Li et al., 2024b, Samsonov et al., 2024b], confirming the same statistical tractability of distributional and classic value-based policy evaluations. To establish these theoretical results, we analyze the linear-categorical Bellman equation in detail and apply the exponential stability argument proposed in Samsonov et al. [2024b]. Our analysis of the linear-categorical Bellman equation lays the foundation for subsequent algorithmic and theoretical advances in distributional reinforcement learning with function approximation.

**Organization.** The remainder of this paper is organized as follows. In Section 2, we recap `Linear-TD` and tabular categorical TD learning. In Section 3, we introduce the linear-categorical parametrization, and use the linear-categorical projected Bellman equation to derive `Linear-CTD`. In Section 4, we employ the exponential stability arguments to analyze the statistical efficiency of `Linear-CTD`. The proof is outlined in Section 5. In Section 6, we conclude our work. See Appendix B for more related work. In Appendix F, we compare various concepts and results between `Linear-TD` and `Linear-CTD`. In Appendix G, we empirically validate the convergence of `Linear-CTD` and compare it with prior algorithms through numerical experiments, confirming our theoretical findings. Details of the proof are given in the appendices.

## 2  Backgrounds

In this section, we recap the basics of policy evaluation and distributional policy evaluation tasks.

### 2.1  Policy Evaluation

A discounted MDP is defined by a 4-tuple $M = \langle \mathcal{S}, \mathcal{A}, \mathcal{P}, \gamma \rangle$. We assume that the state space $\mathcal{S}$ and the action space $\mathcal{A}$ are both Polish spaces, namely complete separable metric spaces. $\mathcal{P}(\cdot, \cdot \mid s, a)$ is the joint distribution of reward and next state condition on $(s, a) \in \mathcal{S} \times \mathcal{A}$. We assume that all rewards are bounded random variables in $[0, 1]$. And $\gamma \in (0, 1)$ is the discount factor.

Given a policy $\pi \colon \mathcal{S} \to \Delta(\mathcal{A})$ and an initial state $s_0 = s \in \mathcal{S}$, a random trajectory $\{(s_t, a_t, r_t)\}_{t=0}^{\infty}$ can be sampled: $a_t \mid s_t \sim \pi(\cdot \mid s_t)$, $(r_t, s_{t+1}) \mid (s_t, a_t) \sim \mathcal{P}(\cdot, \cdot \mid s_t, a_t)$, for any $t \in \mathbb{N}$. We assume the Markov chain $\{s_t\}_{t=0}^{\infty}$ has a unique stationary distribution $\mu_\pi \in \Delta(\mathcal{S})$. We define the return of the trajectory as $G^\pi(s) := \sum_{t=0}^{\infty} \gamma^t r_t$. The value function $V^\pi(s)$ is the expectation of $G^\pi(s)$, and $\boldsymbol{V}^\pi := (V^\pi(s))_{s \in \mathcal{S}} \in \mathbb{R}^{\mathcal{S}}$. It is known that $\boldsymbol{V}^\pi$ satisfies the Bellman equation:

$$V^\pi(s) = \mathbb{E}_{a \sim \pi(\cdot \mid s), (r, s') \sim \mathcal{P}(\cdot, \cdot \mid s, a)}[r + \gamma V^\pi(s')], \quad \forall s \in \mathcal{S}, \tag{1}$$

or in a compact form $\boldsymbol{V}^\pi = \boldsymbol{T}^\pi \boldsymbol{V}^\pi$, where $\boldsymbol{T}^\pi \colon \mathbb{R}^{\mathcal{S}} \to \mathbb{R}^{\mathcal{S}}$ is called the Bellman operator. In the task of policy evaluation, we aim to find the unique solution $\boldsymbol{V}^\pi$ of the equation for some given policy $\pi$.

**Tabular TD Learning.** The policy evaluation problem is reduced to solving the Bellman equation. However, in practical applications $\boldsymbol{T}^\pi$ is usually unknown and the agent only has access to the streaming data $\{(s_t, a_t, r_t)\}_{t=0}^{\infty}$. In this circumstance, we can solve the Bellman equation through linear stochastic approximation (LSA). Specifically, in the $t$-th time-step the updating scheme is

$$V_t(s_t) \leftarrow V_{t-1}(s_t) - \alpha \left( V_{t-1}(s_t) - r_t - \gamma V_{t-1}(s_{t+1}) \right), \quad V_t(s) \leftarrow V_{t-1}(s), \; \forall s \neq s_t. \tag{2}$$

We expect $\boldsymbol{V}_t$ to converge to $\boldsymbol{V}^\pi$ as $t$ tends to infinity. This algorithm is known as TD learning, however, it is computationally tractable only in the tabular setting.

**Linear Function Approximation and Linear TD Learning.** In this part, we introduce linear function approximation and briefly review the more practical `Linear-TD`. To be concrete, we assume there is a $d$-dimensional feature vector for each state $s \in \mathcal{S}$, which is given by the feature map $\boldsymbol{\phi} \colon \mathcal{S} \to \mathbb{R}^d$. We consider the linear function approximation of value functions:

$$\mathscr{V}_{\boldsymbol{\phi}} := \left\{ \boldsymbol{V}_{\boldsymbol{\psi}} = (V_{\boldsymbol{\psi}}(s))_{s \in \mathcal{S}} : V_{\boldsymbol{\psi}}(s) = \boldsymbol{\phi}(s)^\top \boldsymbol{\psi}, \boldsymbol{\psi} \in \mathbb{R}^d \right\} \subset \mathbb{R}^{\mathcal{S}}, \tag{3}$$

$\mu_\pi$-weighted norm $\|\boldsymbol{V}\|_{\mu_\pi} := (\mathbb{E}_{s \sim \mu_\pi}[V(s)^2])^{1/2}$, and linear projection operator $\boldsymbol{\Pi}_{\boldsymbol{\phi}}^\pi \colon \mathbb{R}^{\mathcal{S}} \to \mathscr{V}_{\boldsymbol{\phi}}$:

$$\boldsymbol{\Pi}_{\boldsymbol{\phi}}^\pi \boldsymbol{V} := \operatorname{argmin}_{\boldsymbol{V}_{\boldsymbol{\psi}} \in \mathscr{V}_{\boldsymbol{\phi}}} \|\boldsymbol{V} - \boldsymbol{V}_{\boldsymbol{\psi}}\|_{\mu_\pi}, \quad \forall \boldsymbol{V} \in \mathbb{R}^{\mathcal{S}}.$$

One can check that the linear projected Bellman operator $\mathbf{\Pi}_{\phi}^{\pi}\mathbf{T}^{\pi}$ is a $\gamma$-contraction in the Polish space $(\mathcal{V}_{\phi}, \|\cdot\|_{\mu_{\pi}})$. Hence, $\mathbf{\Pi}_{\phi}^{\pi}\mathbf{T}^{\pi}$ admits a unique fixed point $\mathbf{V}_{\psi^{\star}}$, which satisfies $\|\mathbf{V}^{\pi} - \mathbf{V}_{\psi^{\star}}\|_{\mu_{\pi}} \leq (1-\gamma^2)^{-1/2}\|\mathbf{V}^{\pi} - \mathbf{\Pi}_{\phi}^{\pi}\mathbf{V}^{\pi}\|_{\mu_{\pi}}$ [Bellemare et al., 2023, Theorem 9.8]. In Appendix C.1, we show that $\psi^{\star} \in \mathbb{R}^d$ is the unique solution to the linear system for $\psi \in \mathbb{R}^d$:

$$\left(\mathbf{\Sigma}_{\phi} - \gamma\mathbb{E}_{s,s'}\left[\phi(s)\phi(s')^{\top}\right]\right)\psi = \mathbb{E}_{s,r}\left[\phi(s)r\right], \quad \mathbf{\Sigma}_{\phi} := \mathbb{E}_{s\sim\mu_{\pi}}\left[\phi(s)\phi(s)^{\top}\right]. \qquad (4)$$

In the subscript of the expectation, we abbreviate $s \sim \mu_{\pi}(\cdot), a \sim \pi(\cdot|s), (r, s') \sim \mathcal{P}(\cdot, \cdot \mid s, a)$ as $s, a, r, s'$. For brevity, we will use such abbreviations in this paper when there is no ambiguity. We can use LSA to solve the linear projected Bellman equation (Eqn. 4). As a result, at the $t$-th time-step, the updating scheme of `Linear-TD` is

$$\texttt{Linear-TD:} \qquad \psi_t \leftarrow \psi_{t-1} - \alpha\phi(s_t)\left[(\phi(s_t) - \gamma\phi(s_{t+1}))^{\top}\psi_{t-1} - r_t\right]. \qquad (5)$$

## 2.2 Distributional Policy Evaluation

In certain applications, we are not only interested in finding the expectation of random return $G^{\pi}(s)$ but also want to find the whole distribution of $G^{\pi}(s)$. This task is called distributional policy evaluation. We use $\eta^{\pi}(s) \in \mathscr{P}$ to denote the distribution of $G^{\pi}(s)$ and let $\boldsymbol{\eta}^{\pi} := (\eta^{\pi}(s))_{s\in\mathcal{S}} \in \mathscr{P}^{\mathcal{S}}$. Then $\boldsymbol{\eta}^{\pi}$ satisfies the distributional Bellman equation:

$$\eta^{\pi}(s) = \mathbb{E}_{a\sim\pi(\cdot|s),(r,s')\sim\mathcal{P}(\cdot,\cdot|s,a)}[(b_{r,\gamma})_{\#}\,\eta^{\pi}(s')], \quad \forall s \in \mathcal{S}, \qquad (6)$$

where the RHS is the distribution of $r_0 + \gamma G^{\pi}(s_1)$ conditioned on $s_0 = s$. Here $b_{r,\gamma}(x) := r + \gamma x$ for any $x \in \mathbb{R}$, and $f_{\#}\nu \in \mathscr{P}$ is defined as $f_{\#}\nu(A) := \nu(\{x\colon f(x) \in A\})$ for any function $f\colon \mathbb{R} \to \mathbb{R}$, probability measure $\nu \in \mathscr{P}$ and Borel set $A \subset \mathbb{R}$. The distributional Bellman equation can also be written as $\boldsymbol{\eta}^{\pi} = \mathcal{T}^{\pi}\boldsymbol{\eta}^{\pi}$. The operator $\mathcal{T}^{\pi}\colon \mathscr{P}^{\mathcal{S}} \to \mathscr{P}^{\mathcal{S}}$ is called the distributional Bellman operator. In this task, our goal is to find $\boldsymbol{\eta}^{\pi}$ for some given policy $\pi$.

**Tabular Distributional TD Learning.** In analogy to tabular TD learning (Eqn. (2)), in the tabular setting, we can solve the distributional Bellman equation by LSA and derive the distributional TD learning rule given the streaming data $\{(s_t, a_t, r_t)\}_{t=0}^{\infty}$:

$$\eta_t(s_t) \leftarrow \eta_{t-1}(s_t) - \alpha[\eta_{t-1}(s_t) - (b_{r_t,\gamma})_{\#}\,\eta_{t-1}(s_{t+1})], \quad \eta_t(s) \leftarrow \eta_{t-1}(s), \ \forall s \neq s_t.$$

We comment the algorithm above is not computationally feasible as we need to manipulate infinite-dimensional objects (return distributions) at each iteration.

**Categorical Parametrization and Tabular Categorical TD Learning.** In order to deal with return distributions in a computationally tractable manner, we consider the categorical parametrization as in Bellemare et al. [2017], Rowland et al. [2018, 2024b], Peng et al. [2024]. To be compatible with linear function approximation introduced in the next section, which cannot guarantee non-negative outputs, we will work with $\mathscr{P}^{\mathsf{sign}}$, the signed measure space with total mass 1 as in Bellemare et al. [2019], Lyle et al. [2019], Bellemare et al. [2023] instead of standard probability space $\mathscr{P} \subset \mathscr{P}^{\mathsf{sign}}$:

$$\mathscr{P}^{\mathsf{sign}} := \left\{\nu\colon \nu(\mathbb{R}) = 1, \mathrm{supp}(\nu) \subseteq \left[0, (1-\gamma)^{-1}\right]\right\}.$$

For any $\nu \in \mathscr{P}^{\mathsf{sign}}$, we define its cumulative distribution function (CDF) as $F_{\nu}(x) := \nu([0, x])$. We can naturally define the $L^2$ and $L^1$ distances between CDFs as the Cramér distance $\ell_2$ and 1-Wasserstein distance $W_1$ in $\mathscr{P}^{\mathsf{sign}}$, respectively. The distributional Bellman operator (see Eqn. (6)) can also be extended to the product space $(\mathscr{P}^{\mathsf{sign}})^{\mathcal{S}}$ without modifying its definition.

The space of all categorical parametrized signed measures with total mass 1 is defined as

$$\mathscr{P}_K^{\mathsf{sign}} := \{\nu_{\boldsymbol{p}} = \textstyle\sum_{k=0}^K p_k\delta_{x_k}\colon \boldsymbol{p} = (p_0, \ldots, p_{K-1})^{\top} \in \mathbb{R}^K, p_K = 1 - \textstyle\sum_{k=0}^{K-1} p_k\}, \qquad (7)$$

which is an affine subspace of $\mathscr{P}^{\mathsf{sign}}$. Here $\{x_k = k\iota_K\}_{k=0}^K$ are $K+1$ equally-spaced points of the support, $\iota_K = [K(1-\gamma)]^{-1}$ is the gap between adjacent points, and $p_k$ is the 'probability' (may be negative) that $\nu$ assigns to $x_k$. We define the categorical projection operator $\mathbf{\Pi}_K\colon \mathscr{P}^{\mathsf{sign}} \to \mathscr{P}_K^{\mathsf{sign}}$ as

$$\mathbf{\Pi}_K\nu := \mathrm{argmin}_{\nu_{\boldsymbol{p}}\in\mathscr{P}_K^{\mathsf{sign}}} \ell_2\left(\nu, \nu_{\boldsymbol{p}}\right), \quad \forall \nu \in \mathscr{P}^{\mathsf{sign}}.$$

Following Bellemare et al. [2023, Proposition 5.14], one can show that $\mathbf{\Pi}_K \nu \in \mathscr{P}_K^{\text{sign}}$ is uniquely represented with a vector $\boldsymbol{p}_\nu = (p_k(\nu))_{k=0}^{K-1} \in \mathbb{R}^K$, where

$$p_k(\nu) = \int_{[0,(1-\gamma)^{-1}]} (1 - |(x - x_k)/\iota_K|)_+ \nu(dx). \tag{8}$$

We lift $\mathbf{\Pi}_K$ to the product space by defining $[\mathbf{\Pi}_K \boldsymbol{\eta}](s) := \mathbf{\Pi}_K \eta(s)$. One can check that the categorical Bellman operator $\mathbf{\Pi}_K \mathcal{T}^\pi$ is a $\sqrt{\gamma}$-contraction in the Polish space $((\mathscr{P}_K^{\text{sign}})^{\mathcal{S}}, \ell_{2,\mu_\pi})$, where $\ell_{2,\mu_\pi}(\boldsymbol{\eta}_1, \boldsymbol{\eta}_2) := (\mathbb{E}_{s \sim \mu_\pi}[\ell_2^2(\eta_1(s), \eta_2(s))])^{1/2}$ is the $\mu_\pi$-weighted Cramér distance between $\boldsymbol{\eta}_1, \boldsymbol{\eta}_2 \in (\mathscr{P}^{\text{sign}})^{\mathcal{S}}$. Similarly, $W_{1,\mu_\pi}(\boldsymbol{\eta}_1, \boldsymbol{\eta}_2) := (\mathbb{E}_{s \sim \mu_\pi}[W_1^2(\eta_1(s), \eta_2(s))])^{1/2}$. Hence, the categorical projected Bellman equation $\boldsymbol{\eta} = \mathbf{\Pi}_K \mathcal{T}^\pi \boldsymbol{\eta}$ admits a unique solution $\boldsymbol{\eta}^{\pi,K}$, which satisfies $W_{1,\mu_\pi}(\boldsymbol{\eta}^\pi, \boldsymbol{\eta}^{\pi,K}) \leq (1-\gamma)^{-1} \ell_{2,\mu_\pi}(\boldsymbol{\eta}^\pi, \mathbf{\Pi}_K \boldsymbol{\eta}^\pi)$ [Rowland et al., 2018, Proposition 3]. Applying LSA to solving the equation yields tabular categorical TD learning, and the iteration rule is given by

$$\eta_t(s_t) \leftarrow \eta_{t-1}(s_t) - \alpha[\eta_{t-1}(s_t) - \mathbf{\Pi}_K (b_{r_t,\gamma})_\# \eta_{t-1}(s_{t+1})], \quad \eta_t(s) \leftarrow \eta_{t-1}(s), \ \forall s \neq s_t. \tag{9}$$

## 3  Linear-Categorical TD Learning

In this section, we propose our `Linear-CTD` algorithm (Eqn. (13)) by combining the linear function approximation (Eqn. (3)) with the categorical parametrization (Eqn. (7)). We first introduce the space of linear-categorical parametrized signed measures with total mass 1:

$$\mathscr{P}_{\phi,K}^{\text{sign}} := \left\{ \boldsymbol{\eta}_{\boldsymbol{\theta}} = (\eta_{\boldsymbol{\theta}}(s))_{s \in \mathcal{S}} : \eta_{\boldsymbol{\theta}}(s) = \sum_{k=0}^K p_k(s; \boldsymbol{\theta}) \delta_{x_k}, \boldsymbol{\theta} = (\boldsymbol{\theta}(0)^\top, \ldots, \boldsymbol{\theta}(K-1)^\top)^\top \in \mathbb{R}^{dK} \right\},$$

which is an affine subspace of $(\mathscr{P}_K^{\text{sign}})^{\mathcal{S}}$. Here $p_k(s; \boldsymbol{\theta}) = F_k(s; \boldsymbol{\theta}) - F_{k-1}(s; \boldsymbol{\theta})$, and

$$F_k(s; \boldsymbol{\theta}) = \phi(s)^\top \boldsymbol{\theta}(k) + (k+1)/(K+1) \quad \text{for } k \in \{0, 1, \ldots, K-1\} \tag{10}$$

is CDF of $\eta_{\boldsymbol{\theta}}(s)$ at $x_k$ ($F_{-1}(s; \cdot) \equiv 0, F_K(s; \cdot) \equiv 1$), where $x_k \mapsto (k+1)/(K+1)$ is the CDF of the discrete uniform distribution $\nu$ on $\{x_0, \ldots, x_K\}$ used for normalization [6]. In many cases, especially when formulating and implementing algorithms, it is much more convenient and efficient to work with the matrix version of the parameter $\boldsymbol{\Theta} := (\boldsymbol{\theta}(0), \ldots, \boldsymbol{\theta}(K-1)) \in \mathbb{R}^{d \times K}$ rather than with $\boldsymbol{\theta} = \text{vec}(\boldsymbol{\Theta})$. We define the linear-categorical projection operator $\mathbf{\Pi}_{\phi,K}^\pi : (\mathscr{P}^{\text{sign}})^{\mathcal{S}} \to \mathscr{P}_{\phi,K}^{\text{sign}}$ as follows:

$$\mathbf{\Pi}_{\phi,K}^\pi \boldsymbol{\eta} := \operatorname{argmin}_{\boldsymbol{\eta}_{\boldsymbol{\theta}} \in \mathscr{P}_{\phi,K}^{\text{sign}}} \ell_{2,\mu_\pi}(\boldsymbol{\eta}, \boldsymbol{\eta}_{\boldsymbol{\theta}}), \quad \forall \boldsymbol{\eta} \in (\mathscr{P}^{\text{sign}})^{\mathcal{S}}.$$

$\mathbf{\Pi}_{\phi,K}^\pi$ is in fact an orthogonal projection (see Proposition D.2), and thus is non-expansive. The following proposition characterizes $\mathbf{\Pi}_{\phi,K}^\pi$, whose proof can be found in Appendix D.2.

**Proposition 3.1.** *For any $\boldsymbol{\eta} \in (\mathscr{P}^{\text{sign}})^{\mathcal{S}}$, $\mathbf{\Pi}_{\phi,K}^\pi \boldsymbol{\eta}$ is uniquely given by $\boldsymbol{\eta}_{\tilde{\boldsymbol{\theta}}}$, where $\tilde{\boldsymbol{\theta}} = \text{vec}(\tilde{\boldsymbol{\Theta}})$,*

$$\tilde{\boldsymbol{\Theta}} = \boldsymbol{\Sigma}_\phi^{-1} \mathbb{E}_{s \sim \mu_\pi}[\phi(s)(\boldsymbol{p}_{\boldsymbol{\eta}}(s) - (K+1)^{-1} \mathbf{1}_K)^\top \boldsymbol{C}^\top], \quad \boldsymbol{C} = [\mathbb{1}\{i \geq j\}]_{i,j \in [K]} \in \mathbb{R}^{K \times K}. \tag{11}$$

*Here $\boldsymbol{p}_{\boldsymbol{\eta}}(s) := \boldsymbol{p}_{\eta(s)} = (p_k(\eta(s)))_{k=0}^{K-1}$ is the vector that identifies $\mathbf{\Pi}_K \eta(s)$ defined in Eqn. (8).*

The following proposition characterizes the categorical projected Bellman operator $\mathbf{\Pi}_K \mathcal{T}^\pi$ when it acts on the parametrized $\boldsymbol{\eta}_{\boldsymbol{\theta}}$, whose proof can be found in Appendix D.3.

**Proposition 3.2.** *For any $\boldsymbol{\theta} \in \mathbb{R}^{dK}$ and $s \in \mathcal{S}$, we abbreviate $\boldsymbol{p}_{\mathcal{T}^\pi \boldsymbol{\eta}_{\boldsymbol{\theta}}}(s)$ as $\tilde{\boldsymbol{p}}_{\boldsymbol{\theta}}(s)$, then*

$$\tilde{\boldsymbol{p}}_{\boldsymbol{\theta}}(s) = (\tilde{p}_k(s; \boldsymbol{\theta}))_{k=0}^{K-1} = \mathbb{E}\left[\tilde{\boldsymbol{G}}(r_0) \boldsymbol{C}^{-1} \boldsymbol{\Theta}^\top \phi(s_1) \Big| s_0 = s\right] + \frac{1}{K+1} \sum_{j=0}^K \mathbb{E}\left[\boldsymbol{g}_j(r_0) \Big| s_0 = s\right],$$

*where for any $r \in [0,1]$ and $j, k \in \{0, 1, \ldots, K\}$,*

$$g_{j,k}(r) := (1 - |(r + \gamma x_j - x_k)/\iota_K|)_+, \quad \boldsymbol{g}_j(r) := (g_{j,k}(r))_{k=0}^{K-1} \in \mathbb{R}^K,$$

$$\boldsymbol{G}(r) := [\boldsymbol{g}_0(r), \ldots, \boldsymbol{g}_{K-1}(r)] \in \mathbb{R}^{K \times K}, \quad \tilde{\boldsymbol{G}}(r) := \boldsymbol{G}(r) - \mathbf{1}_K^\top \otimes \boldsymbol{g}_K(r) \in \mathbb{R}^{K \times K}.$$

---

[6] The normalizer $\nu$ is indispensable for achieving tight sample complexity bound, and it also guarantees our estimator has total mass 1, making the estimator more interpretable.

Since $\mathbf{\Pi}_{\phi,K}^{\pi}\mathcal{T}^{\pi}$ is a $\sqrt{\gamma}$-contraction in $(\mathscr{P}_{\phi,K}^{\mathsf{sign}}, \ell_{2,\mu_{\pi}})$ ($\mathcal{T}^{\pi}$ is $\sqrt{\gamma}$-contraction [Bellemare et al., 2023, Lemma 9.14]), in Theorem 4.1, using Proposition 3.1 and 3.2, we generalize the linear projected Bellman equation (Eqn. (4)) to the distributional setting. The proof can be found in Appendix D.4.

**Theorem 3.1.** *The linear-categorical projected Bellman equation $\boldsymbol{\eta_\theta} = \mathbf{\Pi}_{\phi,K}^{\pi}\mathcal{T}^{\pi}\boldsymbol{\eta_\theta}$ admits a unique solution $\boldsymbol{\eta_{\theta^\star}}$, where the matrix parameter $\mathbf{\Theta}^\star$ is the unique solution to the linear system for $\mathbf{\Theta} \in \mathbb{R}^{d \times K}$*

$$\mathbf{\Sigma}_\phi\mathbf{\Theta} - \mathbb{E}_{s,s',r}\Big[\boldsymbol{\phi}(s)\boldsymbol{\phi}(s')^\top\mathbf{\Theta}(C\tilde{G}(r)C^{-1})^\top\Big] = \tfrac{1}{K+1}\mathbb{E}_{s,r}\Big[\boldsymbol{\phi}(s)(\textstyle\sum_{j=0}^{K}\boldsymbol{g}_j(r)-\mathbf{1}_K)^\top C^\top\Big]. \quad (12)$$

In analogy to the approximation bounds of $\|V^\pi - V_{\psi^\star}\|_{\mu_\pi}$ and $W_{1,\mu_\pi}(\boldsymbol{\eta}^\pi, \boldsymbol{\eta}^{\pi,K})$, the following lemma answers how close $\boldsymbol{\eta_{\theta^\star}}$ is to $\boldsymbol{\eta}^\pi$, whose proof can be found in Appendix D.5.

**Proposition 3.3** (Approximation Error of $\boldsymbol{\eta_{\theta^\star}}$). *It holds that*

$$W_{1,\mu_\pi}^2\left(\boldsymbol{\eta}^\pi, \boldsymbol{\eta_{\theta^\star}}\right) \leq K^{-1}(1-\gamma)^{-3} + (1-\gamma)^{-2}\ell_{2,\mu_\pi}^2\left(\mathbf{\Pi}_K\boldsymbol{\eta}^\pi, \mathbf{\Pi}_{\phi,K}^{\pi}\boldsymbol{\eta}^\pi\right),$$

*where the first error term $K^{-1}(1-\gamma)^{-3}$ is due to the categorical parametrization, and the second error term $(1-\gamma)^{-2}\ell_{2,\mu_\pi}^2(\mathbf{\Pi}_K\boldsymbol{\eta}^\pi, \mathbf{\Pi}_{\phi,K}^{\pi}\boldsymbol{\eta}^\pi)$ is due to the additional linear function approximation.*

As before, we solve Eqn. (12) by LSA and get `Linear-CTD` given the streaming data $\{(s_t, a_t, r_t)\}_{t=0}^{\infty}$:

$$\texttt{Linear-CTD:} \qquad \mathbf{\Theta}_t \leftarrow \mathbf{\Theta}_{t-1} - \alpha\boldsymbol{\phi}(s_t)\Big[\boldsymbol{\phi}(s_t)^\top\mathbf{\Theta}_{t-1} - \boldsymbol{\phi}(s_{t+1})^\top\mathbf{\Theta}_{t-1}(C\tilde{G}(r_t)C^{-1})^\top$$
$$- (K+1)^{-1}\left(\textstyle\sum_{j=0}^{K}\boldsymbol{g}_j(r_t)-\mathbf{1}_K\right)^\top C^\top\Big], \qquad (13)$$

for any $t \geq 1$, where $\alpha$ is the constant step size. It is easy to verify that, in the special case of the tabular setting ($\boldsymbol{\phi}(s) = (\mathbb{1}\{s=\tilde{s}\})_{\tilde{s}\in\mathcal{S}}$), `Linear-CTD` is equivalent to tabular categorical TD learning (Eqn. (9)). In this paper, we consider the Polyak-Ruppert tail averaging $\bar{\boldsymbol{\theta}}_T := (T/2+1)^{-1}\sum_{t=T/2}^{T}\boldsymbol{\theta}_t$ (we use an even number $T$) as in the analysis of `Linear-TD` in Samsonov et al. [2024b]. Standard theory of LSA [Mou et al., 2020] says under some conditions, if we take an appropriate step size $\alpha$, $\bar{\boldsymbol{\theta}}_T$ will converge to the solution $\boldsymbol{\theta}^\star$ with rate $T^{-1/2}$ as $T \to \infty$.

In Figure 1, we empirically validate the convergence of `Linear-CTD` through numerical experiments. See Appendix G for details of the numerical experiments.

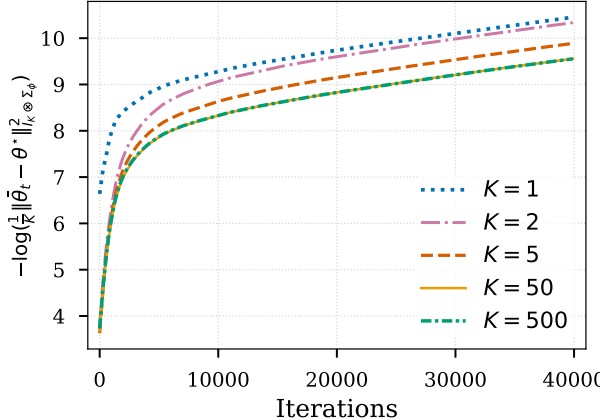

**Figure 1.** Convergence results under varying $K$ for our `Linear-CTD` algorithm with step size $\alpha = 0.01$. These curves exhibit similar trends, demonstrating our algorithm's robustness across different $K$ values.

**Remark 1** (Comparison with Existing Linear Distributional TD Learning Algorithms). *Our `Linear-CTD` can be regarded as a preconditioned version of vanilla stochastic semi-gradient descent (SSGD) with the probability mass function (PMF) representation [Bellemare et al., 2023, Section 9.6]. See Appendix D.6 for the PMF representation, and Appendix D.7 for a self-contained derivation of SSGD with PMF representations. The preconditioning technique is a commonly used methodology to accelerate solving optimization problems by reducing the condition number. We precondition the*

*vanilla algorithm by removing the matrix $\boldsymbol{C}^\top \boldsymbol{C}$ (see Eqn. (26)), whose condition number scales with $K^2$ (Lemma H.2). By introducing the preconditioner $(\boldsymbol{C}^\top \boldsymbol{C})^{-1}$, our Linear-CTD (Eqn. (13)) can achieve a convergence rate independent of $K$, which the vanilla form cannot achieve. See Remark 5 and Appendix G for theoretical and experimental evidence respectively.*

*We comment that Linear-CTD (Eqn. (13)) is equivalent to SSGD with CDF representation, which was also considered in Lyle et al. [2019]. The difference is that our Linear-CTD normalizes the distribution so that the total mass of return distributions always be 1, while the algorithm in Lyle et al. [2019] does not. See Appendix D.7 for a self-contained derivation of SSGD with CDF representations.*

*The previously mentioned works, as well as our Linear-CTD, are all limited by the use of signed measures (Eqn. (7)), which makes them less interpretable. Bellemare et al. [2023, Section 9.5] proposed a softmax-based linear-categorical algorithm, which is more closer to the practical C51 algorithm [Bellemare et al., 2017] and it uses standard probability measures. However, the nonlinearity due to softmax makes it difficult for analysis. We will leave the analysis for it as future work.*

**Remark 2** (Linear-CTD is mean-preserving). *A key property of Linear-CTD is mean preservation. That is, if we use identical initializations ($\mathbb{E}_{G \sim \boldsymbol{\eta}_{\boldsymbol{\theta}_0}}[G] = \boldsymbol{V}_{\boldsymbol{\psi}_0}$) and an identical data stream to update in both Linear-CTD and Linear-TD, it follows that $\mathbb{E}_{G \sim \boldsymbol{\eta}_{\boldsymbol{\theta}_t}}[G] = \boldsymbol{V}_{\boldsymbol{\psi}_t}$ for all $t$. However, the mean-preserving property does not hold for the SSGD with the PMF representation. In this sense, our Linear-CTD is indeed the generalization of Linear-TD. See Appendix D.8 for details.*

# 4 Non-Asymptotic Statistical Analysis

In our task, the quality of estimator $\boldsymbol{\eta}_{\bar{\boldsymbol{\theta}}_T}$ is measured by $\mu_\pi$-weighted 1-Wasserstein error $W_{1,\mu_\pi}(\boldsymbol{\eta}_{\bar{\boldsymbol{\theta}}_T}, \boldsymbol{\eta}^\pi)$. By triangle inequality, the error can be decomposed into the approximation error and the estimation error: $W_{1,\mu_\pi}(\boldsymbol{\eta}^\pi, \boldsymbol{\eta}_{\bar{\boldsymbol{\theta}}_T}) \leq W_{1,\mu_\pi}(\boldsymbol{\eta}^\pi, \boldsymbol{\eta}_{\boldsymbol{\theta}^\star}) + W_{1,\mu_\pi}(\boldsymbol{\eta}_{\boldsymbol{\theta}^\star}, \boldsymbol{\eta}_{\bar{\boldsymbol{\theta}}_T})$. Proposition 3.3 already provided an upper bound for the approximation error $W_{1,\mu_\pi}(\boldsymbol{\eta}^\pi, \boldsymbol{\eta}_{\boldsymbol{\theta}^\star})$, so it suffices to control the estimation error $W_{1,\mu_\pi}(\boldsymbol{\eta}_{\boldsymbol{\theta}^\star}, \boldsymbol{\eta}_{\bar{\boldsymbol{\theta}}_T})$, denoted $\mathcal{L}(\bar{\boldsymbol{\theta}}_T)$.

In the following theorem, we give non-asymptotic convergence rates of $\mathcal{L}(\bar{\boldsymbol{\theta}}_T)$. We start from the generative model setting, *i.e.*, in the $t$-th iteration, we collect samples $s_t \sim \mu_\pi(\cdot), a_t \sim \pi(\cdot|s_t), (r_t, s_t') \sim \mathcal{P}(\cdot, \cdot|s_t, a_t)$ from the generative model, and we replace $s_{t+1}$ with $s_t'$ in Eqn. (13). We give $L^p$ and high-probability convergence results in this setting. These results can be extended to the Markovian setting, *i.e.*, using the streaming data $\{(s_t, a_t, r_t)\}_{t=0}^\infty$.

## 4.1 $L^2$ Convergence

We first provide non-asymptotic convergence rates of $\mathbb{E}^{1/2}[(\mathcal{L}(\bar{\boldsymbol{\theta}}_T))^2]$, which do not grow with the number of supports $K$. The $L^p$ ($p > 2$) convergence results can be found in Theorem E.1.

**Theorem 4.1** ($L^2$ Convergence). *For any $K \geq (1-\gamma)^{-1}$ and $\alpha \in (0, (1-\sqrt{\gamma})/76)$, it holds that*

$$\mathbb{E}^{1/2}[(\mathcal{L}(\bar{\boldsymbol{\theta}}_T))^2] \lesssim \frac{\|\boldsymbol{\theta}^\star\|_{V_1} + 1}{\sqrt{T}(1-\gamma)\sqrt{\lambda_{\min}}} \left(1 + \sqrt{\frac{\alpha}{(1-\gamma)\lambda_{\min}}}\right) + \frac{\|\boldsymbol{\theta}^\star\|_{V_1} + 1}{T\sqrt{\alpha}(1-\gamma)^{\frac{3}{2}}\lambda_{\min}}$$

$$+ \frac{(1 - \frac{1}{2}\alpha(1-\sqrt{\gamma})\lambda_{\min})^{T/2}}{T\sqrt{\alpha}(1-\gamma)\sqrt{\lambda_{\min}}} \left(\frac{1}{\sqrt{\alpha}} + \frac{1}{\sqrt{(1-\gamma)\lambda_{\min}}}\right) \|\boldsymbol{\theta}_0 - \boldsymbol{\theta}^\star\|_{V_2},$$

*where $\|\boldsymbol{\theta}^\star\|_{V_1} := \frac{1}{\sqrt{K}(1-\gamma)} \|\boldsymbol{\theta}^\star\|_{\boldsymbol{I}_K \otimes \boldsymbol{\Sigma}_\phi}$ and $\|\boldsymbol{\theta}_0 - \boldsymbol{\theta}^\star\|_{V_2} := \frac{1}{\sqrt{K}(1-\gamma)} \|\boldsymbol{\theta}_0 - \boldsymbol{\theta}^\star\|$.*

In the upper bound, the first term of order $T^{-1/2}$ is dominant. The second term of order $T^{-1}$ has a worse dependence on $\alpha^{-1}$, leading to a sample size barrier (Eqn. (15)). The third term, corresponding to the initialization error, decays at a geometric rate and can be thus ignored. To prove Theorem 4.1, we conduct a fine-grained analysis of the linear-categorical Bellman equation and apply the exponential stability argument [Samsonov et al., 2024b]. We outline the proof in Section 5. In Remark 3, we compare our Theorem 4.1 with the $L^2$ convergence rate of classic Linear-TD and conclude that learning the distribution of the return is as easy as learning its expectation (value function) with linear function approximation. Rowland et al. [2024b], Peng et al. [2024] discovered this phenomenon in the tabular setting, and we extended it to the function approximation setting.

**Remark 3** (Comparison with Convergence Rate of `Linear-TD`). *The only difference between our Theorem [4.1](#) and the tight $L^2$ convergence rate of classic `Linear-TD` (see Appendix [C.2](#)) lies in replacing $\|\psi^\star\|_{\Sigma_\phi}$ (resp. $\|\psi_0 - \psi^\star\|$) in `Linear-TD` with $\|\theta^\star\|_{V_1}$ (resp. $\|\theta_0 - \theta^\star\|_{V_2}$). We claim that the two pairs should be of the same order, respectively. Note that $\|\theta^\star\|_{V_1}$ and $\|\theta_0 - \theta^\star\|_{V_2}$ are of order $\mathcal{O}((1-\gamma)^{-1})$ if $\eta_{\theta^\star}(s)$ and $\eta_{\theta_0}(s)$ are valid probability distributions for all $s \in \mathcal{S}$. This is because in this case, $F_k(s;\theta) = \phi(s)^\top \theta(k) + (k+1)/(K+1) \in [0,1]$ for $\theta \in \{\theta^\star, \theta_0\}$. While in `Linear-TD`, $\|\psi^\star\|_{\Sigma_\phi}$ and $\|\psi_0 - \psi^\star\|$ are also of order $\mathcal{O}((1-\gamma)^{-1})$ if $V_\psi(s) = \phi(s)^\top \psi \in [0, (1-\gamma)^{-1}]$ for all $s \in \mathcal{S}$ and $\psi \in \{\psi^\star, \psi_0\}$. It is thus reasonable to consider the two pairs with the same order, respectively. Similar arguments also hold in other convergence results presented in this paper. Therefore, in this sense, our results match those of `Linear-TD`.*

One can translate Theorem [4.1](#) into a sample complexity bound.

**Corollary 4.1.** *Under the same conditions as in Theorem [4.1](#), for any $\varepsilon > 0$, suppose*

$$T \gtrsim \frac{\|\theta^\star\|_{V_1}^2 + 1}{\varepsilon^2 (1-\gamma)^2 \lambda_{\min}} \left( 1 + \frac{\alpha}{(1-\gamma)\lambda_{\min}} \right) + \frac{\|\theta^\star\|_{V_1} + 1}{\varepsilon \sqrt{\alpha}(1-\gamma)^{\frac{3}{2}} \lambda_{\min}}$$

$$+ \frac{1}{\alpha(1-\gamma)\lambda_{\min}} \left( \log \frac{\|\theta_0 - \theta^\star\|_{V_2}}{\varepsilon} + \log \left( \frac{1}{T\sqrt{\alpha}(1-\gamma)\sqrt{\lambda_{\min}}} \left( \frac{1}{\sqrt{\alpha}} + \frac{1}{\sqrt{(1-\gamma)\lambda_{\min}}} \right) \right) \right).$$

*Then it holds that $\mathbb{E}^{1/2}[(\mathcal{L}(\bar{\theta}_T))^2] \leq \varepsilon$.*

**Instance-Independent Step Size.** If we take the largest possible instance-independent step size, *i.e.*, $\alpha \simeq (1-\gamma)$, and consider $\varepsilon \in (0,1)$, we obtain the sample complexity bound

$$T = \widetilde{\mathcal{O}} \left( \varepsilon^{-2}(1-\gamma)^{-2} \lambda_{\min}^{-2} \left( \|\theta^\star\|_{V_1}^2 + 1 \right) \right). \tag{14}$$

**Optimal Instance-Dependent Step Size.** If we take the optimal instance-dependent step size $\alpha \simeq (1-\gamma)\lambda_{\min}$ which involves the unknown $\lambda_{\min}$, we obtain a better sample complexity bound

$$T = \widetilde{\mathcal{O}} \left( \left( \varepsilon^{-2} + \lambda_{\min}^{-1} \right) (1-\gamma)^{-2} \lambda_{\min}^{-1} \left( \|\theta^\star\|_{V_1}^2 + 1 \right) \right). \tag{15}$$

There is a sample size barrier in the bound, that is, the dependence on $\lambda_{\min}$ is the optimal $\lambda_{\min}^{-1}$ only when $\varepsilon = \widetilde{\mathcal{O}}(\sqrt{\lambda_{\min}})$, or equivalently, we require a large enough (independent of $\varepsilon$) update steps $T$.

These results match the recent results for classic `Linear-TD` with a constant step size [Li et al., 2024b, Samsonov et al., 2024b]. It is possible to break the sample size barrier in Eqn. [(15)](#) as in `Linear-TD` by applying variance-reduction techniques [Li et al., 2023a]. We leave it for future work.

### 4.2 Convergence with High Probability and Markovian Samples

Applying the $L^p$ convergence result (Theorem [E.1](#)) with $p = 2\log(1/\delta)$ and Markov's inequality, we immediately obtain the high-probability convergence result.

**Theorem 4.2** (High-Probability Convergence). *For any $\varepsilon > 0$ and $\delta \in (0,1)$, suppose $K \geq (1-\gamma)^{-1}$, $\alpha \in (0, (1-\sqrt{\gamma})/[38\log(T/\delta^2)])$, and*

$$T = \widetilde{\mathcal{O}} \left( \frac{\|\theta^\star\|_{V_1}^2 + 1}{\varepsilon^2 (1-\gamma)^2 \lambda_{\min}} \left( 1 + \frac{\alpha \log \frac{1}{\delta}}{(1-\gamma)\lambda_{\min}} \right) \log \frac{1}{\delta} + \frac{\|\theta^\star\|_{V_1} + 1}{\varepsilon \sqrt{\alpha}(1-\gamma)^{\frac{3}{2}} \lambda_{\min}} \log \frac{1}{\delta} + \frac{\log \frac{\|\theta_0 - \theta^\star\|_{V_2}}{\varepsilon}}{\alpha(1-\gamma)\lambda_{\min}} \right).$$

*Then with probability at least $1 - \delta$, it holds that $\mathcal{L}(\bar{\theta}_T) \leq \varepsilon$. Here, the $\widetilde{O}(\cdot)$ does not hide polynomials of $\log(1/\delta)$ (but hides logarithm terms of $\log(1/\delta)$).*

Again, we will obtain concrete sample complexity bounds as in Eqn. [(14)](#) or Eqn. [(15)](#) if we use different step sizes. Compared with the theoretical results for classic `Linear-TD`, our results match Samsonov et al. [2024b, Theorem 4]. Samsonov et al. [2024b] also considered the constant step size, but obtained a worse dependence on $\log(1/\delta)$ than Wu et al. [2024, Theorem 4] which uses the polynomial-decaying step size $\alpha_t = \alpha_0 t^{-\beta}$ with $\beta \in (1/2, 1)$ instead.

**Remark 4** (Markovian Setting). *Using the same argument as in the proof of Samsonov et al. [2024b, Theorem 6], one can immediately derive a high-probability sample complexity bound in the Markovian setting. Compared with the bound in the generative model setting (Theorem 4.2), the bound in the Markovian setting will have an additional dependency on $t_{\mathrm{mix}} \log(T/\delta)$, where $t_{\mathrm{mix}}$ is the mixing time of the Markov chain $\{s_t\}_{t=0}^{\infty}$ in $\mathcal{S}$. We omit this result for brevity.*

## 5 Proof Outlines

In this section, we outline the proofs of our main results (Theorem 4.1). We first state the theoretical properties of the linear-categorical Bellman equation and the exponential stability of `Linear-CTD`. Finally, we highlight some key steps in proving these results.

### 5.1 Vectorization of Linear-CTD

In our analysis, it will be more convenient to work with the vectorization version of the updating scheme of `Linear-CTD` (Eqn. (13)):

$$\boldsymbol{\theta}_t \leftarrow \boldsymbol{\theta}_{t-1} - \alpha\left(\boldsymbol{A}_t\boldsymbol{\theta}_{t-1} - \boldsymbol{b}_t\right), \quad \boldsymbol{A}_t = \left[\boldsymbol{I}_K \otimes \left(\boldsymbol{\phi}(s_t)\boldsymbol{\phi}(s_t)^\top\right)\right] - \left[(\boldsymbol{C}\tilde{\boldsymbol{G}}(r_t)\boldsymbol{C}^{-1}) \otimes \left(\boldsymbol{\phi}(s_t)\boldsymbol{\phi}(s_t')^\top\right)\right],$$

$$\boldsymbol{b}_t = (K+1)^{-1}[\boldsymbol{C}(\textstyle\sum_{j=0}^K \boldsymbol{g}_j(r_t) - \mathbf{1}_K)] \otimes \boldsymbol{\phi}(s_t). \tag{16}$$

We denote by $\bar{\boldsymbol{A}}$ and $\bar{\boldsymbol{b}}$ the expectations of $\boldsymbol{A}_t$ and $\boldsymbol{b}_t$, respectively. Using exponential stability arguments, we can derive an upper bound for $\|\bar{\boldsymbol{A}}(\bar{\boldsymbol{\theta}}_T - \boldsymbol{\theta}^\star)\|$. The following lemma further translates it to an upper bound for $\mathcal{L}(\bar{\boldsymbol{\theta}}_T) = W_{1,\mu_\pi}(\boldsymbol{\eta}_{\boldsymbol{\theta}^\star}, \boldsymbol{\eta}_{\bar{\boldsymbol{\theta}}_T})$, whose proof is given in Appendix E.1.

**Lemma 5.1.** *For any $\boldsymbol{\theta} \in \mathbb{R}^{dK}$, it holds that $\mathcal{L}(\boldsymbol{\theta}) \leq 2K^{-1/2}(1-\gamma)^{-2}\lambda_{\min}^{-1/2}\|\bar{\boldsymbol{A}}(\boldsymbol{\theta} - \boldsymbol{\theta}^\star)\|$.*

### 5.2 Exponential Stability Analysis

First, we introduce some notations. Letting $\boldsymbol{e}_t := \boldsymbol{A}_t\boldsymbol{\theta}^\star - \boldsymbol{b}_t$, we denote by $C_A$ (resp. $C_e$) the almost sure upper bound for $\max\{\|\boldsymbol{A}_t\|, \|\boldsymbol{A}_t - \bar{\boldsymbol{A}}\|\}$ (resp. $\|\boldsymbol{e}_t\|$), and $\boldsymbol{\Sigma}_e := \mathbb{E}[\boldsymbol{e}_t\boldsymbol{e}_t^\top]$ the covariance matrix of $\boldsymbol{e}_t$. The following lemma provides useful upper bounds, whose proof is given in Appendix E.2.

**Lemma 5.2.** *For any $K \geq (1-\gamma)^{-1}$, it holds that*

$$C_A \leq 4, \quad C_e \leq 4(\|\boldsymbol{\theta}^\star\| + \sqrt{K}(1-\gamma)), \quad \mathrm{tr}(\boldsymbol{\Sigma}_e) \leq 18(\|\boldsymbol{\theta}^\star\|_{\boldsymbol{I}_K \otimes \boldsymbol{\Sigma}_\phi}^2 + K(1-\gamma)^2).$$

Let $\boldsymbol{\Gamma}_t^{(\alpha)} := \prod_{i=1}^t (\boldsymbol{I} - \alpha\boldsymbol{A}_i)$ for any $\alpha > 0$ and $t \in \mathbb{N}$. The exponential stability of `Linear-CTD` is summarized in the following lemma, whose proof can be found in Appendix E.3.

**Lemma 5.3.** *For any $p \geq 2$, let $a = (1-\sqrt{\gamma})\lambda_{\min}/2$ and $\alpha_{p,\infty} = (1-\sqrt{\gamma})/(38p)$ $(\alpha_{p,\infty}p \leq 1/2)$. Then for any $\alpha \in (0, \alpha_{p,\infty})$, $\boldsymbol{u} \in \mathbb{R}^{dK}$ and $t \in \mathbb{N}$, it holds that $\mathbb{E}^{1/p}[\|\boldsymbol{\Gamma}_t^{(\alpha)}\boldsymbol{u}\|^p] \leq (1-\alpha a)^t \|\boldsymbol{u}\|$.*

The following theorem states the $L^2$ convergence of $\|\bar{\boldsymbol{A}}(\bar{\boldsymbol{\theta}}_T - \boldsymbol{\theta}^\star)\|$ based on exponential stability arguments. For the general $L^p$ convergence, please refer to Samsonov et al. [2024b, Theorem 2].

**Theorem 5.1.** *[Samsonov et al., 2024b, Theorem 1] For any $\alpha \in (0, \alpha_{2,\infty})$, it holds that*

$$\mathbb{E}^{1/2}[\|\bar{\boldsymbol{A}}(\bar{\boldsymbol{\theta}}_T - \boldsymbol{\theta}^\star)\|^2] \lesssim \frac{\sqrt{\mathrm{tr}(\boldsymbol{\Sigma}_e)}}{\sqrt{T}}\left(1 + \frac{C_A\sqrt{\alpha}}{\sqrt{a}}\right) + \frac{\sqrt{\mathrm{tr}(\boldsymbol{\Sigma}_e)}}{T\sqrt{\alpha a}} + \frac{(1-\alpha a)^{T/2}}{T}\left(\frac{1}{\alpha} + \frac{C_A}{\sqrt{\alpha a}}\right)\|\boldsymbol{\theta}_0 - \boldsymbol{\theta}^\star\|.$$

Combining these lemmas with Theorem 5.1, we can immediately obtain Theorem 4.1.

**Remark 5** (Convergence of SSGD with PMF representation). *In Appendix E.5, we give counterparts of these lemmas and Theorem 4.1 for vanilla SSGD with PMF representation. The results imply that the step size in the algorithm should scale with $(1-\sqrt{\gamma})/K^2$ and the sample complexity grows with $K$. In Appendix G.2, we verify through numerical experiments that as $K$ increases, to ensure convergence, the step size of the vanilla algorithm indeed needs to decay at a rate of $K^{-2}$. In contrast, the step size of our `Linear-CTD` does not need to be adjusted when $K$ increases. Moreover, we find that `Linear-CTD` empirically consistently outperforms the vanilla algorithm under different $K$.*

## 5.3 Key Steps in the Proofs

Here we highlight some key steps in proving the above theoretical results.

**Bounding the Spectral Norm of Expectation of Kronecker Products.** In proving that the $\mathcal{L}(\boldsymbol{\theta})$ can be upper-bounded by $\|\bar{\boldsymbol{A}}(\boldsymbol{\theta} - \boldsymbol{\theta}^\star)\|$ (Lemma 5.1), as well as in verifying the exponential stability condition (Lemma 5.3), one of the most critical steps is to show

$$\left\| \mathbb{E}_{s,r,s'} \left[ \left( \boldsymbol{C}\tilde{\boldsymbol{G}}(r)\boldsymbol{C}^{-1} \right) \otimes \left( \boldsymbol{\Sigma}_{\boldsymbol{\phi}}^{-\frac{1}{2}} \boldsymbol{\phi}(s)\boldsymbol{\phi}(s')^\top \boldsymbol{\Sigma}_{\boldsymbol{\phi}}^{-\frac{1}{2}} \right) \right] \right\| \leq \sqrt{\gamma}, \quad \forall r \in [0,1]. \tag{17}$$

By Lemma H.3, we have $\|\boldsymbol{C}\tilde{\boldsymbol{G}}(r)\boldsymbol{C}^{-1}\| \leq \sqrt{\gamma}$ for any $r \in [0,1]$. In addition, one can check that $\|\mathbb{E}_{s,s'}[\boldsymbol{\Sigma}_{\boldsymbol{\phi}}^{-1/2}\boldsymbol{\phi}(s)\boldsymbol{\phi}(s')^\top\boldsymbol{\Sigma}_{\boldsymbol{\phi}}^{-1/2}]\| \leq 1$. One may speculate that the property $\|\boldsymbol{B}_1 \otimes \boldsymbol{B}_2\| = \|\boldsymbol{B}_1\|\|\boldsymbol{B}_2\|$ (Lemma A.3) is enough to get the desired conclusion. However, the two matrices in the Kronecker product are not independent, preventing us from using this simple property to derive the conclusion. On the other hand, since we only have the upper bound $\mathbb{E}_{s,s'}[\|\boldsymbol{\Sigma}_{\boldsymbol{\phi}}^{-1/2}\boldsymbol{\phi}(s)\boldsymbol{\phi}(s')^\top\boldsymbol{\Sigma}_{\boldsymbol{\phi}}^{-1/2}\|] \leq d$, simply moving the expectation in Eqn. (17) outside the norm will lead to a loose $d\sqrt{\gamma}$ bound. To resolve this problem, we leverage the fact that the second matrix is rank-1 and prove the following result. The proof can be found in the derivation following Eqn. (29).

**Lemma 5.4.** *For any random matrix $\boldsymbol{Y}$ and random vectors $\boldsymbol{x}$, $\boldsymbol{z}$, suppose $\|\boldsymbol{Y}\| \leq C_Y$ almost surely, $\mathbb{E}\left[\boldsymbol{x}\boldsymbol{x}^\top\right] \preccurlyeq C_x \boldsymbol{I}_{d_1}$ and $\mathbb{E}\left[\boldsymbol{z}\boldsymbol{z}^\top\right] \preccurlyeq C_z \boldsymbol{I}_{d_2}$ for some constants $C_Y, C_x, C_z > 0$. Then it holds that*

$$\left\| \mathbb{E}\left[ \boldsymbol{Y} \otimes \left( \boldsymbol{x}\boldsymbol{z}^\top \right) \right] \right\| \leq C_Y \sqrt{C_x C_z}.$$

**Remark 6** (Matrix Representation of Categorical Projected Bellman operator). *The matrix $\boldsymbol{C}\tilde{\boldsymbol{G}}(r)\boldsymbol{C}^{-1}$ also appears in Rowland et al. [2024b, Proposition B.2] as the matrix representation of the categorical projected Bellman operator $\boldsymbol{\Pi}_K \mathcal{T}^\pi$ of a specific one-state MDP. As a result, $\|\boldsymbol{C}\tilde{\boldsymbol{G}}(r)\boldsymbol{C}^{-1}\| \leq \sqrt{\gamma}$ because $\boldsymbol{\Pi}_K \mathcal{T}^\pi$ is a $\sqrt{\gamma}$-contraction in $(\mathscr{P}, \ell_2)$. Our Lemma H.3 provides a new analysis by directly analyzing the matrix.*

**Bounding the Norm of $\boldsymbol{b}_t$.** In proving Lemma 5.2, the most involved step is to upper-bound $\|\boldsymbol{b}_t\|$ (Eqn. (16)). To this end, we need to upper-bound the following term:

$$\frac{1}{K+1}\|\boldsymbol{C}(\textstyle\sum_{j=0}^K \boldsymbol{g}_j(r_t) - \boldsymbol{1}_K)\|, \quad \forall r \in [0,1]. \tag{18}$$

Term (18) is also related to the categorical projected Bellman operator $\boldsymbol{\Pi}_K \mathcal{T}^\pi$. Specifically, let $\nu = \frac{1}{K+1}\sum_{k=0}^K \delta_{x_k}$ be the discrete uniform distribution. One can show that Term (18) equals

$$\|\boldsymbol{C}(\boldsymbol{p}_{(b_{r,\gamma})_\#(\nu)} - \boldsymbol{p}_\nu)\| = \iota_K^{-1/2}\ell_2\left(\boldsymbol{\Pi}_K(b_{r,\gamma})_\#(\nu), \nu\right) \leq \iota_K^{-1/2}\ell_2\left((b_{r,\gamma})_\#(\nu), \nu\right) \leq 3\sqrt{K}(1-\gamma),$$

where we used the fact that $\boldsymbol{\Pi}_K$ is non-expansive and an upper bound for $\ell_2\left((b_{r,\gamma})_\#(\nu), \nu\right)$ when $K \geq (1-\gamma)^{-1}$ (Lemma H.4). The full proof can be found in the derivation following Eqn. (30).

## 6 Conclusions

In this paper, we have bridged a critical theoretical gap in distributional reinforcement learning by establishing the non-asymptotic sample complexity of distributional TD learning with linear function approximation. Specifically, we have proposed Linear-CTD, which is derived by solving the linear-categorical projected Bellman equation. By carefully analyzing the Bellman equation and using the exponential stability arguments, we have shown tight sample complexity bounds for the proposed algorithm. Our finite-sample rates match the state-of-the-art sample complexity bounds for conventional TD learning. These theoretical findings demonstrate that learning the full return distribution under linear function approximation can be statistically as easy as conventional TD learning for value function estimation. Our numerical experiments have provided empirical validation of our theoretical results. Finally, we have noted that it would be possible to improve the convergence rates by applying variance-reduction techniques or using polynomial-decaying step sizes, which we leave for future work.

## Acknowledgments and Disclosure of Funding

The authors would like to thank the reviewers for their constructive feedback, which helped us improve the quality of our work.

This work has been supported by the National Key Research and Development Project of China (No. 2022YFA1004002) and the National Natural Science Foundation of China (No. 12271011 and No. 12350001).

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

# A Kronecker Product

In this section, we will introduce some properties of Kronecker product used in our paper. See Zhang and Ding [2013] for a detailed treatment of Kronecker product.

For any matrices $\boldsymbol{A} \in \mathbb{R}^{m \times n}$ and $\boldsymbol{B} \in \mathbb{R}^{p \times q}$, the Kronecker product $\boldsymbol{A} \otimes \boldsymbol{B}$ is an matrix in $\mathbb{R}^{mp \times nq}$, defined as

$$\boldsymbol{A} \otimes \boldsymbol{B} = \begin{bmatrix} a_{11}\boldsymbol{B} & a_{12}\boldsymbol{B} & \cdots & a_{1n}\boldsymbol{B} \\ a_{21}\boldsymbol{B} & a_{22}\boldsymbol{B} & \cdots & a_{2n}\boldsymbol{B} \\ \vdots & \vdots & \ddots & \vdots \\ a_{m1}\boldsymbol{B} & a_{m2}\boldsymbol{B} & \cdots & a_{mn}\boldsymbol{B} \end{bmatrix}.$$

**Lemma A.1.** *The Kronecker product is bilinear and associative. Furthermore, for any matrices $\boldsymbol{B}_1, \boldsymbol{B}_2, \boldsymbol{B}_3, \boldsymbol{B}_4$ such that $\boldsymbol{B}_1\boldsymbol{B}_3$, $\boldsymbol{B}_2\boldsymbol{B}_4$ can be defined, it holds that $(\boldsymbol{B}_1 \otimes \boldsymbol{B}_2)(\boldsymbol{B}_3 \otimes \boldsymbol{B}_4) = (\boldsymbol{B}_1\boldsymbol{B}_3) \otimes (\boldsymbol{B}_2\boldsymbol{B}_4)$ (mixed-product property).*

*Proof.* See Basic properties and Theorem 3 in Zhang and Ding [2013]. $\square$

**Lemma A.2.** *For any matrices $\boldsymbol{B}_1, \boldsymbol{B}_2, \boldsymbol{B}_3$ such that $\boldsymbol{B}_1\boldsymbol{B}_2\boldsymbol{B}_3$ can be defined, it holds that $\operatorname{vec}(\boldsymbol{B}_1\boldsymbol{B}_2\boldsymbol{B}_3) = (\boldsymbol{B}_3^\top \otimes \boldsymbol{B}_1) \operatorname{vec}(\boldsymbol{B}_2)$.*

*Proof.* See Lemma 4.3.1 in Horn and Johnson [1994]. $\square$

**Lemma A.3.** *For any matrices $\boldsymbol{B}_1$ and $\boldsymbol{B}_2$, it holds that $\|\boldsymbol{B}_1 \otimes \boldsymbol{B}_2\| = \|\boldsymbol{B}_1\| \|\boldsymbol{B}_2\|$, $(\boldsymbol{B}_1 \otimes \boldsymbol{B}_2)^\top = \boldsymbol{B}_1^\top \otimes \boldsymbol{B}_2^\top$. Furthermore, if $\boldsymbol{B}_1$ and $\boldsymbol{B}_2$ are invertible/orthogonal/diagonal/symmetric/normal, $\boldsymbol{B}_1 \otimes \boldsymbol{B}_2$ is also invertible/orthogonal/diagonal/symmetric/normal and $(\boldsymbol{B}_1 \otimes \boldsymbol{B}_2)^{-1} = \boldsymbol{B}_1^{-1} \otimes \boldsymbol{B}_2^{-1}$.*

*Proof.* See Basic properties, Theorem 5 and Theorem 7 in Zhang and Ding [2013]. $\square$

**Lemma A.4.** *For any $K, d \in \mathbb{N}$ and PSD matrices $\boldsymbol{B}_1, \boldsymbol{B}_3 \in \mathbb{R}^{K \times K}, \boldsymbol{B}_2, \boldsymbol{B}_4 \in \mathbb{R}^{d \times d}$ with $\boldsymbol{B}_1 \preccurlyeq \boldsymbol{B}_3$ and $\boldsymbol{B}_2 \preccurlyeq \boldsymbol{B}_4$, it holds that $\boldsymbol{B}_1 \otimes \boldsymbol{B}_2, \boldsymbol{B}_3 \otimes \boldsymbol{B}_4$ are also PSD matrices, furthermore, $\boldsymbol{B}_1 \otimes \boldsymbol{B}_2 \preccurlyeq \boldsymbol{B}_3 \otimes \boldsymbol{B}_4$.*

*Proof.* Consider the spectral decomposition $\boldsymbol{B}_i = \boldsymbol{Q}_i \boldsymbol{D}_i \boldsymbol{Q}_i^\top$, for any $i \in [4]$, by Lemma A.1 and Lemma A.3, we have

$$(\boldsymbol{B}_1 \otimes \boldsymbol{B}_2) = (\boldsymbol{Q}_1 \otimes \boldsymbol{Q}_2)(\boldsymbol{D}_1 \otimes \boldsymbol{D}_2)(\boldsymbol{Q}_1 \otimes \boldsymbol{Q}_2)^\top$$

and

$$(\boldsymbol{B}_3 \otimes \boldsymbol{B}_4) = (\boldsymbol{Q}_3 \otimes \boldsymbol{Q}_4)(\boldsymbol{D}_3 \otimes \boldsymbol{D}_4)(\boldsymbol{Q}_3 \otimes \boldsymbol{Q}_4)^\top$$

are also spectral decomposition of $(\boldsymbol{B}_1 \otimes \boldsymbol{B}_2)$ and $(\boldsymbol{B}_3 \otimes \boldsymbol{B}_4)$ respectively. It is easy to see that they are PSD. Furthermore,

$$\begin{aligned} (\boldsymbol{B}_3 \otimes \boldsymbol{B}_4) - (\boldsymbol{B}_1 \otimes \boldsymbol{B}_2) &= [(\boldsymbol{B}_3 \otimes \boldsymbol{B}_4) - (\boldsymbol{B}_3 \otimes \boldsymbol{B}_2)] + [(\boldsymbol{B}_3 \otimes \boldsymbol{B}_2) - (\boldsymbol{B}_1 \otimes \boldsymbol{B}_2)] \\ &= [\boldsymbol{B}_3 \otimes (\boldsymbol{B}_4 - \boldsymbol{B}_2)] + [(\boldsymbol{B}_3 - \boldsymbol{B}_1) \otimes \boldsymbol{B}_2] \\ &\succcurlyeq \boldsymbol{0}. \end{aligned}$$

$\square$

**Lemma A.5.** *For any $K, d, d_1, d_2 \in \mathbb{N}$, vectors $\boldsymbol{u}, \boldsymbol{v} \in \mathbb{R}^d$ and matrices $\boldsymbol{B}_1 \in \mathbb{R}^{K \times d_1}, \boldsymbol{B}_2 \in \mathbb{R}^{d_2 \times K}, \boldsymbol{B}_3 \in \mathbb{R}^{K \times K}$, it holds that*

$$(\boldsymbol{I}_K \otimes \boldsymbol{u}) \boldsymbol{B}_1 = \boldsymbol{B}_1 \otimes \boldsymbol{u},$$

$$\boldsymbol{B}_2 (\boldsymbol{I}_K \otimes \boldsymbol{v})^\top = \boldsymbol{B}_2 \otimes \boldsymbol{v}^\top,$$

$$(\boldsymbol{I}_K \otimes \boldsymbol{u}) \boldsymbol{B}_3 (\boldsymbol{I}_K \otimes \boldsymbol{v})^\top = \boldsymbol{B}_3 \otimes (\boldsymbol{u}\boldsymbol{v}^\top).$$

*Furthermore, for any matrix $\boldsymbol{B}_4 \in \mathbb{R}^{d_1 \times d_2}$, we have*

$$(\boldsymbol{B}_1 \otimes \boldsymbol{u}) \boldsymbol{B}_4 = (\boldsymbol{B}_1\boldsymbol{B}_4) \otimes \boldsymbol{u}.$$

*Proof.* Let $\boldsymbol{u} = (u_i)_{i=1}^d$ $\boldsymbol{B}_1 = (b_{ij})_{i,j=1}^K$, then

$$
(\boldsymbol{I}_K \otimes \boldsymbol{u}) \, \boldsymbol{B}_1 =
\begin{bmatrix}
\boldsymbol{u} & \boldsymbol{0}_d & \cdots & \boldsymbol{0}_d & \boldsymbol{0}_d \\
\boldsymbol{0}_d & \boldsymbol{u} & \cdots & \boldsymbol{0}_d & \boldsymbol{0}_d \\
\vdots & \vdots & \ddots & \vdots & \vdots \\
\boldsymbol{0}_d & \boldsymbol{0}_d & \cdots & \boldsymbol{u} & \boldsymbol{0}_d \\
\boldsymbol{0}_d & \boldsymbol{0}_d & \cdots & \boldsymbol{0}_d & \boldsymbol{u}
\end{bmatrix}
\begin{bmatrix}
b_{11} & \cdots & b_{1K} \\
\vdots & \ddots & \vdots \\
b_{K1} & \cdots & b_{KK}
\end{bmatrix}
$$

$$
=
\begin{bmatrix}
b_{11}u_1 & \cdots & b_{1K}u_1 \\
\vdots & \ddots & \vdots \\
b_{11}u_d & \cdots & b_{1K}u_d \\
\vdots & \ddots & \vdots \\
b_{K1}u_1 & \cdots & b_{KK}u_1 \\
\vdots & \ddots & \vdots \\
b_{K1}u_d & \cdots & b_{KK}u_d
\end{bmatrix}
$$

$$
=
\begin{bmatrix}
b_{11}\boldsymbol{u} & \cdots & b_{1K}\boldsymbol{u} \\
\vdots & \ddots & \vdots \\
b_{K1}\boldsymbol{u} & \cdots & b_{KK}\boldsymbol{u}
\end{bmatrix}
$$

$$
= \boldsymbol{B}_1 \otimes u.
$$

Hence

$$
\boldsymbol{B}_2 \left(\boldsymbol{I}_K \otimes \boldsymbol{v}\right)^\top = \left[ (\boldsymbol{I}_K \otimes \boldsymbol{v}) \otimes \boldsymbol{B}_2^\top \right]^\top = \left[ \boldsymbol{B}_2^\top \otimes \boldsymbol{v} \right]^\top = \boldsymbol{B}_2 \otimes \boldsymbol{v}^\top.
$$

And in the same way,

$$
\left(\boldsymbol{I}_K \otimes \boldsymbol{u}\right) \boldsymbol{B}_3 \left(\boldsymbol{I}_K \otimes \boldsymbol{v}\right)^\top = \left(\boldsymbol{B}_3 \otimes \boldsymbol{u}\right) \otimes \boldsymbol{v}^\top = \boldsymbol{B}_3 \otimes \left(\boldsymbol{u} \otimes \boldsymbol{v}^\top\right) = \boldsymbol{B}_3 \otimes \left(\boldsymbol{u}\boldsymbol{v}^\top\right).
$$

Furthermore,

$$
\left(\boldsymbol{B}_1 \otimes \boldsymbol{u}\right) \boldsymbol{B}_4 = \left[ (\boldsymbol{I}_K \otimes \boldsymbol{u}) \boldsymbol{B}_1 \right] \boldsymbol{B}_4 = (\boldsymbol{I}_K \otimes \boldsymbol{u}) (\boldsymbol{B}_1 \boldsymbol{B}_4) = (\boldsymbol{B}_1 \boldsymbol{B}_4) \otimes \boldsymbol{u}.
$$

$\square$

# B   Related Work

**Comparisons of Theoretical Results with the Previous Work**   In Table 1, we summarize our main theoretical results and comparing them with prior work. In the table, when the task is policy evaluation, the sample complexity is defined in terms of the $\mu_\pi$-weighted $L^2$ norm as the measure of error; when the task is distributional policy evaluation, the sample complexity is defined in terms of the $\mu_\pi$-weighted $W_1$ metric as the measure of error. The table gives a clear comparison of theoretical results with prior work.

**Distributional Reinforcement Learning.**   Distributional TD learning was first proposed in Bellemare et al. [2017]. Following the distributional perspective in Bellemare et al. [2017], Qu et al. [2019] proposed a distributional version of the gradient TD learning algorithm, Tang et al. [2022] proposed a distributional version of multi-step TD learning, Tang et al. [2024] proposed a distributional version of off-policy Q($\lambda$) and TD($\lambda$) algorithms, and Wu et al. [2023] proposed a distributional version of fitted Q evaluation to solve the distributional offline policy evaluation problem. Qi et al. [2025] also considered the distributional off-policy evaluation problem. Wiltzer et al. [2024b] proposed an approach for evaluating the return distributions for all policies simultaneously when the reward is deterministic or in the finite-horizon setting. Wiltzer et al. [2024a] studied distributional policy evaluation in the multivariate reward setting and proposed corresponding TD learning algorithms. Beyond the tabular setting, Bellemare et al. [2019], Lyle et al. [2019], Bellemare et al. [2023] proposed various distributional TD learning algorithms with linear function approximation under different parametrizations.

A series of recent studies have focused on the theoretical properties of distributional TD learning. Rowland et al. [2018], Böck and Heitzinger [2022], Zhang et al. [2025], Rowland et al. [2024a,b],

**Table 1.** Sample complexity of algorithms for solving policy evaluation and distributional policy evaluation.

| Paper | Sample Complexity | Method |
|---|---|---|
| Samsonov et al. [2024b] | $\widetilde{\mathcal{O}}\left(\frac{\|\boldsymbol{\psi}^\star\|^2_{\boldsymbol{\Sigma}_\phi}+1}{(1-\gamma)^2\lambda_{\min}}\left(\frac{1}{\varepsilon^2}+\frac{1}{\lambda_{\min}}\right)\right)$ | `Linear-TD` |
| Li et al. [2023a] | $\widetilde{\mathcal{O}}\left(\frac{\|\boldsymbol{\psi}^\star\|^2_{\boldsymbol{\Sigma}_\phi}+1}{\varepsilon^2(1-\gamma)^2\lambda_{\min}}\right)$ | `Linear-TD` with Variance Reduction |
| Bellemare et al. [2023, Section 9.7] | Contraction Analysis | Vanilla `Linear-CTD` |
| This Work (Theorem E.2) | $\widetilde{\mathcal{O}}\left(\frac{K^4(\|\boldsymbol{\theta}^\star\|^2_{V_1}+1)}{(1-\gamma)^2\lambda_{\min}}\left(\frac{1}{\varepsilon^2}+\frac{K^2}{\lambda_{\min}}\right)\right)$ | Vanilla `Linear-CTD` |
| This Work (Theorem 4.1) | $\widetilde{\mathcal{O}}\left(\frac{\|\boldsymbol{\theta}^\star\|^2_{V_1}+1}{(1-\gamma)^2\lambda_{\min}}\left(\frac{1}{\varepsilon^2}+\frac{1}{\lambda_{\min}}\right)\right)$ | `Linear-CTD` |

Here, Vanilla `Linear-CTD` refers to the stochastic semi-gradient descent algorithm with the probability mass function representational (see Eqn. (27) in Appendix D.7). The contraction analysis in [Bellemare et al., 2023, Section 9.7] means that they provided a contraction analysis for the dynamic programming version of Vanilla `Linear-CTD` algorithm.

Peng et al. [2024], Kastner et al. [2025] analyzed the asymptotic and non-asymptotic convergence of distributional TD learning (or its model-based variants) in the tabular setting. Among these works, Rowland et al. [2024b], Peng et al. [2024] established that in the tabular setting, learning the full return distribution is statistically as easy as learning its expectation in the model-based and model-free settings, respectively. And Bellemare et al. [2019] provided an asymptotic convergence result for categorical TD learning with linear function approximation.

Beyond the problem of distributional policy evaluation, Rowland et al. [2023], Wang et al. [2023, 2024] showed that theoretically the classic value-based reinforcement learning could benefit from distributional reinforcement learning. Bäuerle and Ott [2011], Chow and Ghavamzadeh [2014], Marthe et al. [2023], Noorani et al. [2023], Moghimi and Ku [2025], Ávila Pires et al. [2025] considered optimizing statistical functionals of the return, and proposed algorithms to solve this harder problem.

**Stochastic Approximation.** Our `Linear-CTD` falls into the category of LSA. The classic TD learning, as one of the most classic LSA problems, has been extensively studied [Bertsekas and Tsitsiklis, 1995, Tsitsiklis and Van Roy, 1996, Bhandari et al., 2018, Dalal et al., 2018, Patil et al., 2023, Duan and Wainwright, 2023, Li et al., 2024a,b, Samsonov et al., 2024a, Wu et al., 2024]. Among these works, Li et al. [2024b], Samsonov et al. [2024b] provided the tightest bounds for `Linear-TD` with constant step sizes, which is also considered in our paper. While Wu et al. [2024] established the tightest bounds for `Linear-TD` with polynomial-decaying step sizes.

For general stochastic approximation problems, extensive works [Lakshminarayanan and Szepesvari, 2018, Srikant and Ying, 2019, Mou et al., 2020, 2022, Huo et al., 2023, Li et al., 2023b, Durmus et al., 2024, Samsonov et al., 2024b, Chen et al., 2024] have provided solid theoretical understandings.

## C   Omitted Results and Proofs in Section 2

### C.1   Linear Projected Bellman Equation

It is worth noting that, $\boldsymbol{\Pi}^\pi_\phi\colon\left(\mathbb{R}^{\mathcal{S}},\|\cdot\|_{\mu_\pi}\right)\to\left(\mathscr{V}_\phi,\|\cdot\|_{\mu_\pi}\right)$ is an orthogonal projection.

We aim to derive Eqn. (4). It is easy to check that, for any $V\in\mathbb{R}^{\mathcal{S}}$, $\boldsymbol{\Pi}^\pi_\phi V$ is uniquely give by $V_{\tilde{\psi}}$ where

$$\tilde{\psi}=\boldsymbol{\Sigma}^{-1}_\phi\mathbb{E}_{s\sim\mu_\pi}\left[\boldsymbol{\phi}(s)V(s)\right].$$

Hence, by the definition of Bellman operator (Eqn. (1)), $\boldsymbol{\psi}^\star$ is the unique solution to the following system of linear equations for $\boldsymbol{\psi}\in\mathbb{R}^d$

$$\begin{aligned}
\boldsymbol{\psi}&=\boldsymbol{\Sigma}^{-1}_\phi\mathbb{E}_{s\sim\mu_\pi}\left[\boldsymbol{\phi}(s)\left[\boldsymbol{T}^\pi\boldsymbol{V}_\psi\right](s)\right]\\
&=\boldsymbol{\Sigma}^{-1}_\phi\mathbb{E}_{s\sim\mu_\pi}\left[\boldsymbol{\phi}(s)\left(\mathbb{E}\left[r_0\mid s_0=s\right]+\gamma\mathbb{E}\left[\boldsymbol{\phi}(s_1)^\top\mid s_0=s\right]\boldsymbol{\psi}\right)\right]\\
&=\boldsymbol{\Sigma}^{-1}_\phi\mathbb{E}_{s,s'}\left[\boldsymbol{\phi}(s)\boldsymbol{\phi}(s')^\top\right]\boldsymbol{\psi}+\boldsymbol{\Sigma}^{-1}_\phi\mathbb{E}_{s,r}\left[\boldsymbol{\phi}(s)r\right],
\end{aligned}$$

or equivalently,

$$\left(\boldsymbol{\Sigma}_{\boldsymbol{\phi}} - \gamma \mathbb{E}_{s,s'}\left[\boldsymbol{\phi}(s)\boldsymbol{\phi}(s')^{\top}\right]\right)\boldsymbol{\psi} = \mathbb{E}_{s,r}\left[\boldsymbol{\phi}(s)r\right].$$

## C.2 Convergence Results for Linear TD Learning

It is worthy noting that, `Linear-TD` is equivalent to the stochastic semi-gradient descent (SSGD) update.

In `Linear-TD`, our goal is to find a good estimator $\hat{\boldsymbol{\psi}}$ such that $\left\|\boldsymbol{V}_{\hat{\boldsymbol{\psi}}} - \boldsymbol{V}_{\boldsymbol{\psi}^{\star}}\right\|_{\mu_{\pi}} = \left\|\hat{\boldsymbol{\psi}} - \boldsymbol{\psi}^{\star}\right\|_{\boldsymbol{\Sigma}_{\boldsymbol{\phi}}} \leq \varepsilon$. Samsonov et al. [2024b] considered the Polyak-Ruppert tail averaging $\bar{\boldsymbol{\psi}}_T := (T/2+1)^{-1}\sum_{t=T/2}^{T}\boldsymbol{\psi}_t$, and showed that in the generative model setting with constant step size $\alpha \simeq (1-\gamma)\lambda_{\min}$,

$$T = \widetilde{\mathcal{O}}\left(\frac{\|\boldsymbol{\psi}^{\star}\|_{\boldsymbol{\Sigma}_{\boldsymbol{\phi}}}^2 + 1}{(1-\gamma)^2\lambda_{\min}}\left(\frac{1}{\varepsilon^2} + \frac{1}{\lambda_{\min}}\right)\right)$$

is sufficient to guarantee that $\left\|\boldsymbol{V}_{\bar{\boldsymbol{\psi}}_T} - \boldsymbol{V}_{\boldsymbol{\psi}^{\star}}\right\|_{\mu_{\pi}} \leq \varepsilon$. They also provided sample complexity bounds when taking the instance-independent (*i.e.*, not dependent on unknown quantity) step size, and in the Markovian setting.

## C.3 Categorical Parametrization is an Isometry

**Proposition C.1.** *The affine space* $\left(\mathscr{P}_K^{\mathsf{sign}}, \ell_2\right)$ *is isometric with* $\left(\mathbb{R}^K, \sqrt{\iota_K}\,\|\cdot\|_{\boldsymbol{C}^{\top}\boldsymbol{C}}\right)$, *in the sense that, for any* $\nu_{\boldsymbol{p}_1}, \nu_{\boldsymbol{p}_2} \in \mathscr{P}_K^{\mathsf{sign}}$, *it holds that* $\ell_2^2(\nu_{\boldsymbol{p}_1}, \nu_{\boldsymbol{p}_2}) = \iota_K \|\boldsymbol{p}_1 - \boldsymbol{p}_2\|_{\boldsymbol{C}^{\top}\boldsymbol{C}}^2$, *where* $\boldsymbol{C}$ *is defined in Eqn.* (11).

*Proof.*

$$\begin{aligned}
\ell_2^2(\nu_{\boldsymbol{p}_1}, \nu_{\boldsymbol{p}_2}) &= \int_0^{(1-\gamma)^{-1}}\left(F_{\nu_{\boldsymbol{p}_1}}(x) - F_{\nu_{\boldsymbol{p}_2}}(x)\right)^2 dx \\
&= \iota_K \sum_{k=0}^{K-1}\left(F_{\nu_{\boldsymbol{p}_1}}(x_k) - F_{\nu_{\boldsymbol{p}_2}}(x_k)\right)^2 \\
&= \iota_K \|\boldsymbol{C}(\boldsymbol{p}_1 - \boldsymbol{p}_2)\|^2 \\
&= \iota_K \|\boldsymbol{p}_1 - \boldsymbol{p}_2\|_{\boldsymbol{C}^{\top}\boldsymbol{C}}^2.
\end{aligned}$$

$\square$

## C.4 Categorical Projection Operator is Orthogonal Projection

**Proposition C.2.** *[Bellemare et al., 2023, Lemma 9.17] For any* $\nu \in \mathscr{P}^{\mathsf{sign}}$ *and* $\nu_{\boldsymbol{p}} \in \mathscr{P}_K^{\mathsf{sign}}$, *it holds that*

$$\ell_2^2(\nu, \nu_{\boldsymbol{p}}) = \ell_2^2(\nu, \boldsymbol{\Pi}_K\nu) + \ell_2^2(\boldsymbol{\Pi}_K\nu, \nu_{\boldsymbol{p}}).$$

## C.5 Categorical Projected Bellman Operator

The following lemma characterizing $\boldsymbol{\Pi}_K\mathcal{T}^{\pi}$ is useful for both practice and theoretical analysis.

**Proposition C.3.** *For any* $\boldsymbol{\eta} \in \left(\mathscr{P}^{\mathsf{sign}}\right)^{\mathcal{S}}$ *and* $s \in \mathcal{S}$, *it holds that*

$$\begin{aligned}
\boldsymbol{p}_{\mathcal{T}^{\pi}\boldsymbol{\eta}}(s) &= \mathbb{E}\left[\boldsymbol{g}_K(r_0) + \left(\boldsymbol{G}(r_0) - \boldsymbol{1}_K^{\top} \otimes \boldsymbol{g}_K(r_0)\right)\boldsymbol{p}_{\boldsymbol{\eta}}(s_1)\,\Big|\,s_0 = s\right] \\
&= \mathbb{E}\left[\tilde{\boldsymbol{G}}(r_0)\left(\boldsymbol{p}_{\boldsymbol{\eta}}(s_1) - \frac{1}{K+1}\boldsymbol{1}_K\right)\,\Big|\,s_0 = s\right] + \frac{1}{K+1}\sum_{j=0}^{K}\mathbb{E}\left[\boldsymbol{g}_j(r_0)\,\Big|\,s_0 = s\right].
\end{aligned}$$

*And in the same way, for any $r \in [0,1]$ and $s' \in \mathcal{S}$, it holds that*

$$\boldsymbol{p}_{(b_{r,\gamma})_\# \eta(s')} = \tilde{\boldsymbol{G}}(r) \left( \boldsymbol{p}_{\boldsymbol{\eta}}(s') - \frac{1}{K+1}\mathbf{1}_K \right) + \frac{1}{K+1}\sum_{j=0}^{K} \boldsymbol{g}_j(r),$$

*where $\tilde{\boldsymbol{G}}$ and $\boldsymbol{g}$ is defined in Theorem 3.1.*

This proposition is a special case of Proposition 3.2, whose proof can be found in Appendix D.3.

# D Omitted Results and Proofs in Section 3

## D.1 Linear-Categorical Parametrization is an Isometry

**Proposition D.1.** *The affine space $\left( \mathscr{P}_{\phi,K}^{\mathsf{sign}}, \ell_{2,\mu_\pi} \right)$ is isometric with $\left( \mathbb{R}^{dK}, \sqrt{\iota_K} \left\| \cdot \right\|_{\boldsymbol{I}_K \otimes \boldsymbol{\Sigma}_\phi} \right)$, in the sense that, for any $\boldsymbol{\eta}_{\boldsymbol{\theta}_1}, \boldsymbol{\eta}_{\boldsymbol{\theta}_2} \in \mathscr{P}_{\phi,K}^{\mathsf{sign}}$, it holds that $\ell_{2,\mu_\pi}^2 \left( \boldsymbol{\eta}_{\boldsymbol{\theta}_1}, \boldsymbol{\eta}_{\boldsymbol{\theta}_2} \right) = \iota_K \left\| \boldsymbol{\theta}_1 - \boldsymbol{\theta}_2 \right\|_{\boldsymbol{I}_K \otimes \boldsymbol{\Sigma}_\phi}^2.$*

*Proof.* For any $\boldsymbol{\eta}_{\boldsymbol{\theta}} \in \mathscr{P}_{\phi,K}^{\mathsf{sign}}$, we denote $\boldsymbol{F}_{\boldsymbol{\theta}}(s) = (F_k(s; \boldsymbol{\theta}))_{k=0}^{K-1} \in \mathbb{R}^K$, then it holds that

$$\begin{aligned}
\ell_{2,\mu_\pi}^2 \left( \boldsymbol{\eta}_{\boldsymbol{\theta}_1}, \boldsymbol{\eta}_{\boldsymbol{\theta}_2} \right) &= \iota_K \mathbb{E}_{s \sim \mu_\pi} \left[ \left\| \boldsymbol{F}_{\boldsymbol{\theta}_1}(s) - \boldsymbol{F}_{\boldsymbol{\theta}_2}(s) \right\|^2 \right] \\
&= \iota_K \operatorname{tr} \left( \boldsymbol{\Sigma}_\phi^{\frac{1}{2}} \left( \boldsymbol{\Theta}_1 - \boldsymbol{\Theta}_2 \right) \left( \boldsymbol{\Theta}_1 - \boldsymbol{\Theta}_2 \right)^\top \boldsymbol{\Sigma}_\phi^{\frac{1}{2}} \right) \\
&= \iota_K \left\| \boldsymbol{\theta}_1 - \boldsymbol{\theta}_2 \right\|_{\boldsymbol{I}_K \otimes \boldsymbol{\Sigma}_\phi}^2.
\end{aligned}$$

$\square$

## D.2 Linear-Categorical Projection Operator

Proposition 3.1 is an immediate corollary of the following lemma. For any $\nu \in \mathscr{P}_K^{\mathsf{sign}}$, we define $\boldsymbol{F}_\nu = (F_k(\nu))_{k=0}^{K-1} = (\nu([0, x_k]))_{k=0}^{K-1} \in \mathbb{R}^K$, and for any $\boldsymbol{\eta} \in \left( \mathscr{P}^{\mathsf{sign}} \right)^{\mathcal{S}}$, we define $\boldsymbol{p}_{\boldsymbol{\eta}}(s) = \boldsymbol{p}_{\Pi_K \eta(s)}$ and $\boldsymbol{F}_{\boldsymbol{\eta}}(s) = \boldsymbol{F}_{\Pi_K \eta(s)}$.

**Lemma D.1.** *For any $\boldsymbol{\eta} \in \left( \mathscr{P}^{\mathsf{sign}} \right)^{\mathcal{S}}$, $\boldsymbol{\theta} \in \mathbb{R}^{dK}$ and $s \in \mathcal{S}$, it holds that*

$$\begin{aligned}
\nabla_{\boldsymbol{\Theta}} \ell_2^2 \left( \eta_{\boldsymbol{\theta}}(s), \eta(s) \right) &= 2\iota_K \phi(s) \left( \boldsymbol{F}_{\boldsymbol{\theta}}(s) - \boldsymbol{F}_{\boldsymbol{\eta}}(s) \right)^\top \\
&= 2\iota_K \phi(s) \left[ \phi(s)^\top \boldsymbol{\Theta} + \left( \frac{1}{K+1}\mathbf{1}_K - \boldsymbol{p}_{\boldsymbol{\eta}}(s) \right)^\top \boldsymbol{C}^\top \right].
\end{aligned} \tag{19}$$

*Furthermore, it holds that*

$$\begin{aligned}
\nabla_{\boldsymbol{\Theta}} \ell_{2,\mu_\pi}^2 \left( \boldsymbol{\eta}_{\boldsymbol{\theta}}, \boldsymbol{\eta} \right) &= \mathbb{E}_{s \sim \mu_\pi} \left[ \nabla_{\boldsymbol{\Theta}} \ell_2^2 \left( \eta_{\boldsymbol{\theta}}(s), \eta(s) \right) \right] \\
&= 2\iota_K \left[ \boldsymbol{\Sigma}_\phi \boldsymbol{\Theta} + \mathbb{E}_{s \sim \mu_\pi} \left[ \phi(s) \left( \frac{1}{K+1}\mathbf{1}_K - \boldsymbol{p}_{\boldsymbol{\eta}}(s) \right)^\top \right] \boldsymbol{C}^\top \right].
\end{aligned}$$

*Proof.* According to Proposition C.2, one has

$$\ell_2^2 \left( \eta_{\boldsymbol{\theta}}(s), \eta(s) \right) = \ell_2^2 \left( \eta_{\boldsymbol{\theta}}(s), \Pi_K \eta(s) \right) + \ell_2^2 \left( \Pi_K \eta(s), \eta(s) \right).$$

Hence,

$$\begin{aligned}
\nabla_{\boldsymbol{\theta}} \ell_2^2 \left( \eta_{\boldsymbol{\theta}}(s), \eta(s) \right) &= \nabla_{\boldsymbol{\theta}} \ell_2^2 \left( \eta_{\boldsymbol{\theta}}(s), \Pi_K \eta(s) \right) \\
&= \iota_K \nabla_{\boldsymbol{\theta}} \left\| \boldsymbol{F}_{\boldsymbol{\theta}}(s) - \boldsymbol{F}_{\boldsymbol{\eta}}(s) \right\|^2 \\
&= 2\iota_K \left( \boldsymbol{I}_K \otimes \phi(s) \right) \left( \boldsymbol{F}_{\boldsymbol{\theta}}(s) - \boldsymbol{F}_{\boldsymbol{\eta}}(s) \right) \\
&= 2\iota_K \left( \boldsymbol{I}_K \otimes \phi(s) \right) \left( \left( \boldsymbol{I}_K \otimes \phi(s) \right)^\top \boldsymbol{\theta} + \boldsymbol{C} \left( \frac{1}{K+1}\mathbf{1}_K - \boldsymbol{p}_{\boldsymbol{\eta}}(s) \right) \right)
\end{aligned}$$

$$= 2\iota_K \left\{ \left[ \boldsymbol{I}_K \otimes \left( \boldsymbol{\phi}(s)\boldsymbol{\phi}(s)^\top \right) \right] \boldsymbol{\theta} + \left[ \left( \boldsymbol{C} \left( \frac{1}{K+1} \boldsymbol{1}_K - \boldsymbol{p}_\eta(s) \right) \right) \otimes \boldsymbol{\phi}(s) \right] \right\}.$$

We also have the following matrix representation:

$$\nabla_\Theta \ell_2^2 \left( \eta_\theta(s), \eta(s) \right) = 2\iota_K \boldsymbol{\phi}(s) \left( \boldsymbol{F}_\theta(s) - \boldsymbol{F}_\eta(s) \right)^\top$$

$$= 2\iota_K \boldsymbol{\phi}(s) \left[ \boldsymbol{\phi}(s)^\top \boldsymbol{\Theta} + \left( \frac{1}{K+1} \boldsymbol{1}_K - \boldsymbol{p}_\eta(s) \right)^\top \boldsymbol{C}^\top \right].$$

$\square$

**Proposition D.2.** *For any* $\eta \in \left( \mathscr{P}^{\mathsf{sign}} \right)^{\mathcal{S}}$ *and* $\eta_\theta \in \mathscr{P}^{\mathsf{sign}}_{\phi,K}$, *it holds that*

$$\ell_{2,\mu_\pi}^2 \left( \boldsymbol{\eta}, \boldsymbol{\eta_\theta} \right) = \ell_{2,\mu_\pi}^2 \left( \boldsymbol{\eta}, \boldsymbol{\Pi}_K \boldsymbol{\eta} \right) + \ell_{2,\mu_\pi}^2 \left( \boldsymbol{\Pi}_K \boldsymbol{\eta}, \boldsymbol{\Pi}^\pi_{\phi,K} \boldsymbol{\eta} \right) + \ell_{2,\mu_\pi}^2 \left( \boldsymbol{\Pi}^\pi_{\phi,K} \boldsymbol{\eta}, \boldsymbol{\eta_\theta} \right).$$

The proof is straightforward and almost the same as that of Proposition C.2 if we utilize the affine structure.

### D.3 Proof of Proposition 3.2

*Proof.* Recall the definition of the distributional Bellman operator Eqn. (6) and categorical projection operator Eqn. (8), we have

$$\tilde{p}_k(s; \boldsymbol{\theta}) = p_k \left( \left[ \boldsymbol{\mathcal{T}}^\pi \boldsymbol{\eta_\theta} \right](s) \right)$$

$$= \mathbb{E}_{X \sim [\boldsymbol{\mathcal{T}}^\pi \boldsymbol{\eta_\theta}](s)} \left[ \left( 1 - \left| \frac{X - x_k}{\iota_K} \right| \right)_+ \right]$$

$$= \mathbb{E} \left[ \mathbb{E}_{G \sim \eta_\theta(s_1)} \left[ \left( 1 - \left| \frac{r_0 + \gamma G - x_k}{\iota_K} \right| \right)_+ \right] \middle| s_0 = s \right]$$

$$= \mathbb{E} \left[ \sum_{j=0}^K p_j(s_1; \boldsymbol{\theta}) \left( 1 - \left| \frac{r_0 + \gamma x_j - x_k}{\iota_K} \right| \right)_+ \middle| s_0 = s \right] \qquad (20)$$

$$= \mathbb{E} \left[ \sum_{j=0}^K p_j(s_1; \boldsymbol{\theta}) g_{j,k}(r_0) \middle| s_0 = s \right]$$

$$= \mathbb{E} \left[ g_{K,k}(r_0) + \sum_{j=0}^{K-1} p_j(s_1; \boldsymbol{\theta}) \left( g_{j,k}(r_0) - g_{K,k}(r_0) \right) \middle| s_0 = s \right].$$

Hence, let $\boldsymbol{W} = \boldsymbol{\Theta} \boldsymbol{C}^{-\top}$ and $\boldsymbol{w} = \mathrm{vec}(\boldsymbol{W}) = \left( \boldsymbol{C}^{-1} \otimes \boldsymbol{I}_d \right) \boldsymbol{\theta}$ (see Appendix D.6 for their meaning), then

$$\tilde{\boldsymbol{p}}_\theta(s) = \left( \tilde{p}_k(s; \boldsymbol{\theta}) \right)_{k=0}^{K-1}$$

$$= \mathbb{E} \left[ \begin{bmatrix} g_{K,1}(r_0) \\ \vdots \\ g_{K,K-1}(r_0) \end{bmatrix} + \sum_{j=0}^{K-1} p_j(s_1; \boldsymbol{\theta}) \begin{bmatrix} g_{j,1}(r_0) - g_{K,1}(r_0) \\ \vdots \\ g_{j,K-1}(r_0) - g_{K,K-1}(r_0) \end{bmatrix} \middle| s_0 = s \right]$$

$$= \mathbb{E} \left[ \boldsymbol{g}_K(r_0) + \sum_{j=0}^{K-1} p_j(s_1; \boldsymbol{\theta}) \left( \boldsymbol{g}_j(r_0) - \boldsymbol{g}_K(r_0) \right) \middle| s_0 = s \right]$$

$$= \mathbb{E} \left[ \boldsymbol{g}_K(r_0) + \left( \boldsymbol{G}(r_0) - \boldsymbol{1}_K^\top \otimes \boldsymbol{g}_K(r_0) \right) \boldsymbol{p}_\theta(s_1) \middle| s_0 = s \right]$$

$$= \mathbb{E} \left[ \boldsymbol{g}_K(r_0) + \left( \boldsymbol{G}(r_0) - \boldsymbol{1}_K^\top \otimes \boldsymbol{g}_K(r_0) \right) \left[ \left( \boldsymbol{I}_K \otimes \boldsymbol{\phi}(s_1) \right)^\top \boldsymbol{w} + \frac{1}{K+1} \boldsymbol{1}_K \right] \middle| s_0 = s \right]$$

$$= \mathbb{E} \left[ \left( \boldsymbol{G}(r_0) - \boldsymbol{1}_K^\top \otimes \boldsymbol{g}_K(r_0) \right) \left( \boldsymbol{I}_K \otimes \boldsymbol{\phi}(s_1) \right)^\top \middle| s_0 = s \right] \boldsymbol{w}$$

$$+ \mathbb{E} \left[ \boldsymbol{g}_K(r_0) + \frac{1}{K+1} \left( \boldsymbol{G}(r_0) - \mathbf{1}_K^\top \otimes \boldsymbol{g}_K(r_0) \right) \mathbf{1}_K \Big| s_0 = s \right]$$

$$= \mathbb{E} \left[ \left( \boldsymbol{G}(r_0) - \mathbf{1}_K^\top \otimes \boldsymbol{g}_K(r_0) \right) \otimes \boldsymbol{\phi}(s_1)^\top \Big| s_0 = s \right] \boldsymbol{w} + \frac{1}{K+1} \sum_{j=0}^K \mathbb{E} \left[ \boldsymbol{g}_j(r_0) \Big| s_0 = s \right],$$

or equivalently,

$$\tilde{\boldsymbol{p}}_{\boldsymbol{\theta}}(s) = \mathbb{E} \left[ \tilde{\boldsymbol{G}}(r_0) \boldsymbol{W}^\top \boldsymbol{\phi}(s_1) \Big| s_0 = s \right] + \frac{1}{K+1} \sum_{j=0}^K \mathbb{E} \left[ \boldsymbol{g}_j(r_0) \Big| s_0 = s \right]$$

$$= \mathbb{E} \left[ \tilde{\boldsymbol{G}}(r_0) \boldsymbol{C}^{-1} \boldsymbol{\Theta}^\top \boldsymbol{\phi}(s_1) \Big| s_0 = s \right] + \frac{1}{K+1} \sum_{j=0}^K \mathbb{E} \left[ \boldsymbol{g}_j(r_0) \Big| s_0 = s \right].$$

$\square$

## D.4 Linear-Categorical Projected Bellman Equation (Proof of Theorem 3.1)

Combining Proposition 3.1 with Proposition 3.2, we know that $\boldsymbol{\Theta}^\star$ is the unique solution to the following system of linear equations for $\boldsymbol{\Theta} \in \mathbb{R}^{d \times K}$

$$\boldsymbol{\Theta} = \boldsymbol{\Sigma}_{\boldsymbol{\phi}}^{-1} \mathbb{E}_{s \sim \mu_\pi} \left[ \boldsymbol{\phi}(s) \left( \tilde{\boldsymbol{p}}_{\boldsymbol{\theta}}(s) - \frac{1}{K+1} \mathbf{1}_K \right)^\top \boldsymbol{C}^\top \right]$$

$$= \boldsymbol{\Sigma}_{\boldsymbol{\phi}}^{-1} \mathbb{E}_{s \sim \mu_\pi} \left[ \boldsymbol{\phi}(s) \left( \frac{1}{K+1} \mathbf{1}_K - \mathbb{E} \left[ \tilde{\boldsymbol{G}}(r_0) \boldsymbol{C}^{-1} \boldsymbol{\Theta}^\top \boldsymbol{\phi}(s_1) \Big| s_0 = s \right] - \frac{1}{K+1} \sum_{j=0}^K \mathbb{E} \left[ \boldsymbol{g}_j(r_0) \Big| s_0 = s \right] \right)^\top \boldsymbol{C}^\top \right]$$

$$= \boldsymbol{\Sigma}_{\boldsymbol{\phi}}^{-1} \mathbb{E}_{s \sim \mu_\pi} \left[ \boldsymbol{\phi}(s) \boldsymbol{\phi}(s')^\top \boldsymbol{\Theta} \left( \boldsymbol{C} \tilde{\boldsymbol{G}}(r) \boldsymbol{C}^{-1} \right)^\top \right] + \frac{1}{K+1} \boldsymbol{\Sigma}_{\boldsymbol{\phi}}^{-1} \mathbb{E}_{s \sim \mu_\pi} \left[ \boldsymbol{\phi}(s) \left( \sum_{j=0}^K \boldsymbol{g}_j(r) - \mathbf{1}_K \right)^\top \boldsymbol{C}^\top \right],$$

or equivalently,

$$\boldsymbol{\Sigma}_{\boldsymbol{\phi}} \boldsymbol{\Theta} - \mathbb{E}_{s \sim \mu_\pi} \left[ \boldsymbol{\phi}(s) \boldsymbol{\phi}(s')^\top \boldsymbol{\Theta} \left( \boldsymbol{C} \tilde{\boldsymbol{G}}(r) \boldsymbol{C}^{-1} \right)^\top \right] = \frac{1}{K+1} \mathbb{E}_{s \sim \mu_\pi} \left[ \boldsymbol{\phi}(s) \left( \sum_{j=0}^K \boldsymbol{g}_j(r) - \mathbf{1}_K \right)^\top \boldsymbol{C}^\top \right],$$

which is the desired conclusion. The uniqueness and existence of the solution is guaranteed by the fact that the LHS is an invertible linear transformation of $\boldsymbol{\Theta}$, which is justified by Eqn. (28). 

## D.5 Proof of Proposition 3.3

*Proof.* By the basic inequality (Lemma H.1), we only need to show

$$\ell_{2,\mu_\pi}^2 \left( \boldsymbol{\eta}^\pi, \boldsymbol{\eta}_{\boldsymbol{\theta}^\star} \right) \leq \frac{\ell_{2,\mu_\pi}^2 \left( \boldsymbol{\eta}^\pi, \boldsymbol{\Pi}_{\boldsymbol{\phi},K}^\pi \boldsymbol{\eta}^\pi \right)}{1 - \gamma}$$

$$= \frac{\ell_{2,\mu_\pi}^2 \left( \boldsymbol{\eta}^\pi, \boldsymbol{\Pi}_K \boldsymbol{\eta}^\pi \right) + \ell_{2,\mu_\pi}^2 \left( \boldsymbol{\Pi}_K \boldsymbol{\eta}^\pi, \boldsymbol{\Pi}_{\boldsymbol{\phi},K}^\pi \boldsymbol{\eta}^\pi \right)}{1 - \gamma}$$

$$\leq \frac{1}{K(1 - \gamma)^2} + \frac{\ell_{2,\mu_\pi}^2 \left( \boldsymbol{\Pi}_K \boldsymbol{\eta}^\pi, \boldsymbol{\Pi}_{\boldsymbol{\phi},K}^\pi \boldsymbol{\eta}^\pi \right)}{1 - \gamma},$$

where we used Bellemare et al. [2023, Proposition 9.18 and Eqn. (5.28)]. $\square$

## D.6 Linear-Categorical Parametrization with Probability Mass Function Representation

We introduce new notations for linear-categorical parametrization with probability mass function (PMF) representation. Let $\boldsymbol{W} := \boldsymbol{\Theta} \boldsymbol{C}^{-\top} = (\boldsymbol{\theta}(0), \boldsymbol{\theta}(1) - \boldsymbol{\theta}(0), \cdots, \boldsymbol{\theta}(K-1) - \boldsymbol{\theta}(K-2)) \in \mathbb{R}^{d \times K}$ and the vectorization of $\boldsymbol{W}$, $\boldsymbol{w} := \operatorname{vec}(\boldsymbol{W}) = (\boldsymbol{C}^{-1} \otimes \boldsymbol{I}_d) \boldsymbol{\theta} \in \mathbb{R}^{dK}$. We abbreviate $\boldsymbol{p}_{\boldsymbol{\eta}_{\boldsymbol{\theta}}}$ as $\boldsymbol{p}_{\boldsymbol{w}}$ in this section. Then by Lemma A.2, for any $s \in \mathcal{S}$, it holds that

$$\boldsymbol{p}_{\boldsymbol{w}}(s) = \boldsymbol{W}^{\top} \boldsymbol{\phi}(s) + (K+1)^{-1} \mathbf{1}_K. \tag{21}$$

PMF and CDF representations are equivalent because $\boldsymbol{C}$ is invertible.

For any $\boldsymbol{\eta}_{\boldsymbol{w}_1}, \boldsymbol{\eta}_{\boldsymbol{w}_2} \in \mathscr{P}_{\boldsymbol{\phi}, K}^{\mathsf{sign}}$, by Proposition C.1,

$$
\begin{aligned}
\ell_{2,\mu_\pi}^2 \left(\boldsymbol{\eta}_{\boldsymbol{w}_1}, \boldsymbol{\eta}_{\boldsymbol{w}_2}\right) &= \mathbb{E}_{s \sim \mu_\pi} \left[ \ell_2^2 \left(\eta_{\boldsymbol{w}_1}(s), \eta_{\boldsymbol{w}_2}(s)\right) \right] \\
&= \iota_K \mathbb{E}_{s \sim \mu_\pi} \left[ \left\| \boldsymbol{C} \left(\boldsymbol{p}_{\boldsymbol{w}_1}(s) - \boldsymbol{p}_{\boldsymbol{w}_2}(s)\right) \right\|^2 \right] \\
&= \iota_K \operatorname{tr} \left( \boldsymbol{\Sigma}_{\boldsymbol{\phi}}^{\frac{1}{2}} (\boldsymbol{W}_1 - \boldsymbol{W}_2) \boldsymbol{C}^{\top} \boldsymbol{C} (\boldsymbol{W}_1 - \boldsymbol{W}_2)^{\top} \boldsymbol{\Sigma}_{\boldsymbol{\phi}}^{\frac{1}{2}} \right) \\
&= \iota_K \left\| \boldsymbol{w}_1 - \boldsymbol{w}_2 \right\|_{(\boldsymbol{C}^{\top} \boldsymbol{C}) \otimes \boldsymbol{\Sigma}_{\boldsymbol{\phi}}}^2,
\end{aligned}
\tag{22}
$$

hence the affine space $\left( \mathscr{P}_{\boldsymbol{\phi}, K}^{\mathsf{sign}}, \ell_{2,\mu_\pi} \right)$ is isometric with the Euclidean space $\left( \mathbb{R}^{Kd}, \sqrt{\iota_K} \left\| \cdot \right\|_{(\boldsymbol{C}^{\top} \boldsymbol{C}) \otimes \boldsymbol{\Sigma}_{\boldsymbol{\phi}}} \right)$ if we consider the PMF representation.

Following the proof of Lemma D.1 in Appendix D.2, we can also derive the gradient when we use the PMF parametrization:

$$
\begin{aligned}
\nabla_{\boldsymbol{w}} \ell_2^2 \left(\eta_{\boldsymbol{w}}(s), \eta(s)\right) &= \nabla_{\boldsymbol{w}} \ell_2^2 \left(\eta_{\boldsymbol{w}}(s), \boldsymbol{\Pi}_K \eta(s)\right) \\
&= \iota_K \nabla_{\boldsymbol{w}} \left\| \boldsymbol{C} \left(\boldsymbol{p}_{\boldsymbol{w}}(s) - \boldsymbol{p}_{\boldsymbol{\eta}}(s)\right) \right\|^2 \\
&= 2\iota_K \left(\boldsymbol{I}_K \otimes \boldsymbol{\phi}(s)\right) \boldsymbol{C}^{\top} \boldsymbol{C} \left(\boldsymbol{p}_{\boldsymbol{w}}(s) - \boldsymbol{p}_{\boldsymbol{\eta}}(s)\right) \\
&= 2\iota_K \left(\boldsymbol{I}_K \otimes \boldsymbol{\phi}(s)\right) \boldsymbol{C}^{\top} \boldsymbol{C} \left( \left(\boldsymbol{I}_K \otimes \boldsymbol{\phi}(s)\right)^{\top} \boldsymbol{w} + \frac{1}{K+1} \mathbf{1}_K - \boldsymbol{p}_{\boldsymbol{\eta}}(s) \right) \\
&= 2\iota_K \left\{ \left[ (\boldsymbol{C}^{\top} \boldsymbol{C}) \otimes (\boldsymbol{\phi}(s) \boldsymbol{\phi}(s)^{\top}) \right] \boldsymbol{w} + \left[ \left( \boldsymbol{C}^{\top} \boldsymbol{C} \left( \frac{1}{K+1} \mathbf{1}_K - \boldsymbol{p}_{\boldsymbol{\eta}}(s) \right) \right) \otimes \boldsymbol{\phi}(s) \right] \right\},
\end{aligned}
$$

$$
\begin{aligned}
\nabla_{\boldsymbol{W}} \ell_2^2 \left(\eta_{\boldsymbol{w}}(s), \eta(s)\right) &= 2\iota_K \boldsymbol{\phi}(s) \left(\boldsymbol{p}_{\boldsymbol{w}}(s) - \boldsymbol{p}_{\boldsymbol{\eta}}(s)\right)^{\top} \boldsymbol{C}^{\top} \boldsymbol{C} \\
&= 2\iota_K \boldsymbol{\phi}(s) \left[ \boldsymbol{\phi}(s)^{\top} \boldsymbol{W} + \left( \frac{1}{K+1} \mathbf{1}_K - \boldsymbol{p}_{\boldsymbol{\eta}}(s) \right)^{\top} \right] \boldsymbol{C}^{\top} \boldsymbol{C}.
\end{aligned}
\tag{23}
$$

## D.7 Stochastic Semi-Gradient Descent with Linear Function Approximation

We denote by $\boldsymbol{\mathcal{T}}_t^\pi$ the corresponding empirical distributional Bellman operator at the $t$-th iteration, which satisfies

$$\left[\boldsymbol{\mathcal{T}}_t^\pi \boldsymbol{\eta}\right](s_t) = (b_{r_t, \gamma})_{\#} (\boldsymbol{\eta}(s_{t+1})), \quad \forall \boldsymbol{\eta} \in \mathscr{P}^{\mathcal{S}}. \tag{24}$$

### D.7.1 Probability Mass Function Representation

Consider the stochastic semi-gradient descent (SSGD) with the probability mass function (PMF) representation

$$\boldsymbol{W}_t \leftarrow \boldsymbol{W}_{t-1} - \alpha \nabla_{\boldsymbol{W}} \ell_2^2 \left( \eta_{\boldsymbol{w}_{t-1}}(s_t), \left[\boldsymbol{\mathcal{T}}_t^\pi \boldsymbol{\eta}_{\boldsymbol{w}_{t-1}}\right](s_t) \right),$$

where $\nabla_{\boldsymbol{W}}$ stands for taking gradient w.r.t. $\boldsymbol{W}_{t-1} \in \mathbb{R}^{d \times K}$ in the first term $\eta_{\boldsymbol{w}_{t-1}}(s_t)$ (the second term is regarding as a constant, that's why we call it a semi-gradient). We can check that $\nabla_{\boldsymbol{W}} \ell_2^2 \left( \eta_{\boldsymbol{w}_{t-1}}(s_t), \left[\boldsymbol{\mathcal{T}}_t^\pi \boldsymbol{\eta}_{\boldsymbol{w}_{t-1}}\right](s_t) \right)$ is an unbiased estimate of $\nabla_{\boldsymbol{W}} \ell_{2,\mu_\pi}^2 \left( \boldsymbol{\eta}_{\boldsymbol{w}_{t-1}}, \boldsymbol{\mathcal{T}}^\pi \boldsymbol{\eta}_{\boldsymbol{w}_{t-1}} \right)$.

Now, let us compute the gradient term. By Eqn. (23), we have

$$\nabla_{\boldsymbol{W}} \ell_2^2 \left( \eta_{\boldsymbol{w}_{t-1}}(s_t), \left[\boldsymbol{\mathcal{T}}_t^\pi \boldsymbol{\eta}_{\boldsymbol{w}_{t-1}}\right](s_t) \right) = 2\iota_K \boldsymbol{\phi}(s_t) \left( \boldsymbol{p}_{\boldsymbol{w}_{t-1}}(s_t) - \boldsymbol{p}_{\boldsymbol{\mathcal{T}}_t^\pi \boldsymbol{\eta}_{\boldsymbol{w}_{t-1}}}(s_t) \right)^{\top} \boldsymbol{C}^{\top} \boldsymbol{C}$$

$$= 2\iota_K \phi(s_t) \left[ \phi(s_t)^\top \boldsymbol{W}_{t-1} + \left( \frac{1}{K+1} \mathbf{1}_K - \boldsymbol{p}_{\mathcal{T}_t^\pi \boldsymbol{\eta}_{\boldsymbol{w}_{t-1}}}(s_t) \right)^\top \right] \boldsymbol{C}^\top \boldsymbol{C}.$$

where $\boldsymbol{p}_{\mathcal{T}_t^\pi \boldsymbol{\eta}_{\boldsymbol{w}_{t-1}}}(s_t) = \boldsymbol{p}_{\boldsymbol{\Pi}_K \mathcal{T}_t^\pi \boldsymbol{\eta}_{\boldsymbol{w}_{t-1}}}(s_t) = \left( p_k \left( \left[ \mathcal{T}_t^\pi \boldsymbol{\eta}_{\boldsymbol{w}_{t-1}} \right](s_t) \right) \right)_{k=0}^{K-1} \in \mathbb{R}^K$. Now, we turn to compute $\boldsymbol{p}_{\mathcal{T}_t^\pi \boldsymbol{\eta}_{\boldsymbol{w}_{t-1}}}(s_t)$. According to Eqn. (8),

$$\begin{aligned} p_k \left( \left[ \mathcal{T}_t^\pi \boldsymbol{\eta}_{\boldsymbol{w}_{t-1}} \right](s_t) \right) &= \mathbb{E}_{X \sim \left[ \mathcal{T}_t^\pi \boldsymbol{\eta}_{\boldsymbol{w}_{t-1}} \right](s_t)} \left[ \left( 1 - \left| \frac{X - x_k}{\iota_K} \right| \right)_+ \right] \\ &= \mathbb{E}_{G \sim \eta_{\boldsymbol{w}_{t-1}}(s_{t+1})} \left[ \left( 1 - \left| \frac{r_t + \gamma G - x_k}{\iota_K} \right| \right)_+ \right] \\ &= \sum_{j=0}^{K} p_j(s_{t+1}; \boldsymbol{w}_{t-1}) g_{j,k}(r_t) \\ &= g_{K,k}(r_t) + \sum_{j=0}^{K-1} p_j(s_{t+1}; \boldsymbol{w}_{t-1}) \left( g_{j,k}(r_t) - g_{K,k}(r_t) \right), \end{aligned}$$

which has the same form as Eqn. (20). Following the proof of Proposition 3.2 in Appendix D.3, one can show that

$$\boldsymbol{p}_{\mathcal{T}_t^\pi \boldsymbol{\eta}_{\boldsymbol{w}_{t-1}}}(s_t) = \tilde{\boldsymbol{G}}(r_t) \boldsymbol{W}_{t-1}^\top \phi(s_{t+1}) + \frac{1}{K+1} \sum_{j=0}^{K} \boldsymbol{g}_j(r_t). \tag{25}$$

Hence, the update scheme is

$$\begin{aligned} \boldsymbol{W}_t \leftarrow & \boldsymbol{W}_{t-1} - 2\iota_K \alpha \phi(s_t) \left( \boldsymbol{p}_{\boldsymbol{w}_{t-1}}(s_t) - \boldsymbol{p}_{\mathcal{T}_t^\pi \boldsymbol{\eta}_{\boldsymbol{w}_{t-1}}}(s_t) \right)^\top \boldsymbol{C}^\top \boldsymbol{C} \\ = & \boldsymbol{W}_{t-1} - 2\iota_K \alpha \phi(s_t) \left[ \phi(s_t)^\top \boldsymbol{W}_{t-1} - \phi(s_{t+1})^\top \boldsymbol{W}_{t-1} \tilde{\boldsymbol{G}}^\top(r_t) - \frac{1}{K+1} \left( \sum_{j=0}^{K} \boldsymbol{g}_j(r_t) - \mathbf{1}_K \right)^\top \right] \boldsymbol{C}^\top \boldsymbol{C}. \end{aligned}$$
$$\tag{26}$$

Note that our `Linear-CTD` (Eqn. (13)) is equivalent to

$$\boldsymbol{W}_t \leftarrow \boldsymbol{W}_{t-1} - \alpha \phi(s_t) \left[ \phi(s_t)^\top \boldsymbol{W}_{t-1} - \phi(s_{t+1})^\top \boldsymbol{W}_{t-1} \tilde{\boldsymbol{G}}^\top(r_t) - \frac{1}{K+1} \left( \sum_{j=0}^{K} \boldsymbol{g}_j(r_t) - \mathbf{1}_K \right)^\top \right],$$
$$\tag{27}$$

in the PMF representation. Compared to Eqn. (27), the SSGD (Eqn. (26)) has an additional $\boldsymbol{C}^\top \boldsymbol{C}$, and the step size is $2\iota_K \alpha$.

### D.7.2 Cumulative Distribution Function Representation

Consider the SSGD with the CDF representation

$$\boldsymbol{\Theta}_t \leftarrow \boldsymbol{\Theta}_{t-1} - \alpha \nabla_{\boldsymbol{\Theta}} \ell_2^2 \left( \eta_{\boldsymbol{\theta}_{t-1}}(s_t), \left[ \mathcal{T}_t^\pi \boldsymbol{\eta}_{\boldsymbol{\theta}_{t-1}} \right](s_t) \right),$$

where $\nabla_{\boldsymbol{\Theta}}$ stands for taking gradient w.r.t. $\boldsymbol{\Theta}_{t-1} = \boldsymbol{\theta}_{t-1} \boldsymbol{C}^\top \in \mathbb{R}^{d \times K}$ in the first term $\eta_{\boldsymbol{\theta}_{t-1}}(s_t)$ (the second term is regarding as a constant). One can check that $\nabla_{\boldsymbol{\Theta}} \ell_2^2 \left( \eta_{\boldsymbol{\theta}_{t-1}}(s_t), \left[ \mathcal{T}_t^\pi \boldsymbol{\eta}_{\boldsymbol{\theta}_{t-1}} \right](s_t) \right)$ is an unbiased estimate of $\nabla_{\boldsymbol{\Theta}} \ell_{2, \mu_\pi}^2 \left( \boldsymbol{\eta}_{\boldsymbol{\theta}_{t-1}}, \mathcal{T}^\pi \boldsymbol{\eta}_{\boldsymbol{\theta}_{t-1}} \right)$.

Now, let us compute the gradient term. By Eqn. (19) and Eqn. (25), we have

$$\begin{aligned} & \nabla_{\boldsymbol{\Theta}} \ell_2^2 \left( \eta_{\boldsymbol{\theta}_{t-1}}(s_t), \left[ \mathcal{T}_t^\pi \boldsymbol{\eta}_{\boldsymbol{\theta}_{t-1}} \right](s_t) \right) \\ & = 2\iota_K \phi(s_t) \left( \boldsymbol{F}_{\boldsymbol{\theta}_{t-1}}(s_t) - \boldsymbol{F}_{\mathcal{T}_t^\pi \boldsymbol{\eta}_{\boldsymbol{\theta}_{t-1}}}(s_t) \right)^\top \\ & = 2\iota_K \phi(s_t) \left[ \phi(s_t)^\top \boldsymbol{\Theta}_{t-1} + \left( \frac{1}{K+1} \mathbf{1}_K - \boldsymbol{p}_{\mathcal{T}_t^\pi \boldsymbol{\eta}_{\boldsymbol{\theta}_{t-1}}}(s) \right)^\top \boldsymbol{C}^\top \right] \end{aligned}$$

$$= 2\iota_K \phi(s_t) \left[ \phi(s_t)^\top \Theta_{t-1} - \phi(s_{t+1})^\top \Theta_{t-1} C^{-\top} \tilde{G}^\top (r_t) C^\top - \frac{1}{K+1} \left( \sum_{j=0}^{K} g_j(r_t) - \mathbf{1}_K \right)^\top C^\top \right].$$

Hence, the update scheme is

$$\Theta_t \leftarrow \Theta_{t-1} - 2\iota_K \alpha \phi(s_t) \left( F_{\theta_{t-1}}(s_t) - F_{\mathcal{T}_t^\pi \eta_{\theta_{t-1}}}(s_t) \right)^\top$$

$$= \Theta_{t-1} - 2\iota_K \alpha \phi(s_t) \left[ \phi(s_t)^\top \Theta_{t-1} - \phi(s_{t+1})^\top \Theta_{t-1} C^{-\top} \tilde{G}^\top (r_t) C^\top - \frac{1}{K+1} \left( \sum_{j=0}^{K} g_j(r_t) - \mathbf{1}_K \right)^\top C^\top \right],$$

which has the same form as Linear-CTD (Eqn. (13)) with the step size $2\alpha\iota_K$.

### D.8 `Linear-CTD` is mean-preserving

We will show that our `Linear-CTD` is mean-preserving, which was first discovered by Lyle et al. [2019, Proposition 8]. In this section, we assume the first coordinate of the feature is a constant, *i.e.*, $\phi_1(s) = 1/\sqrt{d}$ for any $s \in \mathcal{S}$. As stated before, we will always assume this to ensure $\mathscr{P}_{\phi,K}$ can be uniquely defined.

**Proposition D.3.** *Let $V_\theta := (V_\theta(s))_{s\in\mathcal{S}}$ be the value function corresponding to $\theta$, i.e., $V_\theta(s) = \mathbb{E}_{G\sim\eta_\theta(s)}[G]$, then for any initialization of the `Linear-TD` parameter $\psi_0$, there exists a (not unique) corresponding `Linear-CTD` parameter $\theta_0$ such that $V_{\theta_0} = V_{\psi_0}$, furthermore, for any $t \geq 1$ and even number $T \geq 2$, it holds that*

$$V_{\theta_t} = V_{\psi_t}, \quad V_{\bar{\theta}_T} = V_{\bar{\psi}_T}.$$

*Proof of Proposition D.3.* Recall that the updating scheme of `Linear-TD` is given by

$$\psi_t \leftarrow \psi_{t-1} - \alpha \phi(s_t) \left( V_{\psi_{t-1}}(s_t) - r_t - \gamma V_{\psi_{t-1}}(s_{t+1}) \right).$$

And the updating scheme of `Linear-CTD` is given by

$$\Theta_t \leftarrow \Theta_{t-1} - \alpha \phi(s_t) \left( F_{\theta_{t-1}}(s_t) - F_{\mathcal{T}_t^\pi \eta_{\theta_{t-1}}}(s_t) \right)^\top.$$

Let $V_\theta := (V_\theta(s))_{s\in\mathcal{S}}$ be the value function corresponding to $\theta$, we have

$$V_\theta(s) = \iota_K \sum_{k=0}^{K-1} (1 - F_k(s; \theta))$$
$$= \iota_K \left( \mathbf{1}_K - F_\theta(s) \right)^\top \mathbf{1}_K$$
$$= \frac{1}{2(1-\gamma)} - \iota_K \phi(s)^\top \Theta \mathbf{1}_K.$$

Hence, if we take $\psi_{0,1} = \frac{\sqrt{d}}{2(1-\gamma)}$, $\psi_{0,i} = 0$ for any $i \in \{2, \ldots, d\}$, and $\theta_0 = \mathbf{0}_{dK}$, it holds that

$$V_{\psi_0} = V_{\theta_0} = \frac{1}{2(1-\gamma)} \mathbf{1}_\mathcal{S}.$$

We can also show that for any $\psi_0 \in \mathbb{R}^d$, there exists $\theta_0 \in \mathbb{R}^{d\times K}$ such that $V_{\psi_0} = V_{\theta_0}$. That is we need to find $\theta_0$ such that for any $s \in \mathcal{S}$,

$$\iota_K \sum_{k=0}^{K-1} \phi(s)^\top \theta_0(k) = \frac{1}{2(1-\gamma)} - \phi(s)^\top \psi_0.$$

We can take $\theta_0$ satisfying the following equations to make the above equation hold

$$\iota_K \sum_{k=0}^{K-1} \theta_0(k, 1) = \frac{\sqrt{d}}{2(1-\gamma)} - \psi_{0,1},$$

and

$$\iota_K \sum_{k=0}^{K-1} \boldsymbol{\theta}_0(k)_{-1} = -\boldsymbol{\psi}_{0,-1}.$$

Furthermore, for any $t \geq 1$, we have for any $s \in \mathcal{S}$,

$$
\begin{aligned}
V_{\boldsymbol{\theta}_t}(s) =& \frac{1}{2(1-\gamma)} - \iota_K \boldsymbol{\phi}(s)^\mathsf{T} \boldsymbol{\Theta}_t \mathbf{1}_K \\
=& \frac{1}{2(1-\gamma)} - \iota_K \boldsymbol{\phi}(s)^\mathsf{T} \left( \boldsymbol{\Theta}_{t-1} - \alpha \boldsymbol{\phi}(s_t) \left( \boldsymbol{F}_{\boldsymbol{\theta}_{t-1}}(s_t) - \boldsymbol{F}_{\mathcal{T}_t^\pi \boldsymbol{\eta}_{\boldsymbol{\theta}_{t-1}}}(s_t) \right)^\mathsf{T} \right) \mathbf{1}_K \\
=& V_{\boldsymbol{\theta}_{t-1}}(s) + \alpha \iota_K \left( \boldsymbol{\phi}(s)^\mathsf{T} \boldsymbol{\phi}(s_t) \right) \left( \boldsymbol{F}_{\boldsymbol{\theta}_{t-1}}(s_t) - \boldsymbol{F}_{\mathcal{T}_t^\pi \boldsymbol{\eta}_{\boldsymbol{\theta}_{t-1}}}(s_t) \right)^\mathsf{T} \mathbf{1}_K
\end{aligned}
$$

$$
\begin{aligned}
V_{\boldsymbol{\psi}_t}(s) =& \boldsymbol{\phi}(s)^\mathsf{T} \boldsymbol{\psi}_t \\
=& \boldsymbol{\phi}(s)^\mathsf{T} \left( \boldsymbol{\psi}_{t-1} - \alpha \boldsymbol{\phi}(s_t) \left( V_{\boldsymbol{\psi}_{t-1}}(s_t) - r_t - \gamma V_{\boldsymbol{\psi}_{t-1}}(s_{t+1}) \right) \right) \\
=& V_{\boldsymbol{\psi}_{t-1}}(s) - \alpha \left( \boldsymbol{\phi}(s)^\mathsf{T} \boldsymbol{\phi}(s_t) \right) \left( V_{\boldsymbol{\psi}_{t-1}}(s_t) - r_t - \gamma V_{\boldsymbol{\psi}_{t-1}}(s_{t+1}) \right).
\end{aligned}
$$

We need to check that, if $\boldsymbol{V}_{\boldsymbol{\theta}_{t-1}} = \boldsymbol{V}_{\boldsymbol{\psi}_{t-1}}$, it holds that

$$\iota_K \left( \boldsymbol{F}_{\mathcal{T}_t^\pi \boldsymbol{\eta}_{\boldsymbol{\theta}_{t-1}}}(s_t) - \boldsymbol{F}_{\boldsymbol{\theta}_{t-1}}(s_t) \right)^\mathsf{T} \mathbf{1}_K = V_{\boldsymbol{\psi}_{t-1}}(s_t) - r_t - \gamma V_{\boldsymbol{\psi}_{t-1}}(s_{t+1}),$$

which is the direct corollary of the following fact

$$\text{LHS} = V_{\boldsymbol{\psi}_t}(s_t) - \mathbb{E}_{X \sim \boldsymbol{\Pi}_K (b_{r,\gamma})_\# \boldsymbol{\eta}_{\boldsymbol{\theta}_{t-1}}(s_{t+1})}[X] = V_{\boldsymbol{\psi}_t}(s_t) - r_t - \gamma V_{\boldsymbol{\theta}_{t-1}}(s_{t+1}),$$

by Lemma D.2. And we can obtain $\boldsymbol{V}_{\bar{\boldsymbol{\theta}}_T} = \boldsymbol{V}_{\bar{\boldsymbol{\psi}}_T}$ by using the facts that $\mathscr{P}_{\phi,K}^{\text{sign}}$ is an affine space, $\mathscr{V}_\phi$ is a linear space and taking expectation is a linear operator. $\qquad\square$

**Lemma D.2.** *For any $\nu \in \mathscr{P}^{\text{sign}}$, it holds that*

$$\mathbb{E}_{X \sim \nu}[X] = \mathbb{E}_{X \sim \boldsymbol{\Pi}_K \nu}[X].$$

*Proof.* By Eqn. (8), $\boldsymbol{\Pi}_K \nu \in \mathscr{P}_K^{\text{sign}}$ is uniquely identified with a vector $\boldsymbol{p}_\nu = (p_k(\nu))_{k=0}^{K-1} \in \mathbb{R}^K$, where

$$p_k(\nu) = \int_{[0,(1-\gamma)^{-1}]} (1 - |(x-x_k)/\iota_K|)_+ \nu(dx).$$

Hence, for any $x \in [0, (1-\gamma)^{-1}]$, we define $x_{\text{lb}} := \max\{y \in \{x_0, \ldots, x_K\} : x \leq y\}$, then

$$
\begin{aligned}
\sum_{k=0}^K x_k (1 - |(x-x_k)/\iota_K|)_+ =& x_{\text{lb}} \left( 1 - \frac{x - x_{\text{lb}}}{\iota_K} \right) + (x_{\text{lb}} + \iota_K) \left( 1 - \frac{x_{\text{lb}} + \iota_K - x}{\iota_K} \right) \\
=& x_{\text{lb}} + \iota_K \left( 1 - \frac{x_{\text{lb}} + \iota_K - x}{\iota_K} \right) \\
=& x,
\end{aligned}
$$

therefore,

$$
\begin{aligned}
\mathbb{E}_{X \sim \boldsymbol{\Pi}_K \nu}[X] =& \sum_{k=0}^K x_k \int_{[0,(1-\gamma)^{-1}]} (1 - |(x-x_k)/\iota_K|)_+ \nu(dx) \\
=& \int_{[0,(1-\gamma)^{-1}]} \sum_{k=0}^K x_k (1 - |(x-x_k)/\iota_K|)_+ \nu(dx) \\
=& \int_{[0,(1-\gamma)^{-1}]} x \nu(dx) \\
=& \mathbb{E}_{X \sim \nu}[X].
\end{aligned}
$$

$\qquad\square$

# E  Omitted Results and Proofs in Section 4

## E.1  Proof of Lemma 5.1

*Proof.* By Lemma H.1 and Eqn. (22), we have

$$
\begin{aligned}
(\mathcal{L}(\boldsymbol{\theta}))^2 &= W_{1,\mu_\pi}^2\left(\boldsymbol{\eta_\theta}, \boldsymbol{\eta_{\theta^\star}}\right) \\
&\leq \frac{1}{1-\gamma} \ell_{2,\mu_\pi}^2\left(\boldsymbol{\eta_\theta}, \boldsymbol{\eta_{\theta^\star}}\right) \\
&= \frac{\iota_K}{1-\gamma} \operatorname{tr}\left((\boldsymbol{\Theta} - \boldsymbol{\Theta}^\star)^\top \boldsymbol{\Sigma_\phi} (\boldsymbol{\Theta} - \boldsymbol{\Theta}^\star)\right) \\
&= \frac{1}{K(1-\gamma)^2} \|\boldsymbol{\theta} - \boldsymbol{\theta}^\star\|_{\boldsymbol{I}_K \otimes \boldsymbol{\Sigma_\phi}}^2 .
\end{aligned}
$$

We only need to show that

$$
\boldsymbol{I}_K \otimes \boldsymbol{\Sigma_\phi} \preccurlyeq \frac{1}{(1-\sqrt{\gamma})^2} \bar{\boldsymbol{A}}^\top \left(\boldsymbol{I}_K \otimes \boldsymbol{\Sigma_\phi}^{-1}\right) \bar{\boldsymbol{A}} \left(\preccurlyeq \frac{4}{(1-\gamma)^2 \lambda_{\min}} \bar{\boldsymbol{A}}^\top \bar{\boldsymbol{A}}\right), \tag{28}
$$

or equivalently,

$$
\left(\boldsymbol{I}_K \otimes \boldsymbol{\Sigma_\phi}^{-\frac{1}{2}}\right) \bar{\boldsymbol{A}}^\top \left(\boldsymbol{I}_K \otimes \boldsymbol{\Sigma_\phi}^{-1}\right) \bar{\boldsymbol{A}} \left(\boldsymbol{I}_K \otimes \boldsymbol{\Sigma_\phi}^{-\frac{1}{2}}\right) \succcurlyeq (1-\sqrt{\gamma})^2 \boldsymbol{I}_{dK}.
$$

Recall

$$
\bar{\boldsymbol{A}} = (\boldsymbol{I}_K \otimes \boldsymbol{\Sigma_\phi}) - \mathbb{E}_{s,r,s'}\left[\left(\boldsymbol{C}\tilde{\boldsymbol{G}}(r)\boldsymbol{C}^{-1}\right) \otimes \left(\boldsymbol{\phi}(s)\boldsymbol{\phi}(s')^\top\right)\right],
$$

then for any $\boldsymbol{\theta} \in \mathbb{R}^{dK}$ with $\|\boldsymbol{\theta}\| = 1$,

$$
\begin{aligned}
&\boldsymbol{\theta}^\top \left(\boldsymbol{I}_K \otimes \boldsymbol{\Sigma_\phi}^{-\frac{1}{2}}\right) \bar{\boldsymbol{A}}^\top \left(\boldsymbol{I}_K \otimes \boldsymbol{\Sigma_\phi}^{-1}\right) \bar{\boldsymbol{A}} \left(\boldsymbol{I}_K \otimes \boldsymbol{\Sigma_\phi}^{-\frac{1}{2}}\right) \boldsymbol{\theta} \\
&= \left\|\left(\boldsymbol{I}_K \otimes \boldsymbol{\Sigma_\phi}^{-\frac{1}{2}}\right) \bar{\boldsymbol{A}} \left(\boldsymbol{I}_K \otimes \boldsymbol{\Sigma_\phi}^{-\frac{1}{2}}\right) \boldsymbol{\theta}\right\|^2 \\
&= \left\|\boldsymbol{\theta} - \mathbb{E}_{s,r,s'}\left[\left(\boldsymbol{C}\tilde{\boldsymbol{G}}(r)\boldsymbol{C}^{-1}\right) \otimes \left(\boldsymbol{\Sigma_\phi}^{-\frac{1}{2}}\boldsymbol{\phi}(s)\boldsymbol{\phi}(s')^\top\boldsymbol{\Sigma_\phi}^{-\frac{1}{2}}\right)\right] \boldsymbol{\theta}\right\|^2 \\
&\geq \left(1 - \left\|\mathbb{E}_{s,r,s'}\left[\left(\boldsymbol{C}\tilde{\boldsymbol{G}}(r)\boldsymbol{C}^{-1}\right) \otimes \left(\boldsymbol{\Sigma_\phi}^{-\frac{1}{2}}\boldsymbol{\phi}(s)\boldsymbol{\phi}(s')^\top\boldsymbol{\Sigma_\phi}^{-\frac{1}{2}}\right)\right] \boldsymbol{\theta}\right\|\right)^2 .
\end{aligned}
$$

It suffices to show that

$$
\left\|\mathbb{E}_{s,r,s'}\left[\left(\boldsymbol{C}\tilde{\boldsymbol{G}}(r)\boldsymbol{C}^{-1}\right) \otimes \left(\boldsymbol{\Sigma_\phi}^{-\frac{1}{2}}\boldsymbol{\phi}(s)\boldsymbol{\phi}(s')^\top\boldsymbol{\Sigma_\phi}^{-\frac{1}{2}}\right)\right]\right\| \leq \sqrt{\gamma}. \tag{29}
$$

For brevity, we abbreviate $\boldsymbol{C}\tilde{\boldsymbol{G}}(r)\boldsymbol{C}^{-1}$ as $\boldsymbol{Y}(r) = (y_{ij}(r))_{i,j=1}^K \in \mathbb{R}^{K \times K}$. Thus, it suffices to show that

$$
\left\|\mathbb{E}_{s,r,s'}\left[\boldsymbol{Y}(r) \otimes \left(\boldsymbol{\Sigma_\phi}^{-\frac{1}{2}}\boldsymbol{\phi}(s)\boldsymbol{\phi}(s')^\top\boldsymbol{\Sigma_\phi}^{-\frac{1}{2}}\right)\right]\right\| \leq \sqrt{\gamma}.
$$

For any vectors $\boldsymbol{w} = \left(\boldsymbol{w}(0)^\top, \cdots, \boldsymbol{w}(K-1)^\top\right)$ and $\boldsymbol{v} = \left(\boldsymbol{v}(0)^\top, \cdots, \boldsymbol{v}(K-1)^\top\right)$ in $\mathbb{R}^{dK}$, we define the corresponding matrices $\boldsymbol{W} = (\boldsymbol{w}(0), \cdots, \boldsymbol{w}(K-1))$ and $\boldsymbol{V} = (\boldsymbol{v}(0), \cdots, \boldsymbol{v}(K-1))$ in $\mathbb{R}^{d \times K}$, then $\|\boldsymbol{w}\| = \|\boldsymbol{W}\|_F = \sqrt{\operatorname{tr}(\boldsymbol{W}^\top\boldsymbol{W})}$ and $\|\boldsymbol{v}\| = \|\boldsymbol{V}\|_F = \sqrt{\operatorname{tr}(\boldsymbol{V}^\top\boldsymbol{V})}$. With these notations, we have

$$
\begin{aligned}
&\left\|\mathbb{E}_{s,r,s'}\left[\boldsymbol{Y}(r) \otimes \left(\boldsymbol{\Sigma_\phi}^{-\frac{1}{2}}\boldsymbol{\phi}(s)\boldsymbol{\phi}(s')^\top\boldsymbol{\Sigma_\phi}^{-\frac{1}{2}}\right)\right]\right\| \\
&= \sup_{\|\boldsymbol{w}\|=\|\boldsymbol{v}\|=1} \boldsymbol{w}^\top \mathbb{E}_{s,r,s'}\left[\boldsymbol{Y}(r) \otimes \left(\boldsymbol{\Sigma_\phi}^{-\frac{1}{2}}\boldsymbol{\phi}(s)\boldsymbol{\phi}(s')^\top\boldsymbol{\Sigma_\phi}^{-\frac{1}{2}}\right)\right] \boldsymbol{v} \\
&= \sup_{\|\boldsymbol{w}\|=\|\boldsymbol{v}\|=1} \mathbb{E}_{s,r,s'}\left[\sum_{i,j=1}^K y_{ij}(r)\boldsymbol{w}(i)^\top\boldsymbol{\Sigma_\phi}^{-\frac{1}{2}}\boldsymbol{\phi}(s)\boldsymbol{\phi}(s')^\top\boldsymbol{\Sigma_\phi}^{-\frac{1}{2}}\boldsymbol{v}(j)\right],
\end{aligned}
$$

it is easy to check that

$$
\boldsymbol{w}(i)^\top\boldsymbol{\Sigma_\phi}^{-\frac{1}{2}}\boldsymbol{\phi}(s)\boldsymbol{\phi}(s')^\top\boldsymbol{\Sigma_\phi}^{-\frac{1}{2}}\boldsymbol{v}(j) = \left(\boldsymbol{W}^\top\boldsymbol{\Sigma_\phi}^{-\frac{1}{2}}\boldsymbol{\phi}(s)\boldsymbol{\phi}(s')^\top\boldsymbol{\Sigma_\phi}^{-\frac{1}{2}}\boldsymbol{V}\right)_{ij},
$$

hence

$$\left\| \mathbb{E}_{s,r,s'} \left[ \boldsymbol{Y}(r) \otimes \left( \boldsymbol{\Sigma}_{\boldsymbol{\phi}}^{-\frac{1}{2}} \boldsymbol{\phi}(s)\boldsymbol{\phi}(s')^\top \boldsymbol{\Sigma}_{\boldsymbol{\phi}}^{-\frac{1}{2}} \right) \right] \right\|$$

$$= \sup_{\|\boldsymbol{W}\|_F=\|\boldsymbol{V}\|_F=1} \mathbb{E}_{s,r,s'} \left[ \sum_{i,j=1}^{K} y_{ij}(r) \left( \boldsymbol{W}^\top \boldsymbol{\Sigma}_{\boldsymbol{\phi}}^{-\frac{1}{2}} \boldsymbol{\phi}(s)\boldsymbol{\phi}(s')^\top \boldsymbol{\Sigma}_{\boldsymbol{\phi}}^{-\frac{1}{2}} \boldsymbol{V} \right)_{ij} \right]$$

$$= \sup_{\|\boldsymbol{W}\|_F=\|\boldsymbol{V}\|_F=1} \mathbb{E}_{s,r,s'} \left[ \mathrm{tr} \left( \boldsymbol{Y}(r)\boldsymbol{V}^\top \boldsymbol{\Sigma}_{\boldsymbol{\phi}}^{-\frac{1}{2}} \boldsymbol{\phi}(s')\boldsymbol{\phi}(s)^\top \boldsymbol{\Sigma}_{\boldsymbol{\phi}}^{-\frac{1}{2}} \boldsymbol{W} \right) \right]$$

$$= \sup_{\|\boldsymbol{W}\|_F=\|\boldsymbol{V}\|_F=1} \mathbb{E}_{s,r,s'} \left[ \boldsymbol{\phi}(s)^\top \boldsymbol{\Sigma}_{\boldsymbol{\phi}}^{-\frac{1}{2}} \boldsymbol{W} \boldsymbol{Y}(r)\boldsymbol{V}^\top \boldsymbol{\Sigma}_{\boldsymbol{\phi}}^{-\frac{1}{2}} \boldsymbol{\phi}(s') \right]$$

$$\le \sup_{\|\boldsymbol{W}\|_F=\|\boldsymbol{V}\|_F=1} \mathbb{E}_{s,r,s'} \left[ \left\| \boldsymbol{W}^\top \boldsymbol{\Sigma}_{\boldsymbol{\phi}}^{-\frac{1}{2}} \boldsymbol{\phi}(s) \right\| \left\| \boldsymbol{Y}(r) \right\| \left\| \boldsymbol{V}^\top \boldsymbol{\Sigma}_{\boldsymbol{\phi}}^{-\frac{1}{2}} \boldsymbol{\phi}(s') \right\| \right]$$

$$\le \sqrt{\gamma} \sup_{\|\boldsymbol{W}\|_F=\|\boldsymbol{V}\|_F=1} \mathbb{E}_{s,s'} \left[ \left\| \boldsymbol{W}^\top \boldsymbol{\Sigma}_{\boldsymbol{\phi}}^{-\frac{1}{2}} \boldsymbol{\phi}(s) \right\| \left\| \boldsymbol{V}^\top \boldsymbol{\Sigma}_{\boldsymbol{\phi}}^{-\frac{1}{2}} \boldsymbol{\phi}(s') \right\| \right]$$

$$\le \sqrt{\gamma} \sup_{\|\boldsymbol{W}\|_F=\|\boldsymbol{V}\|_F=1} \sqrt{ \mathbb{E}_s \left[ \left\| \boldsymbol{W}^\top \boldsymbol{\Sigma}_{\boldsymbol{\phi}}^{-\frac{1}{2}} \boldsymbol{\phi}(s) \right\|^2 \right] \mathbb{E}_{s'} \left[ \left\| \boldsymbol{V}^\top \boldsymbol{\Sigma}_{\boldsymbol{\phi}}^{-\frac{1}{2}} \boldsymbol{\phi}(s') \right\|^2 \right]}$$

$$= \sqrt{\gamma} \sup_{\|\boldsymbol{W}\|_F=\|\boldsymbol{V}\|_F=1} \sqrt{ \mathbb{E}_s \left[ \boldsymbol{\phi}(s)^\top \boldsymbol{\Sigma}_{\boldsymbol{\phi}}^{-\frac{1}{2}} \boldsymbol{W}\boldsymbol{W}^\top \boldsymbol{\Sigma}_{\boldsymbol{\phi}}^{-\frac{1}{2}} \boldsymbol{\phi}(s) \right] \mathbb{E}_{s'} \left[ \boldsymbol{\phi}(s')^\top \boldsymbol{\Sigma}_{\boldsymbol{\phi}}^{-\frac{1}{2}} \boldsymbol{V}\boldsymbol{V}^\top \boldsymbol{\Sigma}_{\boldsymbol{\phi}}^{-\frac{1}{2}} \boldsymbol{\phi}(s') \right]}$$

$$= \sqrt{\gamma} \sup_{\|\boldsymbol{W}\|_F=\|\boldsymbol{V}\|_F=1} \sqrt{ \mathrm{tr} \left( \boldsymbol{W}\boldsymbol{W}^\top \boldsymbol{\Sigma}_{\boldsymbol{\phi}}^{-\frac{1}{2}} \mathbb{E}_s \left[ \boldsymbol{\phi}(s)\boldsymbol{\phi}(s)^\top \right] \boldsymbol{\Sigma}_{\boldsymbol{\phi}}^{-\frac{1}{2}} \right) \mathrm{tr} \left( \boldsymbol{V}\boldsymbol{V}^\top \boldsymbol{\Sigma}_{\boldsymbol{\phi}}^{-\frac{1}{2}} \mathbb{E}_{s'} \left[ \boldsymbol{\phi}(s')\boldsymbol{\phi}(s')^\top \right] \boldsymbol{\Sigma}_{\boldsymbol{\phi}}^{-\frac{1}{2}} \right)}$$

$$= \sqrt{\gamma} \sup_{\|\boldsymbol{W}\|_F=\|\boldsymbol{V}\|_F=1} \sqrt{ \mathrm{tr} \left( \boldsymbol{W}\boldsymbol{W}^\top \right) \mathrm{tr} \left( \boldsymbol{V}\boldsymbol{V}^\top \right) }$$

$$= \sqrt{\gamma} \sup_{\|\boldsymbol{W}\|_F=\|\boldsymbol{V}\|_F=1} \|\boldsymbol{W}\|_F \|\boldsymbol{V}\|_F$$

$$= \sqrt{\gamma},$$

where we used $\|\boldsymbol{Y}(r)\| \le \sqrt{\gamma}$ for any $r \in [0,1]$ by Lemma H.3, and Cauchy-Schwarz inequality.

To summarize, we have shown the desired result

$$\left( \boldsymbol{I}_K \otimes \boldsymbol{\Sigma}_{\boldsymbol{\phi}}^{-\frac{1}{2}} \right) \bar{\boldsymbol{A}}^\top \left( \boldsymbol{I}_K \otimes \boldsymbol{\Sigma}_{\boldsymbol{\phi}}^{-1} \right) \bar{\boldsymbol{A}} \left( \boldsymbol{I}_K \otimes \boldsymbol{\Sigma}_{\boldsymbol{\phi}}^{-\frac{1}{2}} \right) \succcurlyeq (1 - \sqrt{\gamma})^2 \, \boldsymbol{I}_{dK}.$$

$\square$

### E.2 Proof of Lemma 5.2

*Proof.* For simplicity, we omit $t$ in the random variables, for example, we use $\boldsymbol{A}$ to denote a random matrix with the same distribution as $\boldsymbol{A}_t$. In addition, we omit the subscripts in the expectation, the involving random variables are $s \sim \mu_\pi, a \sim \pi(\cdot \mid s), (r, s') \sim \mathcal{P}(\cdot, \cdot \mid s, a)$.

**Bounding $C_A$.** By Lemma A.3,

$$\|\boldsymbol{A}\| \le \left\| \boldsymbol{I}_K \otimes \left( \boldsymbol{\phi}(s)\boldsymbol{\phi}(s)^\top \right) \right\| + \left\| \left( \boldsymbol{C}\tilde{\boldsymbol{G}}(r)\boldsymbol{C}^{-1} \right) \otimes \left( \boldsymbol{\phi}(s)\boldsymbol{\phi}(s')^\top \right) \right\|$$

$$= \|\boldsymbol{\phi}(s)\|^2 + \|\boldsymbol{\phi}(s)\| \|\boldsymbol{\phi}(s')\| \left\| \boldsymbol{C}\tilde{\boldsymbol{G}}(r)\boldsymbol{C}^{-1} \right\|$$

$$\le 1 + \sqrt{\gamma},$$

where we used Lemma H.3. Hence, $C_A \le 2(1 + \sqrt{\gamma})$.

**Bounding $C_e$.** By Eqn. (32),

$$\|A\theta^\star\|^2 = (\theta^\star)^\top A^\top A\theta^\star$$
$$\leq 2\left(\|\theta^\star\|^2_{I_K \otimes (\phi(s)\phi(s)^\top)} + \gamma \|\theta^\star\|^2_{I_K \otimes (\phi(s')\phi(s')^\top)}\right)$$
$$\leq 2(1+\gamma)\sup_{s\in\mathcal{S}}\|\theta^\star\|^2_{I_K \otimes (\phi(s)\phi(s)^\top)}$$
$$\leq 2(1+\gamma)\sup_{s\in\mathcal{S}}\|\phi(s)\|^2\|\theta^\star\|^2$$
$$\leq 2(1+\gamma)\|\theta^\star\|^2.$$

Hence

$$\|A\theta^\star\| \leq \sqrt{2(1+\gamma)}\|\theta^\star\|.$$

As for $\|b\|$,

$$\|b\| = \frac{1}{K+1}\left\|\left[C\left(\sum_{j=0}^K g_j(r) - \mathbf{1}_K\right)\right]\otimes\phi(s)\right\|$$
$$\leq \frac{1}{K+1}\left\|C\left(\sum_{j=0}^K g_j(r) - \mathbf{1}_K\right)\right\|\|\phi(s)\| \tag{30}$$
$$\leq \frac{1}{K+1}\left\|C\left(\sum_{j=0}^K g_j(r) - \mathbf{1}_K\right)\right\|.$$

By Proposition C.3 with $\eta \in \mathscr{P}_K^{\text{sign}}$ satisfying $\eta(\tilde{s}) = \nu$ for all $\tilde{s} \in \mathcal{S}$, where $\nu = (K+1)^{-1}\sum_{k=0}^K \delta_{x_k}$ is the discrete uniform distribution, we can derive that, for any $r \in [0,1]$ and $s' \in \mathcal{S}$, it holds that

$$\frac{1}{K+1}\left(\sum_{j=0}^K g_j(r) - \mathbf{1}_K\right) = \left(p_{(b_{r,\gamma})\#\eta(s')} - \frac{1}{K+1}\mathbf{1}_K\right) - \tilde{G}(r)\left(p_\eta(s') - \frac{1}{K+1}\mathbf{1}_K\right)$$

$$= p_{(b_{r,\gamma})\#\nu} - \frac{1}{K+1}\mathbf{1}_K,$$

Hence,

$$\|b\| \leq \frac{1}{K+1}\left\|C\left(\sum_{j=0}^K g_j(r) - \mathbf{1}_K\right)\right\|$$
$$= \left\|C\left(p_{(b_{r,\gamma})\#\nu} - \frac{1}{K+1}\mathbf{1}_K\right)\right\|$$
$$= \frac{1}{\sqrt{\iota_K}}\ell_2\left(\Pi_K(b_{r,\gamma})_\#(\nu), \nu\right)$$
$$\leq \sqrt{K(1-\gamma)}\ell_2\left((b_{r,\gamma})_\#(\nu), \nu\right)$$
$$\leq 3\sqrt{K}(1-\gamma),$$

where we used the orthogonal decomposition (Proposition C.2) and an upper bound for $\ell_2\left((b_{r,\gamma})_\#(\nu), \nu\right)$ (Lemma H.4).

In summary,

$$\|e\| = \|A\theta^\star - b\|$$
$$\leq \|A\theta^\star\| + \|b\|$$
$$\leq \sqrt{2(1+\gamma)}\|\theta^\star\| + 3\sqrt{K}(1-\gamma).$$

Hence, $C_e \leq \sqrt{2(1+\gamma)}\|\theta^\star\| + 3\sqrt{K}(1-\gamma)$.

**Bounding** $\operatorname{tr}(\boldsymbol{\Sigma}_e)$**.**

$$\operatorname{tr}(\boldsymbol{\Sigma}_e) = \mathbb{E}\left[\|\boldsymbol{A}\boldsymbol{\theta}^\star - \boldsymbol{b}\|^2\right] \leq 2(\boldsymbol{\theta}^\star)^\top \mathbb{E}\left[\boldsymbol{A}^\top \boldsymbol{A}\right]\boldsymbol{\theta}^\star + 2\mathbb{E}\left[\boldsymbol{b}^\top \boldsymbol{b}\right].$$

By Eqn. (33),

$$(\boldsymbol{\theta}^\star)^\top \mathbb{E}\left[\boldsymbol{A}^\top \boldsymbol{A}\right]\boldsymbol{\theta}^\star \leq 2(1+\gamma)\|\boldsymbol{\theta}^\star\|_{\boldsymbol{I}_K \otimes \boldsymbol{\Sigma}_\phi}^2.$$

And by Lemma H.4,

$$\mathbb{E}\left[\boldsymbol{b}^\top \boldsymbol{b}\right] \leq 9K(1-\gamma)^2.$$

To summarize,

$$\operatorname{tr}(\boldsymbol{\Sigma}_e) \leq 4(1+\gamma)\|\boldsymbol{\theta}^\star\|_{\boldsymbol{I}_K \otimes \boldsymbol{\Sigma}_\phi}^2 + 18K(1-\gamma)^2.$$

$\square$

### E.3  Proof of Lemma 5.3

*Proof.* For simplicity, we use the same abbreviations as in Appendix E.2. As in the proof of [Lemma 2 Samsonov et al., 2024b], we only need to show that, for any $p \in \mathbb{N}$, $\alpha \in \left(0, (1-\sqrt{\gamma})/(38p)\right)$, it holds that

$$\mathbb{E}\left\{\left[(\boldsymbol{I}_{dK} - \alpha\boldsymbol{A})^\top (\boldsymbol{I}_{dK} - \alpha\boldsymbol{A})\right]^p\right\} \preccurlyeq \boldsymbol{I}_{dK} - \frac{1}{2}\alpha p(1-\gamma)\boldsymbol{I}_K \otimes \boldsymbol{\Sigma}_\phi \left(\preccurlyeq \left(1 - \frac{1}{2}\alpha p(1-\gamma)\lambda_{\min}\right)\boldsymbol{I}_{dK}\right).$$

Let $\boldsymbol{B} := \boldsymbol{A} + \boldsymbol{A}^\top - \alpha\boldsymbol{A}^\top \boldsymbol{A}$ which satisfies $(\boldsymbol{I}_{dK} - \alpha\boldsymbol{A})^\top (\boldsymbol{I}_{dK} - \alpha\boldsymbol{A}) = \boldsymbol{I}_{dK} - \alpha\boldsymbol{B}$. To give an upper bound of $\mathbb{E}\left[(\boldsymbol{I}_{dK} - \alpha\boldsymbol{B})^p\right]$, it suffices to show that

$$\mathbb{E}\left[\boldsymbol{B}\right] \succcurlyeq (1-\sqrt{\gamma})\boldsymbol{I}_K \otimes \boldsymbol{\Sigma}_\phi, \quad \mathbb{E}\left[\boldsymbol{B}^p\right] \preccurlyeq \frac{17}{16}4^p \boldsymbol{I}_K \otimes \boldsymbol{\Sigma}_\phi, \quad \forall p \in \{2, 3, \cdots\},$$

if we take $\alpha \in \left(0, (1-\sqrt{\gamma})/(2(1+\gamma))\right)$.

Given these results, we have, when $\alpha \in \left(0, (1-\sqrt{\gamma})/(38p)\right)$, it holds that

$$\begin{aligned}
\mathbb{E}\left[(\boldsymbol{I}_{dK} - \alpha\boldsymbol{B})^p\right] &\preccurlyeq \boldsymbol{I} - \alpha p\mathbb{E}\left[\boldsymbol{B}\right] + \sum_{l=2}^p \alpha^l \binom{p}{l}\mathbb{E}\left[\boldsymbol{B}^l\right] \\
&\preccurlyeq \boldsymbol{I} - \left(\alpha p(1-\sqrt{\gamma}) - \frac{17}{16}\sum_{l=2}^\infty (4\alpha p)^l\right)\boldsymbol{I}_K \otimes \boldsymbol{\Sigma}_\phi \\
&= \boldsymbol{I} - \left(\alpha p(1-\sqrt{\gamma}) - \frac{17\alpha^2 p^2}{1-4\alpha p}\right)\boldsymbol{I}_K \otimes \boldsymbol{\Sigma}_\phi \\
&\preccurlyeq \boldsymbol{I} - \frac{1}{2}\alpha p(1-\sqrt{\gamma})\boldsymbol{I}_K \otimes \boldsymbol{\Sigma}_\phi.
\end{aligned}$$

**Lower Bound of** $\mathbb{E}\left[\boldsymbol{B}\right]$**.** To show $\mathbb{E}\left[\boldsymbol{B}\right] \succcurlyeq (1-\sqrt{\gamma})\boldsymbol{I}_K \otimes \boldsymbol{\Sigma}_\phi$, we first show that $\mathbb{E}\left[\boldsymbol{A} + \boldsymbol{A}^\top\right] \succcurlyeq 2(1-\sqrt{\gamma})\boldsymbol{I}_K \otimes \boldsymbol{\Sigma}_\phi$, which is equivalent to $(\boldsymbol{I}_K \otimes \boldsymbol{\Sigma}_\phi)^{-\frac{1}{2}}\mathbb{E}\left[\boldsymbol{A} + \boldsymbol{A}^\top\right](\boldsymbol{I}_K \otimes \boldsymbol{\Sigma}_\phi)^{-\frac{1}{2}} \succcurlyeq 2(1-\sqrt{\gamma})$, where

$$\mathbb{E}\left[\boldsymbol{A} + \boldsymbol{A}^\top\right] = 2(\boldsymbol{I}_K \otimes \boldsymbol{\Sigma}_\phi) - \mathbb{E}\left[\left(\boldsymbol{C}\tilde{\boldsymbol{G}}(r)\boldsymbol{C}^{-1}\right) \otimes \left(\boldsymbol{\phi}(s)\boldsymbol{\phi}(s')^\top\right)\right] - \mathbb{E}\left[\left(\boldsymbol{C}\tilde{\boldsymbol{G}}(r)\boldsymbol{C}^{-1}\right) \otimes \left(\boldsymbol{\phi}(s)\boldsymbol{\phi}(s')^\top\right)\right]^\top.$$

Then, for any $\boldsymbol{\theta} \in \mathbb{R}^{dK}$ with $\|\boldsymbol{\theta}\| = 1$,

$$\begin{aligned}
\boldsymbol{\theta}^\top (\boldsymbol{I}_K \otimes \boldsymbol{\Sigma}_\phi)^{-\frac{1}{2}}&\mathbb{E}\left[\boldsymbol{A} + \boldsymbol{A}^\top\right](\boldsymbol{I}_K \otimes \boldsymbol{\Sigma}_\phi)^{-\frac{1}{2}}\boldsymbol{\theta} \\
&= 2 - 2\boldsymbol{\theta}^\top (\boldsymbol{I}_K \otimes \boldsymbol{\Sigma}_\phi)^{-\frac{1}{2}}\mathbb{E}\left[\left(\boldsymbol{C}\tilde{\boldsymbol{G}}(r)\boldsymbol{C}^{-1}\right) \otimes \left(\boldsymbol{\phi}(s)\boldsymbol{\phi}(s')^\top\right)\right](\boldsymbol{I}_K \otimes \boldsymbol{\Sigma}_\phi)^{-\frac{1}{2}}\boldsymbol{\theta} \\
&\geq 2 - 2\left\|\mathbb{E}_{s,r,s'}\left[\left(\boldsymbol{C}\tilde{\boldsymbol{G}}(r)\boldsymbol{C}^{-1}\right) \otimes \left(\boldsymbol{\Sigma}_\phi^{-\frac{1}{2}}\boldsymbol{\phi}(s)\boldsymbol{\phi}(s')^\top \boldsymbol{\Sigma}_\phi^{-\frac{1}{2}}\right)\right]\right\| \\
&\geq 2(1-\sqrt{\gamma}),
\end{aligned}$$

where we used the result Eqn. (29).

Next, we give an upper bound for $\mathbb{E}\left[\boldsymbol{A}^\top \boldsymbol{A}\right]$, we need to compute the following terms: by Lemma A.4,

$$
\begin{aligned}
\left[\boldsymbol{I}_K \otimes \left(\boldsymbol{\phi}(s)\boldsymbol{\phi}(s)^\top\right)\right]^2 &= \boldsymbol{I}_K \otimes \left(\boldsymbol{\phi}(s)\boldsymbol{\phi}(s)^\top \boldsymbol{\phi}(s)\boldsymbol{\phi}(s)^\top\right) \\
&= \|\boldsymbol{\phi}(s)\|^2 \, \boldsymbol{I}_K \otimes \left(\boldsymbol{\phi}(s)\boldsymbol{\phi}(s)^\top\right) \\
&\preccurlyeq \boldsymbol{I}_K \otimes \left(\boldsymbol{\phi}(s)\boldsymbol{\phi}(s)^\top\right).
\end{aligned}
\tag{31}
$$

Hence

$$
\mathbb{E}\left[\boldsymbol{I}_K \otimes \left(\boldsymbol{\phi}(s)\boldsymbol{\phi}(s)^\top\right)\right]^2 \preccurlyeq \boldsymbol{I}_K \otimes \boldsymbol{\Sigma}_{\boldsymbol{\phi}}.
$$

And by Lemma A.4 and Lemma H.3,

$$
\begin{aligned}
&\left[\left(\boldsymbol{C}\tilde{\boldsymbol{G}}(r)\boldsymbol{C}^{-1}\right) \otimes \left(\boldsymbol{\phi}(s)\boldsymbol{\phi}(s')^\top\right)\right]^\top \left[\left(\boldsymbol{C}\tilde{\boldsymbol{G}}(r)\boldsymbol{C}^{-1}\right) \otimes \left(\boldsymbol{\phi}(s)\boldsymbol{\phi}(s')^\top\right)\right] \\
&= \left(\boldsymbol{C}^{-\top}\tilde{\boldsymbol{G}}^\top(r)\boldsymbol{C}^\top \boldsymbol{C}\tilde{\boldsymbol{G}}(r)\boldsymbol{C}^{-1}\right) \otimes \left(\boldsymbol{\phi}(s')\boldsymbol{\phi}(s)^\top \boldsymbol{\phi}(s)\boldsymbol{\phi}(s')^\top\right) \\
&= \|\boldsymbol{\phi}(s)\|^2 \left(\boldsymbol{C}^{-\top}\tilde{\boldsymbol{G}}^\top(r)\boldsymbol{C}^\top \boldsymbol{C}\tilde{\boldsymbol{G}}(r)\boldsymbol{C}^{-1}\right) \otimes \left(\boldsymbol{\phi}(s')\boldsymbol{\phi}(s')^\top\right) \\
&\preccurlyeq \left\|\boldsymbol{C}\tilde{\boldsymbol{G}}(r)\boldsymbol{C}^{-1}\right\|^2 \boldsymbol{I}_K \otimes \left(\boldsymbol{\phi}(s')\boldsymbol{\phi}(s')^\top\right) \\
&\preccurlyeq \gamma \boldsymbol{I}_K \otimes \left(\boldsymbol{\phi}(s')\boldsymbol{\phi}(s')^\top\right).
\end{aligned}
$$

To summarize, by the basic inequality $(\boldsymbol{B}_1 - \boldsymbol{B}_2)^\top (\boldsymbol{B}_1 - \boldsymbol{B}_2) \preccurlyeq 2\left(\boldsymbol{B}_1^\top \boldsymbol{B}_1 + \boldsymbol{B}_2^\top \boldsymbol{B}_2\right)$, we have

$$
\boldsymbol{A}^\top \boldsymbol{A} \preccurlyeq 2\boldsymbol{I}_K \otimes \left(\boldsymbol{\phi}(s)\boldsymbol{\phi}(s)^\top + \gamma \boldsymbol{\phi}(s')\boldsymbol{\phi}(s')^\top\right),
\tag{32}
$$

and, after taking expectation,

$$
\mathbb{E}\left[\boldsymbol{A}^\top \boldsymbol{A}\right] \preccurlyeq 2\left(1+\gamma\right)\boldsymbol{I}_K \otimes \boldsymbol{\Sigma}_{\boldsymbol{\phi}}.
\tag{33}
$$

Combining these together, we obtain

$$
\mathbb{E}\left[\boldsymbol{B}\right] \succcurlyeq 2\left[(1-\sqrt{\gamma}) - \alpha(1+\gamma)\right]\boldsymbol{I}_K \otimes \boldsymbol{\Sigma}_{\boldsymbol{\phi}} \succcurlyeq (1-\sqrt{\gamma})\boldsymbol{I}_K \otimes \boldsymbol{\Sigma}_{\boldsymbol{\phi}},
$$

if we take $\alpha \in \left(0, (1-\sqrt{\gamma})/(2(1+\gamma))\right)$.

**Upper Bound of $\mathbb{E}\left[\boldsymbol{B}^p\right]$.** Because $\boldsymbol{B}^2$ is always PSD, we have the following upper bound

$$
\boldsymbol{B}^p \preccurlyeq \|\boldsymbol{B}\|^{p-2} \, \boldsymbol{B}^2.
$$

We first give an almost-sure upper bound for $\|\boldsymbol{B}\|$. By Lemma 5.2, $\|\boldsymbol{A}\| \leq 1 + \sqrt{\gamma}$. And by Eqn. (32),

$$
\begin{aligned}
\left\|\boldsymbol{A}^\top \boldsymbol{A}\right\| &\leq 2\left\|\boldsymbol{I}_K \otimes \left(\boldsymbol{\phi}(s)\boldsymbol{\phi}(s)^\top + \gamma \boldsymbol{\phi}(s')\boldsymbol{\phi}(s')^\top\right)\right\| \\
&\leq 2\left\|\boldsymbol{\phi}(s)\boldsymbol{\phi}(s)^\top + \gamma \boldsymbol{\phi}(s')\boldsymbol{\phi}(s')^\top\right\| \\
&\leq 2(1+\gamma).
\end{aligned}
\tag{34}
$$

Hence,

$$
\begin{aligned}
\|\boldsymbol{B}\| &= \left\|\boldsymbol{A} + \boldsymbol{A}^\top - \alpha \boldsymbol{A}^\top \boldsymbol{A}\right\| \\
&\leq 2\|\boldsymbol{A}\| + \alpha \left\|\boldsymbol{A}^\top \boldsymbol{A}\right\| \\
&\leq 2(1+\sqrt{\gamma}) + 2\alpha(1+\gamma) \\
&\leq 4,
\end{aligned}
\tag{35}
$$

because $\alpha \in \left(0, (1-\sqrt{\gamma})/(2(1+\gamma))\right)$.

Now, we aim to give an upper bound for $\mathbb{E}\left[\boldsymbol{B}^2\right]$,

$$
\begin{aligned}
\boldsymbol{B}^2 &= \left(\boldsymbol{A} + \boldsymbol{A}^\top - \alpha \boldsymbol{A}^\top \boldsymbol{A}\right)^2 \\
&\preccurlyeq (1+\beta)\left(\boldsymbol{A} + \boldsymbol{A}^\top\right)^2 + \left(1+\beta^{-1}\right)\alpha^2 \left(\boldsymbol{A}^\top \boldsymbol{A}\right)^2 \\
&\preccurlyeq 2(1+\beta)\left(\boldsymbol{A}^\top \boldsymbol{A} + \boldsymbol{A}\boldsymbol{A}^\top\right) + (1+\beta^{-1})\alpha^2 \left(\boldsymbol{A}^\top \boldsymbol{A}\right)^2,
\end{aligned}
$$

where we used the fact that $(\boldsymbol{B}_1 + \boldsymbol{B}_2)^2 \preccurlyeq (1+\beta)\boldsymbol{B}_1^2 + (1+\beta^{-1})\boldsymbol{B}_2^2$ for any symmetric matrices $\boldsymbol{B}_1, \boldsymbol{B}_2$, since $\beta\boldsymbol{B}_1^2 + \beta^{-1}\boldsymbol{B}_2^2 - \boldsymbol{B}_1\boldsymbol{B}_2 - \boldsymbol{B}_2\boldsymbol{B}_1 = \left(\sqrt{\beta}\boldsymbol{B}_1 - \sqrt{\beta^{-1}}\boldsymbol{B}_2\right)^2 \succcurlyeq \boldsymbol{0}$, $\beta \in (0,1)$ to be determined; and the fact that $\boldsymbol{A}^2 + \left(\boldsymbol{A}^\top\right)^2 \preccurlyeq \boldsymbol{A}^\top\boldsymbol{A} + \boldsymbol{A}\boldsymbol{A}^\top$ since the square of the skew-symmetric matrix is negative semi-definite $\left(\boldsymbol{A} - \boldsymbol{A}^\top\right)^2 \preccurlyeq \boldsymbol{0}$. By Eqn. (34) and Eqn. (33), we have

$$\left\|\boldsymbol{A}^\top\boldsymbol{A}\right\| \leq 2(1+\gamma),$$

$$\mathbb{E}\left[\boldsymbol{A}^\top\boldsymbol{A}\right] \preccurlyeq 2(1+\gamma)\boldsymbol{I}_K \otimes \boldsymbol{\Sigma}_\phi, \tag{36}$$

thus, by $\alpha \in \left(0, (1-\sqrt{\gamma})/(2(1+\gamma))\right)$, it holds that

$$\alpha^2 \mathbb{E}\left[\left(\boldsymbol{A}^\top\boldsymbol{A}\right)^2\right] \preccurlyeq 4\alpha^2(1+\gamma)^2 \boldsymbol{I}_K \otimes \boldsymbol{\Sigma}_\phi \preccurlyeq (1-\sqrt{\gamma})^2 \boldsymbol{I}_K \otimes \boldsymbol{\Sigma}_\phi. \tag{37}$$

As for $\mathbb{E}\left[\boldsymbol{A}\boldsymbol{A}^\top\right]$, by the basic inequality $(\boldsymbol{B}_1 - \boldsymbol{B}_2)(\boldsymbol{B}_1 - \boldsymbol{B}_2)^\top \preccurlyeq 2\left(\boldsymbol{B}_1\boldsymbol{B}_1^\top + \boldsymbol{B}_2\boldsymbol{B}_2^\top\right)$, we have

$$\boldsymbol{A}\boldsymbol{A}^\top = \left\{\left[\boldsymbol{I}_K \otimes \left(\boldsymbol{\phi}(s)\boldsymbol{\phi}(s)^\top\right)\right] - \left[\left(\boldsymbol{C}\tilde{\boldsymbol{G}}(r)\boldsymbol{C}^{-1}\right) \otimes \left(\boldsymbol{\phi}(s)\boldsymbol{\phi}(s')^\top\right)\right]\right\}$$
$$\cdot \left\{\left[\boldsymbol{I}_K \otimes \left(\boldsymbol{\phi}(s)\boldsymbol{\phi}(s)^\top\right)\right] - \left[\left(\boldsymbol{C}\tilde{\boldsymbol{G}}(r)\boldsymbol{C}^{-1}\right) \otimes \left(\boldsymbol{\phi}(s)\boldsymbol{\phi}(s')^\top\right)\right]\right\}^\top$$
$$\preccurlyeq 2\left[\boldsymbol{I}_K \otimes \left(\boldsymbol{\phi}(s)\boldsymbol{\phi}(s)^\top\right)\right]^2 + 2\left[\left(\boldsymbol{C}\tilde{\boldsymbol{G}}(r)\boldsymbol{C}^{-1}\right) \otimes \left(\boldsymbol{\phi}(s)\boldsymbol{\phi}(s')^\top\right)\right]\left[\left(\boldsymbol{C}\tilde{\boldsymbol{G}}(r)\boldsymbol{C}^{-1}\right) \otimes \left(\boldsymbol{\phi}(s)\boldsymbol{\phi}(s')^\top\right)\right]^\top.$$

By Eqn. (31), we have

$$\left[\boldsymbol{I}_K \otimes \left(\boldsymbol{\phi}(s)\boldsymbol{\phi}(s)^\top\right)\right]^2 \preccurlyeq \boldsymbol{I}_K \otimes \left(\boldsymbol{\phi}(s)\boldsymbol{\phi}(s)^\top\right).$$

And by Lemma H.3,

$$\left[\left(\boldsymbol{C}\tilde{\boldsymbol{G}}(r)\boldsymbol{C}^{-1}\right) \otimes \left(\boldsymbol{\phi}(s)\boldsymbol{\phi}(s')^\top\right)\right]\left[\left(\boldsymbol{C}\tilde{\boldsymbol{G}}(r)\boldsymbol{C}^{-1}\right) \otimes \left(\boldsymbol{\phi}(s)\boldsymbol{\phi}(s')^\top\right)\right]^\top$$
$$= \left(\boldsymbol{C}\tilde{\boldsymbol{G}}(r)\boldsymbol{C}^{-1}\boldsymbol{C}^{-T}\tilde{\boldsymbol{G}}^\top(r)\boldsymbol{C}^\top\right) \otimes \left(\boldsymbol{\phi}(s)\boldsymbol{\phi}(s')^\top\boldsymbol{\phi}(s')\boldsymbol{\phi}(s)^\top\right)$$
$$= \|\boldsymbol{\phi}(s')\|^2 \left(\boldsymbol{C}\tilde{\boldsymbol{G}}(r)\boldsymbol{C}^{-1}\boldsymbol{C}^{-T}\tilde{\boldsymbol{G}}^\top(r)\boldsymbol{C}^\top\right) \otimes \left(\boldsymbol{\phi}(s)\boldsymbol{\phi}(s)^\top\right)$$
$$\preccurlyeq \left\|\boldsymbol{C}\tilde{\boldsymbol{G}}(r)\boldsymbol{C}^{-1}\right\|^2 \boldsymbol{I}_K \otimes \left(\boldsymbol{\phi}(s)\boldsymbol{\phi}(s)^\top\right)$$
$$\preccurlyeq \gamma \boldsymbol{I}_K \otimes \left(\boldsymbol{\phi}(s)\boldsymbol{\phi}(s)^\top\right).$$

To summarize, we have

$$\boldsymbol{A}\boldsymbol{A}^\top \preccurlyeq 2(1+\gamma)\boldsymbol{I}_K \otimes \left(\boldsymbol{\phi}(s)\boldsymbol{\phi}(s)^\top\right),$$

and after taking expectation

$$\mathbb{E}\left[\boldsymbol{A}\boldsymbol{A}^\top\right] \preccurlyeq 2(1+\gamma)\boldsymbol{I}_K \otimes \boldsymbol{\Sigma}_\phi. \tag{38}$$

By putting everything together (Eqn. (36), Eqn. (37), Eqn. (38)), we have

$$\boldsymbol{B}^2 \preccurlyeq 2(1+\beta)\left(\boldsymbol{A}^\top\boldsymbol{A} + \boldsymbol{A}\boldsymbol{A}^\top\right) + (1+\beta^{-1})\alpha^2 \left(\boldsymbol{A}^\top\boldsymbol{A}\right)^2$$
$$\preccurlyeq \left[8(1+\gamma)(1+\beta) + (1+\beta^{-1})(1-\sqrt{\gamma})^2\right]\boldsymbol{I}_K \otimes \boldsymbol{\Sigma}_\phi$$
$$\preccurlyeq (17 + 9\gamma - 10\sqrt{\gamma})\boldsymbol{I}_K \otimes \boldsymbol{\Sigma}_\phi$$
$$\preccurlyeq 17\boldsymbol{I}_K \otimes \boldsymbol{\Sigma}_\phi,$$

where we take $\beta = \frac{1-\sqrt{\gamma}}{\sqrt{8(1+\gamma)}}$. Therefore, by Eqn. (35),

$$\boldsymbol{B}^p \preccurlyeq \|\boldsymbol{B}\|^{p-2}\boldsymbol{B}^2$$
$$\preccurlyeq 4^{p-2} 17 \boldsymbol{I}_K \otimes \boldsymbol{\Sigma}_\phi$$
$$= \frac{17}{16} 4^p \boldsymbol{I}_K \otimes \boldsymbol{\Sigma}_\phi.$$

$\square$

## E.4 $L^p$ Convergence

**Theorem E.1** ($L^p$ Convergence). *For any $K \geq (1-\gamma)^{-1}$, $p > 2$ and $\alpha \in (0, (1-\sqrt{\gamma})/[38(p + \log T)])$, it holds that*

$$
\mathbb{E}^{1/p}\left[\left(\mathcal{L}\left(\bar{\boldsymbol{\theta}}_T\right)\right)^p\right] \lesssim \frac{\sqrt{p}}{\sqrt{T}} \frac{\frac{1}{\sqrt{K}(1-\gamma)}\|\boldsymbol{\theta}^\star\|_{\boldsymbol{I}_K \otimes \boldsymbol{\Sigma}_\phi} + 1}{(1-\gamma)\sqrt{\lambda_{\min}}}\left(1 + \frac{\sqrt{\alpha p} + \alpha p}{\sqrt{(1-\gamma)\lambda_{\min}}}\right)
$$
$$
+ \frac{p}{T} \frac{\frac{1}{\sqrt{K}(1-\gamma)}\|\boldsymbol{\theta}^\star\|_{\boldsymbol{I}_K \otimes \boldsymbol{\Sigma}_\phi} + 1}{(1-\gamma)^{\frac{3}{2}}\lambda_{\min}}\left(1 + \frac{1}{\sqrt{\alpha p}}\right)
$$
$$
+ \frac{1}{T}\frac{(1-\frac{1}{2}\alpha(1-\sqrt{\gamma})\lambda_{\min})^{T/2}}{\sqrt{\alpha}(1-\gamma)\sqrt{\lambda_{\min}}}\left(\frac{1}{\sqrt{\alpha}} + \frac{p}{\sqrt{(1-\gamma)\lambda_{\min}}}\right)\frac{1}{\sqrt{K}(1-\gamma)}\|\boldsymbol{\theta}_0 - \boldsymbol{\theta}^\star\|.
$$

*Proof.* Combining Lemma 5.1, Lemma 5.2 and Lemma 5.3 with [Theorem 2 Samsonov et al., 2024b], we have

$$
\mathbb{E}^{1/p}\left[\left(\mathcal{L}\left(\bar{\boldsymbol{\theta}}_T\right)\right)^p\right]
$$
$$
\lesssim \frac{1}{\sqrt{K(1-\gamma)^4\lambda_{\min}}}\left[\sqrt{\frac{p\,\mathrm{tr}\left(\boldsymbol{\Sigma}_e\right)}{T}}\left(1 + \frac{C_A\sqrt{\alpha p}}{\sqrt{a}} + \frac{C_A C_e \alpha p}{\sqrt{\mathrm{tr}\left(\boldsymbol{\Sigma}_e\right)}}\right) + \frac{(1+C_A)C_e p}{T}\right.
$$
$$
\left. + \frac{p\sqrt{\mathrm{tr}\left(\boldsymbol{\Sigma}_e\right)}}{\sqrt{a}T}\left(1 + \frac{1}{\sqrt{\alpha p}}\right) + (1-\alpha a)^{T/2}\left(\frac{1}{\alpha T} + \frac{C_A p}{\sqrt{\alpha a}T}\right)\|\boldsymbol{\theta}_0 - \boldsymbol{\theta}^\star\|\right]
$$
$$
\lesssim \frac{1}{\sqrt{K(1-\gamma)^4\lambda_{\min}}}\left[\sqrt{p}\frac{\|\boldsymbol{\theta}^\star\|_{\boldsymbol{I}_K \otimes \boldsymbol{\Sigma}_\phi} + \sqrt{K}(1-\gamma)}{\sqrt{T}}\left(1 + \frac{\sqrt{\alpha p} + \alpha p}{\sqrt{(1-\gamma)\lambda_{\min}}}\right) + \frac{p\left(\|\boldsymbol{\theta}^\star\| + \sqrt{K}(1-\gamma)\right)}{T}\right.
$$
$$
+ p\frac{\|\boldsymbol{\theta}^\star\|_{\boldsymbol{I}_K \otimes \boldsymbol{\Sigma}_\phi} + \sqrt{K}(1-\gamma)}{\sqrt{(1-\gamma)\lambda_{\min}}T}\left(1 + \frac{1}{\sqrt{\alpha p}}\right)
$$
$$
\left. + (1-\frac{1}{2}\alpha(1-\sqrt{\gamma})\lambda_{\min})^{T/2}\left(\frac{1}{\alpha T} + \frac{p}{\sqrt{\alpha(1-\gamma)\lambda_{\min}}T}\right)\|\boldsymbol{\theta}_0 - \boldsymbol{\theta}^\star\|\right]
$$
$$
\lesssim \frac{\sqrt{p}}{\sqrt{T}} \frac{\frac{1}{\sqrt{K}(1-\gamma)}\|\boldsymbol{\theta}^\star\|_{\boldsymbol{I}_K \otimes \boldsymbol{\Sigma}_\phi} + 1}{(1-\gamma)\sqrt{\lambda_{\min}}}\left(1 + \frac{\sqrt{\alpha p} + \alpha p}{\sqrt{(1-\gamma)\lambda_{\min}}}\right)
$$
$$
+ \frac{p}{T} \frac{\frac{1}{\sqrt{K}(1-\gamma)}\|\boldsymbol{\theta}^\star\|_{\boldsymbol{I}_K \otimes \boldsymbol{\Sigma}_\phi} + 1}{(1-\gamma)^{\frac{3}{2}}\lambda_{\min}}\left(1 + \frac{1}{\sqrt{\alpha p}}\right)
$$
$$
+ \frac{1}{T}\frac{(1-\frac{1}{2}\alpha(1-\sqrt{\gamma})\lambda_{\min})^{T/2}}{\sqrt{\alpha}(1-\gamma)\sqrt{\lambda_{\min}}}\left(\frac{1}{\sqrt{\alpha}} + \frac{p}{\sqrt{(1-\gamma)\lambda_{\min}}}\right)\frac{1}{\sqrt{K}(1-\gamma)}\|\boldsymbol{\theta}_0 - \boldsymbol{\theta}^\star\|,
$$

where we used the fact that $\|\boldsymbol{\theta}^\star\| \leq (\lambda_{\min})^{-1/2}\|\boldsymbol{\theta}^\star\|_{\boldsymbol{I}_K \otimes \boldsymbol{\Sigma}_\phi}$. $\qquad\square$

## E.5 Convergence Results for SSGD with the PMF Representation

In this section, we present the counterparts of Lemma 5.1, Lemma 5.2, Lemma 5.3 and Theorem 4.1 for stochastic semi-gradient descent (SSGD) with the probability mass function (PMF) representation. These results will additionally depend on $K$. The additional $K$-dependent terms arise because the condition number of $\boldsymbol{C}^\top\boldsymbol{C}$ scales with $K^2$ (Lemma H.2). These terms are inevitable within our theoretical framework. The proofs of these results require only minor modifications to the original proofs, and we omit them for brevity.

In fact, in Appendix G, we validate some theoretical results through numerical experiments. To be concrete, we find that empirically, as $K$ increases, to ensure convergence, the step size of the vanilla algorithm in [Bellemare et al., 2023, Section 9.6] indeed needs to decay at a rate of $K^{-2}$. In contrast, the step size of our `Linear-CTD` does not need to be adjusted when $K$ increases. Moreover, we find that `Linear-CTD` empirically consistently outperforms the vanilla algorithm under different $K$.

Recall Eqn. (26), the updating scheme of the algorithm is

$$\boldsymbol{W}_t \leftarrow \boldsymbol{W}_{t-1} - \alpha\boldsymbol{\phi}(s_t)\left(\boldsymbol{p}_{\boldsymbol{w}_{t-1}}(s_t) - \boldsymbol{p}_{\mathcal{T}_t^\pi \boldsymbol{\eta}_{\boldsymbol{w}_{t-1}}}(s_t)\right)^\top \boldsymbol{C}^\top \boldsymbol{C}$$

$$= \boldsymbol{W}_{t-1} - \alpha\boldsymbol{\phi}(s_t)\left[\boldsymbol{\phi}(s_t)^\top \boldsymbol{W}_{t-1} - \boldsymbol{\phi}(s_{t+1})^\top \boldsymbol{W}_{t-1}\tilde{\boldsymbol{G}}^\top(r_t) - \frac{1}{K+1}\left(\sum_{j=0}^{K}\boldsymbol{g}_j(r_t) - \mathbf{1}_K\right)^\top\right]\boldsymbol{C}^\top \boldsymbol{C},$$

which is equivalent to

$$\boldsymbol{W}_t\boldsymbol{C}^\top \leftarrow \boldsymbol{W}_{t-1}\boldsymbol{C}^\top - \alpha\boldsymbol{\phi}(s_t)\left[\boldsymbol{\phi}(s_t)^\top \boldsymbol{W}_{t-1}\boldsymbol{C}^\top - \boldsymbol{\phi}(s_{t+1})^\top \boldsymbol{W}_{t-1}\boldsymbol{C}^\top(\boldsymbol{C}\tilde{\boldsymbol{G}}(r_t)\boldsymbol{C}^{-1})^\top - \frac{1}{K+1}\left(\sum_{j=0}^{K}\boldsymbol{g}_j(r_t) - \mathbf{1}_K\right)^\top\boldsymbol{C}^\top\right]\boldsymbol{C}\boldsymbol{C}^\top,$$

here we drop the additional $2\iota_K$ in the step size for brevity. Letting $\boldsymbol{\Theta}_{\mathrm{PMF},t} := \boldsymbol{W}_t\boldsymbol{C}^\top$ be the CDF parameter, the algorithm becomes

$$\boldsymbol{\Theta}_{\mathrm{PMF},t} \leftarrow \boldsymbol{\Theta}_{\mathrm{PMF},t-1} - \alpha\boldsymbol{\phi}(s_t)\left[\boldsymbol{\phi}(s_t)^\top \boldsymbol{\Theta}_{\mathrm{PMF},t-1} - \boldsymbol{\phi}(s_{t+1})^\top \boldsymbol{\Theta}_{\mathrm{PMF},t-1}(\boldsymbol{C}\tilde{\boldsymbol{G}}(r_t)\boldsymbol{C}^{-1})^\top - \frac{1}{K+1}\left(\sum_{j=0}^{K}\boldsymbol{g}_j(r_t) - \mathbf{1}_K\right)^\top\boldsymbol{C}^\top\right]\boldsymbol{C}\boldsymbol{C}^\top.$$

$$(39)$$

Here, we add the subscript PMF to the original notations to indicate the difference. Then, the algorithm corresponds to the following linear system for $\boldsymbol{\theta} \in \mathbb{R}^{dK}$

$$\bar{\boldsymbol{A}}_{\mathrm{PMF}}\boldsymbol{\theta} = \bar{\boldsymbol{b}}_{\mathrm{PMF}},$$

where

$$\bar{\boldsymbol{A}}_{\mathrm{PMF}} = \left[(\boldsymbol{C}\boldsymbol{C}^\top) \otimes \boldsymbol{\Sigma}_\phi\right] - \mathbb{E}_{s,r,s'}\left[\left(\boldsymbol{C}\boldsymbol{C}^\top(\boldsymbol{C}\tilde{\boldsymbol{G}}(r_t)\boldsymbol{C}^{-1})\right) \otimes \left(\boldsymbol{\phi}(s)\boldsymbol{\phi}(s')^\top\right)\right],$$

$$\bar{\boldsymbol{b}}_{\mathrm{PMF}} = \frac{1}{K+1}\mathbb{E}_{s,r}\left\{\left[\boldsymbol{C}\boldsymbol{C}^\top\boldsymbol{C}\left(\sum_{j=0}^{K}\boldsymbol{g}_j(r) - \mathbf{1}_K\right)\right] \otimes \boldsymbol{\phi}(s)\right\}.$$

Compared to our `Linear-CTD` (Eqn. (16)), this algorithm has an additional matrix $\boldsymbol{C}\boldsymbol{C}^\top$ with the condition number of order $K^2$ (see Lemma H.2).

Now, we are ready to state the theoretical results for the algorithm.

**Lemma E.1.** *For any $\boldsymbol{\theta} \in \mathbb{R}^{dK}$, it holds that*

$$\mathcal{L}(\boldsymbol{\theta}) \lesssim \frac{1}{\sqrt{K}(1-\gamma)^2\sqrt{\lambda_{\min}}}\left\|\bar{\boldsymbol{A}}_{\mathrm{PMF}}(\boldsymbol{\theta} - \boldsymbol{\theta}^\star)\right\|. \tag{40}$$

Lemma E.1 achieves the same order of bound as prior results for `Linear-CTD` (Lemma 5.1), as the minimum eigenvalue of $\boldsymbol{C}\boldsymbol{C}^\top$ remains $\Omega(1)$ (Lemma H.2). However, from the numerical experiments (Figure 6) in Appendix G.2, we observe that after substituting $\bar{\boldsymbol{\theta}}_t$ into $\boldsymbol{\theta}$, as $K$ grows, the RHS grows with $K$, while the LHS remains almost unchanged. This might be because when the matrix $\boldsymbol{C}\boldsymbol{C}^\top$ acts on the relevant random vectors, the stretching coefficient (*i.e.*, $\left\|\boldsymbol{C}\boldsymbol{C}^\top\boldsymbol{x}\right\| / \|\boldsymbol{x}\|$ for some vector $\boldsymbol{x}$) is usually of order $K^2$ rather than a constant order. For example, consider the case where the matrix $\boldsymbol{C}\boldsymbol{C}^\top$ acts on a random vector $\mathbf{X}$ that follows a uniform distribution over the surface of unit sphere ($\|\mathbf{X}\| = 1$). Since the $k$-th largest eigenvalue of the matrix $\boldsymbol{C}\boldsymbol{C}^\top$ is of order $k^2$, by Hanson-Wright inequality [Vershynin, 2018, Theorem 6.2.1], we have $\left\|\boldsymbol{C}\boldsymbol{C}^\top\mathbf{X}\right\|$ is of order $K^2$ with high probability.

**Lemma E.2.** *It holds that*

$$C_{A,\mathrm{PMF}} \lesssim K^2 C_A, \quad C_{e,\mathrm{PMF}} \lesssim K^2 C_e, \quad \mathrm{tr}(\boldsymbol{\Sigma}_{e,\mathrm{PMF}}) \lesssim K^4 \mathrm{tr}(\boldsymbol{\Sigma}_e).$$

Lemma E.2 introduces an additional factor of $K^2$ (or $K^4$) compared to previous results for `Linear-CTD` (Lemma 5.2) , since the maximum eigenvalue of $\boldsymbol{C}\boldsymbol{C}^\top$ is of order $K^2$.

**Lemma E.3.** *For any $p \geq 2$, let $a_{\mathrm{PMF}} \simeq (1 - \sqrt{\gamma})\lambda_{\min}$ and $\alpha_{p,\infty}^{\mathrm{PMF}} \simeq (1 - \sqrt{\gamma})/(pK^2)$ $(\alpha_{p,\infty}^{\mathrm{PMF}}p \leq 1/2)$. Then for any $\alpha \in (0, \alpha_{p,\infty}^{\mathrm{PMF}})$, $\boldsymbol{u} \in \mathbb{R}^{dK}$ and $t \in \mathbb{N}$*

$$\mathbb{E}^{1/p}\left[\left\|\boldsymbol{\Gamma}_{t,\mathrm{PMF}}^{(\alpha)}\boldsymbol{u}\right\|^p\right] \leq (1 - \alpha a_{\mathrm{PMF}})^t \|\boldsymbol{u}\|.$$

| Concepts | Linear-TD | Linear-CTD |
|---|---|---|
| Parametrization | $V_{\boldsymbol{\psi}}(s) = \boldsymbol{\phi}(s)^\top \boldsymbol{\psi}$ | $F_k(s; \boldsymbol{\theta}) = \boldsymbol{\phi}(s)^\top \boldsymbol{\theta}(k) + \frac{k+1}{K+1}$ |
| Bellman Operator | $(\boldsymbol{T}^\pi V)(s) = \mathbb{E}[r_0 + \gamma V(s_1) \mid s_0 = s]$ | $(\boldsymbol{\mathcal{T}}^\pi \eta)(s) = \mathbb{E}[(b_{r_0,\gamma})_\# \eta(s_1) \mid s_0 = s]$ |
| Projection Operator | $\boldsymbol{V}_{\tilde{\boldsymbol{\psi}}} = \boldsymbol{\Pi}_{\boldsymbol{\phi}}^\pi V$, $\boldsymbol{\psi} = \boldsymbol{\Sigma}_{\boldsymbol{\phi}}^{-1} \mathbb{E}_{s\sim\mu_\pi}[\boldsymbol{\phi}(s) V(s)]$ | $\boldsymbol{\eta}_{\tilde{\boldsymbol{\theta}}} = \boldsymbol{\Pi}_{\boldsymbol{\phi},K}^\pi \boldsymbol{\eta}$, $\boldsymbol{\Theta} = \boldsymbol{\Sigma}_{\boldsymbol{\phi}}^{-1} \mathbb{E}_{s\sim\mu_\pi}[\boldsymbol{\phi}(s)(\boldsymbol{F}_{\boldsymbol{\eta}}(s) - \boldsymbol{F}_\nu)^\top]$ |
| Projected Bellman Equation | $\boldsymbol{V}_{\boldsymbol{\psi}} = \boldsymbol{\Pi}_{\boldsymbol{\phi}}^\pi \boldsymbol{T}^\pi \boldsymbol{V}_{\boldsymbol{\psi}}$, [See Eqn. (4)] | $\boldsymbol{\eta}_{\boldsymbol{\theta}} = \boldsymbol{\Pi}_{\boldsymbol{\phi},K}^\pi \boldsymbol{\mathcal{T}}^\pi \boldsymbol{\eta}_{\boldsymbol{\theta}}$, [See Eqn. (12)] |
| Update Rule | $\boldsymbol{\psi}_t \leftarrow \boldsymbol{\psi}_{t-1} - \alpha\boldsymbol{\phi}(s_t)\left[(\boldsymbol{\phi}(s_t) - \gamma\boldsymbol{\phi}(s_{t+1}))^\top \boldsymbol{\psi}_{t-1} - r_t\right]$ | [See Eqn. (13)] |
| $\boldsymbol{A}_t$ | $\boldsymbol{\phi}(s_t)\boldsymbol{\phi}(s_t)^\top - \gamma\boldsymbol{\phi}(s_t)\boldsymbol{\phi}(s_{t+1})^\top$ | $[\boldsymbol{I}_K \otimes (\boldsymbol{\phi}(s_t)\boldsymbol{\phi}(s_t)^\top)] - [(\boldsymbol{C}\tilde{\boldsymbol{G}}^{-1}(r_t)\boldsymbol{C}^{-1}) \otimes (\boldsymbol{\phi}(s_t)\boldsymbol{\phi}(s_{t+1})^\top)]$ |
| Key Quantity in $\boldsymbol{A}_t$ | $\gamma$ | $\boldsymbol{C}\tilde{\boldsymbol{G}}^{-1}(r_t)\boldsymbol{C}^{-1}$ with spectral norm $\sqrt{\gamma}$ |
| $\boldsymbol{b}_t$ | $r_t\boldsymbol{\phi}(s_t)$ | $\frac{1}{K+1}[\boldsymbol{C}(\sum_{j=0}^K \boldsymbol{g}_j(r_t) - \boldsymbol{1}_K)] \otimes \boldsymbol{\phi}(s_t)$ |
| Key Quantity in $\boldsymbol{b}_t$ | $r_t \leq 1$ | $K^{-3/2}(1-\gamma)^{-1}\|\boldsymbol{C}(\sum_{j=0}^K \boldsymbol{g}_j(r_t) - \boldsymbol{1}_K)\| \leq 1$ |
| Measure of Error | $\|\boldsymbol{V} - \boldsymbol{V}^\pi\|_{\mu_\pi} = (\mathbb{E}_{s\sim\mu_\pi}[(V(s) - V^\pi(s))^2])^{1/2}$ | $W_{1,\mu_\pi}(\boldsymbol{\eta}, \boldsymbol{\eta}^\pi) = (\mathbb{E}_{s\sim\mu_\pi}[W_1^2(\eta(s), \eta^\pi(s))])^{1/2}$ |
| Approximation Error | $\|\boldsymbol{V}_{\boldsymbol{\psi}^\star} - \boldsymbol{V}^\pi\|_{\mu_\pi} \leq (1-\gamma^2)^{-1/2}\|\boldsymbol{\Pi}_{\boldsymbol{\phi}}^\pi \boldsymbol{V}^\pi - \boldsymbol{V}^\pi\|_{\mu_\pi}$ | [See Proposition 3.3] |
| Sample Complexity | $\widetilde{\mathcal{O}}\left(\frac{\|\boldsymbol{\psi}^\star\|_{\boldsymbol{\Sigma}_{\boldsymbol{\phi}}}^2 + 1}{(1-\gamma)^2\lambda_{\min}}\left(\frac{1}{\varepsilon^2} + \frac{1}{\lambda_{\min}}\right)\right)$ | $\widetilde{\mathcal{O}}\left(\frac{\|\boldsymbol{\theta}^\star\|_{V_1}^2 + 1}{(1-\gamma)^2\lambda_{\min}}\left(\frac{1}{\varepsilon^2} + \frac{1}{\lambda_{\min}}\right)\right)$ |

**Table 2:** Comparison between `Linear-TD` and `Linear-CTD`.

As before, in this lemma, $a_{\text{PMF}}$ does not depend on $K$ because the minimum eigenvalue of $\boldsymbol{C}\boldsymbol{C}^\top$ is $\Omega(1)$, and $\alpha_{p,\infty}^{\text{PMF}}$ scales with $K^{-2}$ because the maximum eigenvalue of $\boldsymbol{C}\boldsymbol{C}^\top$ is of order $K^2$.

**Theorem E.2.** *For any $K \geq (1-\gamma)^{-1}$ and $\alpha \in (0, \alpha_{p,\infty}^{\text{PMF}})$, it holds that*

$$
\mathbb{E}^{1/2}[(\mathcal{L}(\bar{\boldsymbol{\theta}}_{\text{PMF},T}))^2] \lesssim \frac{K^2(\|\boldsymbol{\theta}^\star\|_{V_1} + 1)}{\sqrt{T}(1-\gamma)\sqrt{\lambda_{\min}}}\left(1 + K\sqrt{\frac{\alpha K^2}{(1-\gamma)\lambda_{\min}}}\right) + \frac{K^3(\|\boldsymbol{\theta}^\star\|_{V_1} + 1)}{T\sqrt{\alpha K^2}(1-\gamma)^{\frac{3}{2}}\lambda_{\min}}
$$
$$
+ K\frac{(1 - \frac{1}{2}(\alpha K^2)K^{-2}(1-\sqrt{\gamma})\lambda_{\min})^{T/2}}{T\sqrt{\alpha K^2}(1-\gamma)\sqrt{\lambda_{\min}}}\left(\frac{K}{\sqrt{\alpha K^2}} + \frac{K^2}{\sqrt{(1-\gamma)\lambda_{\min}}}\right)\|\boldsymbol{\theta}_{\text{PMF},0} - \boldsymbol{\theta}^\star\|_{V_2}.
$$

This theorem for the PMF version algorithm yields an upper bound that is $K^3$ times looser than Theorem 4.1 for our `Linear-CTD`. The appearance of the $K^3$ factor is due to the fact that the condition number of the redundant matrix $\boldsymbol{C}\boldsymbol{C}^\top$ is of order $K^2$. This factor is unavoidable within our theoretical analysis framework.

However, from the numerical experiments (Table 3 and Table 4) in Appendix G.2, we can only observe that our `Linear-CTD` consistently outperforms the PMF version algorithm under different values of $K$, but the performance gap does not increase significantly when $K$ becomes larger as predicted by Theorem 4.1 and Theorem E.2. The reason for this might be, as discussed after Lemma E.1: in the experimental environment we have set, when the matrix $\boldsymbol{C}\boldsymbol{C}^\top$ acts on the vectors it encountered, the stretching coefficient is usually of order $K^2$ rather than a constant order. See the numerical experiments (Figure 6) in Appendix G.2 for some evidence.

# F Comparison between Linear-TD and Linear-CTD

To further improve the readability of the paper, we compare various concepts and results between `Linear-TD` and `Linear-CTD` in Table 2.

# G Numerical Experiment

In this appendix, we validate the proposed `Linear-CTD` algorithm (Eqn. (13)) with numerical experiments, and show its advantage over the baseline algorithm, stochastic semi-gradient descent (SSGD) with the probability mass function (PMF) representation (Eqn. (39)).

To empirically evaluate our `Linear-CTD` algorithm, we consider a 3-state MDP with $\gamma = 0.75$. When the number of states is finite, we denote by $\boldsymbol{\Phi} = (\boldsymbol{\phi}(s))_{s\in\mathcal{S}} \in \mathbb{R}^{d\times\mathcal{S}}$ the feature matrix. Here, we set the feature matrix $\boldsymbol{\Phi}$ to be a full-rank matrix in $\mathbb{R}^{3\times3}$. The following experiments share zero initialization $\boldsymbol{\theta}_0 = \boldsymbol{0}$ with max_iteration=500000 and batch_size=25.

All of the experiments are conducted on a server with 4 NVIDIA RTX 4090 GPUs and Intel(R) Xeon(R) Gold 6132 CPU @ 2.60GHz.

### G.1 Empirical Convergence of `Linear-CTD`

We employ the `Linear-CTD` algorithm in the above environment and have the following convergence results in Figure 2. This figure shows the negative logarithm of $\frac{1}{K} \left\| \bar{\boldsymbol{\theta}}_t - \boldsymbol{\theta}^\star \right\|^2_{\boldsymbol{I}_K \otimes \boldsymbol{\Sigma}_\phi} = (1 -$

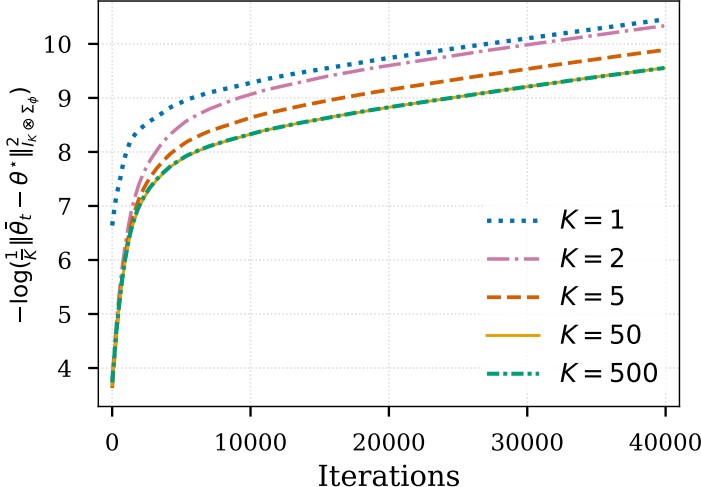

**Figure 2.** Convergence results under varying $K$ for our `Linear-CTD` algorithm with step size $\alpha = 0.01$. These curves exhibit similar trends, demonstrating our algorithm's robustness across different $K$ values.

$\gamma)\ell^2_{2,\mu_\pi}(\boldsymbol{\eta}_{\bar{\boldsymbol{\theta}}_t}, \boldsymbol{\eta}_{\boldsymbol{\theta}^\star})$ along iterations. We observe that our `Linear-CTD` algorithm can converge for different values of $K$ when we set the step size as $\alpha = 0.01$.

### G.2 Comparison with SSGD with the PMF Representation

First, we repeat the same experiment as in the previous section for the baseline algorithm, SSGD with the PMF representation. The experimental results in Figure 3 demonstrate that when the baseline algorithm uses a fixed step size $\alpha = 0.01$, it does not converge when $K$ is large ($K \geq 44$). The results in Figure 2 and Figure 3 verify the advantage of our `Linear-CTD` over the baseline algorithm as mentioned in Remark 5: when $K$ increases, the step size of the baseline algorithm needs to decay (Lemma E.3). In contrast, the step size of our `Linear-CTD` does not need to be adjusted when $K$ increases (Lemma 5.3).

Next, we will verify that the maximum step size $\alpha_\infty^{\mathrm{PMF},(K)}$ that ensures the convergence of the baseline algorithm scales with $K^{-2}$, as predicted in Lemma E.3. Then we will compare the convergence rate of the baseline algorithm with that of our `Linear-CTD` algorithm.

In Figure 4, we employ the baseline algorithm with different step sizes under fixed $K = 150$, and we find that the baseline algorithm converges when the step size does not exceed $8.6\mathrm{e}{-4}$, and it does not converge when the step size exceeds $8.7\mathrm{e}{-4}$. This indicates that $\alpha_\infty^{\mathrm{PMF},(150)} \in [8.6\mathrm{e}{-4}, 8.7\mathrm{e}{-4}]$ in this environment, providing a good approximation of $\alpha_\infty^{\mathrm{PMF},(150)}$.

We repeat the above experiments under varying $K$, looking for a step size that can ensure convergence (a lower bound of $\alpha_\infty^{\mathrm{PMF},(K)}$) and a step size that leads to divergence (an upper bound of $\alpha_\infty^{\mathrm{PMF},(K)}$) such that the two step sizes are as close as possible and thereby we can get a good approximation of $\alpha_\infty^{\mathrm{PMF},(K)}$. The results are summarized in Table 3.

In Figure 5, we use the approximate values of $\alpha_\infty^{\mathrm{PMF},(K)}$ provided in Table 3 to perform a quadratic function fitting of $1/\alpha_\infty^{\mathrm{PMF},(K)}$ with respect to $K$. We find that $\alpha_\infty^{\mathrm{PMF},(K)}$ indeed approximately scales with $K^{-2}$, which verifies our theoretical result (Lemma E.3).

To compare the statistical efficiency of our `Linear-CTD` algorithm and the baseline algorithm, in Table 3, we also report the number of iterations required for the error to reach below $2\mathrm{e}{-6}$ when the

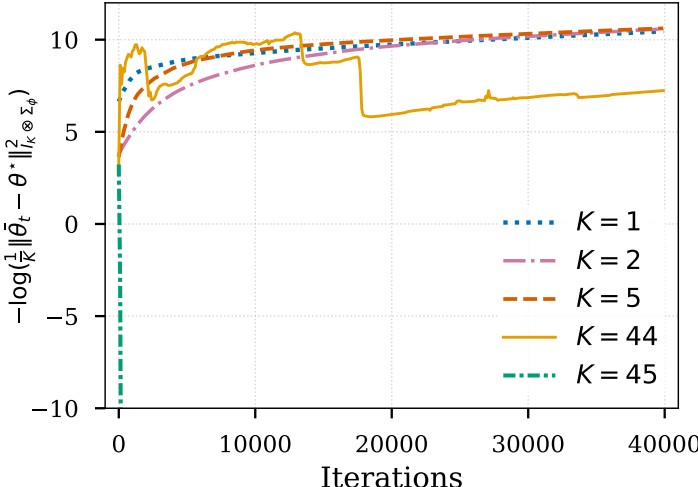

**Figure 3.** Convergence results under varying $K$ for the baseline algorithm, SSGD with the PMF representation with step size $\alpha = 0.01$. We remark that when $K = 45$, the program reports errors of inf and nan. In contrast to results of `Linear-CTD` in Figure 2, the baseline algorithm no longer converges when $K$ is large ($K \geq 44$).

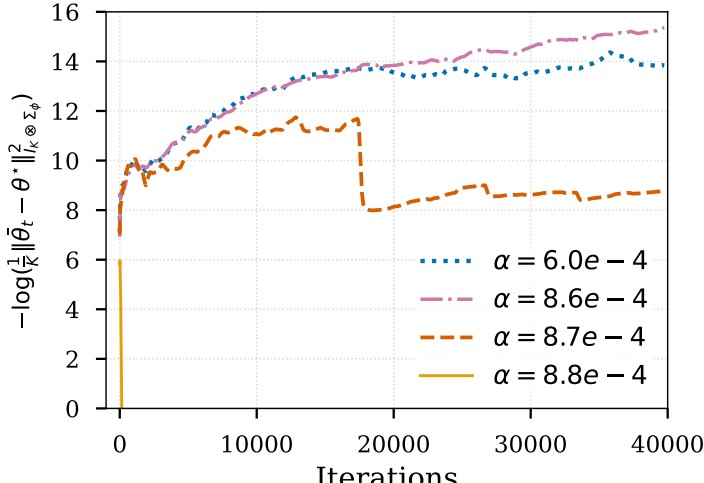

**Figure 4.** Convergence results with different step sizes for the baseline algorithm, SSGD with the PMF representation under fixed $K = 150$. We remark that when we take $\alpha = 8.8e-4$, the program reports errors of inf and nan. The baseline algorithm converges when the step size does not exceed $8.6e-4$, and it does not converge when the step size exceeds $8.7e-4$. Therefore, $\alpha_\infty^{\mathrm{PMF},(150)} \in [8.6e-4, 8.7e-4]$ in this environment.

step size satisfies $\alpha \approx 0.2\alpha_\infty^{\mathrm{PMF},(K)}$. In addition, we present the parallel results of our `Linear-CTD` in Table 4. In Table 4, we find that the value of $\alpha_\infty^{(K)}$ for our `Linear-CTD` algorithm is much larger than $\alpha_\infty^{\mathrm{PMF},(K)}$, and it does not decrease significantly with the growth of $K$. Moreover, by comparing the **Iterations** columns in Table 3 and Table 4, we find that the sample complexity of our `Linear-CTD` does not increase significantly with the growth of $K$, and `Linear-CTD` empirically consistently outperforms the baseline algorithm under different $K$.

However, the performance gap does not increase significantly as expected when $K$ increases as predicted by Theorem 4.1 and Theorem E.2. The reason for this might be that, as discussed after Lemma E.1, in the experimental environment we have set, when the matrix $CC^\top$ acts on the vectors

| $K$ | Lower Bound of $\alpha_\infty^{\mathrm{PMF},(K)}$ | Upper Bound of $\alpha_\infty^{\mathrm{PMF},(K)}$ | Iterations |
|---|---|---|---|
| 30 | 2.1e-2 | 2.2e-2 | 37245 |
| 45 | 9e-3 | 9.5e-3 | 39262 |
| 75 | 3.4e-3 | 3.5e-3 | 38286 |
| 105 | 1.75e-3 | 1.8e-3 | 38123 |
| 150 | 8.6e-4 | 8.7e-4 | 38556 |
| 225 | 3.8e-4 | 3.9e-4 | 38317 |
| 300 | 2.1e-4 | 2.2e-4 | 38999 |
| 375 | 1.35e-4 | 1.4e-4 | 38674 |
| 450 | 9.5e-5 | 9.8e-5 | 38506 |

**Table 3.** Lower and upper bounds of the maximum step size $\alpha_\infty^{\mathrm{PMF},(K)}$ that ensures the convergence under varying $K$ for the baseline algorithm, SSGD with the PMF representation. The bounds are determined using the same method as that in Figure 4. The **Iterations** column refers to the number of iterations required for the error to reach below 2e−6 when the step size satisfies $\alpha \approx 0.2\alpha_\infty^{\mathrm{PMF},(K)}$.

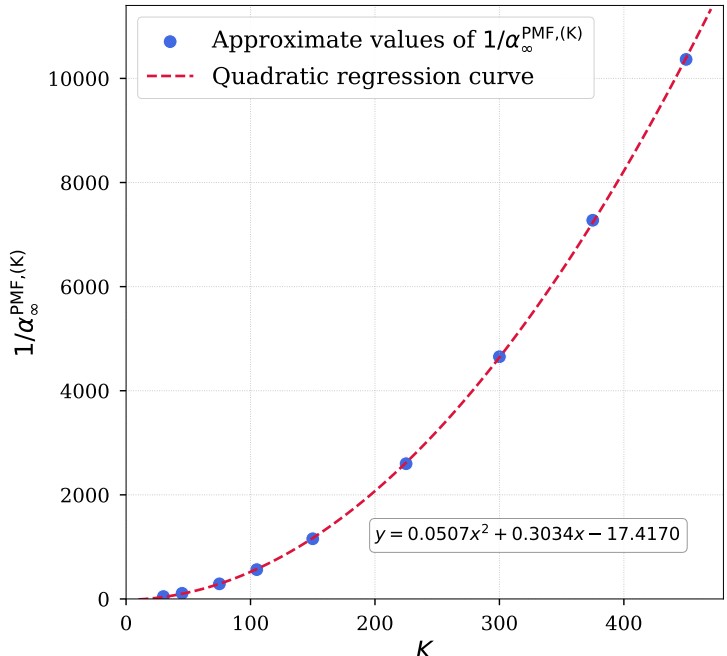

**Figure 5.** The approximate values of of maximum step sizes $1/\alpha_\infty^{\mathrm{PMF},(K)}$ under varying $K$. Here we take the average of the upper and lower bounds of $\alpha_\infty^{\mathrm{PMF},(K)}$ provided in Table 3 as an approximation of $\alpha_\infty^{\mathrm{PMF},(K)}$ and perform quadratic regression of $1/\alpha_\infty^{\mathrm{PMF},(K)}$ on $K$. This fit achieves a mean squared error of 425.85 and $R^2$ of 0.99996, which indicates that $1/\alpha_\infty^{\mathrm{PMF},(K)}$ indeed grows quadratically with respect to $K$, aligning with our theoretical results (Lemma E.3).

it encountered, the stretching coefficient (*i.e.*, $\left\|CC^\top x\right\| / \|x\|$ for some vector $x$) is usually of order $K^2$ rather than a constant order.

We verify this conjecture through the following experiment. We focus on the LHS and RHS of Eqn. (40) in Lemma E.1:

$$\mathcal{L}(\boldsymbol{\theta}) \lesssim \frac{1}{\sqrt{K}(1-\gamma)^2\sqrt{\lambda_{\min}}} \left\|\bar{A}_{\mathrm{PMF}}\left(\boldsymbol{\theta}-\boldsymbol{\theta}^\star\right)\right\|,$$

where

$$\bar{A}_{\mathrm{PMF}} = \left[\left(CC^\top\right) \otimes \Sigma_\phi\right] - \mathbb{E}_{s,r,s'}\left[\left(CC^\top(C\tilde{G}(r_t)C^{-1})\right) \otimes \left(\phi(s)\phi(s')^\top\right)\right].$$

In our theoretical analysis, we first give an upper bound of the RHS, and then apply Lemma E.1 bound the loss function $\mathcal{L}(\boldsymbol{\theta})$ in the LHS with the RHS. However, since the minimum eigenvalue

| $K$ | Lower Bound of $\alpha_\infty^{(K)}$ | Upper Bound of $\alpha_\infty^{(K)}$ | Iterations |
|---|---|---|---|
| 30 | 1.65 | 1.7 | 17908 |
| 45 | 1.65 | 1.7 | 17925 |
| 75 | 1.65 | 1.7 | 17942 |
| 105 | 1.6 | 1.65 | 18623 |
| 150 | 1.5 | 1.55 | 21947 |
| 225 | 1.55 | 1.6 | 20890 |
| 300 | 1.55 | 1.6 | 20890 |
| 375 | 1.55 | 1.65 | 20595 |
| 450 | 1.5 | 1.55 | 21947 |

**Table 4.** Lower and upper bounds of the maximum step size $\alpha_\infty^{(K)}$ that ensures the convergence under varying $K$ for our `Linear-CTD`. The bounds are determined using the same method as that in Figure 4. The **Iterations** column refers to the number of iterations required for the error to reach below $2e-6$ the step size satisfies $\alpha \approx 0.2\alpha_\infty^{(K)}$.

of the matrix $CC^\top$ in the RHS is only of a constant order, we are unable to have a term of $1/K^2$ in the RHS. Therefore, our conjecture can be verified by checking whether the bound provided in Lemma E.1 is tight in this environment, which is presented in Figure 6. The left sub-graph of Figure 6 corresponds to the LHS, and the right sub-graph corresponds to the RHS. We omit the constants that are independent of $K$. From Figure 6, we can find that the LHS remains almost unchanged under different $K$, but the RHS increases as $K$ becomes larger. This indicates that the stretching coefficient of the matrix $CC^\top$ that we frequently encounters during the iterative process grows with $K$ rather than remaining a constant order. A similar analysis also holds for $a_{\mathrm{PMF}}$ in Lemma E.3, and we omit it for brevity. These factors result in the performance gap between our `Linear-CTD` algorithm and the baseline algorithm not increasing significantly when $K$ becomes larger, as predicted by Theorem 4.1 and Theorem E.2.

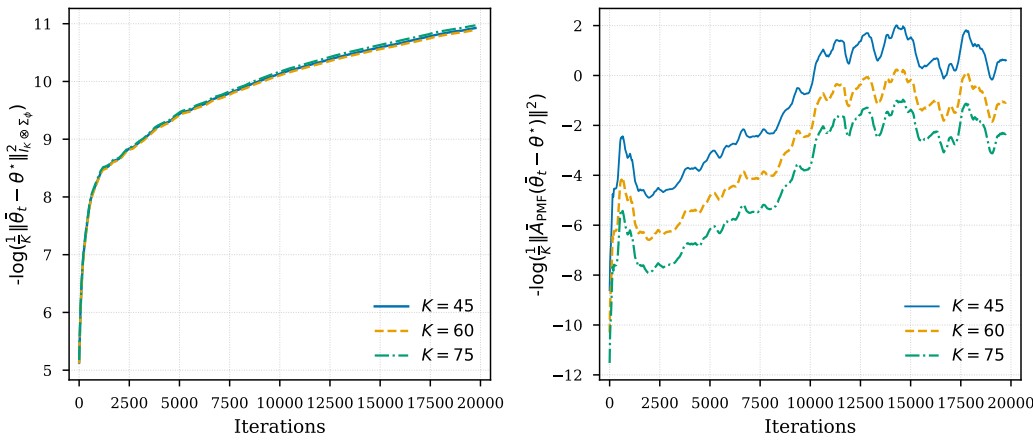

**Figure 6.** LHS and RHS of Eqn. (40) in Lemma E.1 under varying $K$. The left sub-graph corresponds to the LHS, and the right sub-graph corresponds to the RHS. We omit the constants that are independent of $K$. We can find that the LHS remains almost unchanged under different $K$, but the RHS increases as $K$ becomes larger, indicating that the stretching coefficient of the matrix $CC^\top$ that we frequently encounters during the iterative process grows with $K$ rather than remaining a constant order.

## H  Other Technical Lemmas

**Lemma H.1.** *For any $\nu_1, \nu_2 \in \mathscr{P}^{\mathsf{sign}}$, we have $W_1(\nu_1, \nu_2) \leq \frac{1}{\sqrt{1-\gamma}} \ell_2(\nu_1, \nu_2)$.*

*Proof.* By Cauchy-Schwarz inequality,

$$
\begin{aligned}
W_1(\nu_1, \nu_2) &= \int_0^{\frac{1}{1-\gamma}} |F_{\nu_1}(x) - F_{\nu_2}(x)| dx \\
&\leq \sqrt{\int_0^{\frac{1}{1-\gamma}} 1^2 dx} \sqrt{\int_0^{\frac{1}{1-\gamma}} |F_{\nu_1}(x) - F_{\nu_2}(x)|^2 dx} \\
&= \frac{1}{\sqrt{1-\gamma}} \ell_2(\nu_1, \nu_2).
\end{aligned}
$$

$\square$

**Lemma H.2.** *Let $C \in \mathbb{R}^{K \times K}$ be the matrix defined in Eqn. (11), it holds that the eigenvalues of $C^T C$ are $1/(4\cos^2(k\pi/(2K+1))$ for $k \in [K]$, and thus*

$$
\left\| C^\top C \right\| = \frac{1}{4\sin^2 \frac{\pi}{4K+2}} \leq K^2, \quad \left\| (C^\top C)^{-1} \right\| = 4\cos^2 \frac{\pi}{2K+1} \leq 4.
$$

*Proof.* One can check that

$$
C^\top C = \begin{bmatrix}
K & K-1 & \cdots & 2 & 1 \\
K-1 & K-1 & \cdots & 2 & 1 \\
\vdots & \vdots & \ddots & \vdots & \vdots \\
2 & 2 & \cdots & 2 & 1 \\
1 & 1 & \cdots & 1 & 1
\end{bmatrix},
$$

$$
(C^\top C)^{-1} = \begin{bmatrix}
1 & -1 & 0 & \cdots & 0 & 0 \\
-1 & 2 & -1 & \cdots & 0 & 0 \\
0 & -1 & 2 & \cdots & 0 & 0 \\
\vdots & \vdots & \vdots & \ddots & \vdots & \vdots \\
0 & 0 & 0 & \cdots & 2 & -1 \\
0 & 0 & 0 & \cdots & -1 & 1
\end{bmatrix}.
$$

Then, one can work with the the inverse of $C^\top C$ and calculate its singular values by induction, which has similar forms to the analysis of Toeplitz's matrix. See Godsil [1985] for more details. $\square$

**Lemma H.3.** *For any $r \in [0,1]$, it holds that $\left\| C\tilde{G}(r)C^{-1} \right\| \leq \sqrt{\gamma}$ and $\left\| (C^\top C)^{1/2} \tilde{G}(r) (C^\top C)^{-1/2} \right\| \leq \sqrt{\gamma}.$*

*Proof.* One can check that

$$
C^{-1} = \begin{bmatrix}
1 & 0 & \cdots & 0 & 0 \\
-1 & 1 & \cdots & 0 & 0 \\
0 & -1 & \cdots & 0 & 0 \\
\vdots & \vdots & \ddots & \vdots & \vdots \\
0 & 0 & \cdots & -1 & 1
\end{bmatrix}.
$$

It is clear that

$$
\begin{aligned}
\left\| (C^\top C)^{1/2} \tilde{G}(r) (C^\top C)^{-1/2} \right\| \leq \sqrt{\gamma} &\iff (C^\top C)^{1/2} \tilde{G}(r)(C^\top C)^{-1}\tilde{G}^\top(r) (C^\top C)^{1/2} \preccurlyeq \gamma I_K \\
&\iff \tilde{G}(r)(C^\top C)^{-1}\tilde{G}^\top(r) \preccurlyeq \gamma (C^\top C)^{-1} \\
&\iff C\tilde{G}(r)(C^\top C)^{-1}\tilde{G}^\top(r)C^\top \preccurlyeq \gamma I_K \\
&\iff \left\| C\tilde{G}(r)C^{-1} \right\| \leq \sqrt{\gamma}.
\end{aligned}
$$

By Lemma I.2 and an upper bound on the spectral norm (Riesz–Thorin interpolation theorem) [Serre, 2002, Theorem 7.3], we obtain that

$$
\left\| C\tilde{G}(r)C^{-1} \right\| \leq \sqrt{\left\| C\tilde{G}(r)C^{-1} \right\|_1 \left\| C\tilde{G}(r)C^{-1} \right\|_\infty} \leq \sqrt{1 \cdot \gamma} = \sqrt{\gamma}.
$$

$\square$

**Lemma H.4.** *Suppose $K \geq (1-\gamma)^{-1}$, $\nu = (K+1)^{-1} \sum_{k=0}^{K} \delta_{x_k}$ is the discrete uniform distribution, then for any $r \in [0,1]$, it holds that*

$$\ell_2 \left((b_{r,\gamma})_\# (\nu), \nu\right) \leq 3\sqrt{1-\gamma}.$$

*Proof.* Let $\tilde{\nu}$ be the continuous uniform distribution on $\left[0, (1-\gamma)^{-1} + \iota_K\right]$, we consider the following decomposition

$$\ell_2 \left(\nu, (b_{r,\gamma})_\# (\nu)\right) \leq \ell_2 \left(\nu, \tilde{\nu}\right) + \ell_2 \left(\tilde{\nu}, (b_{r,\gamma})_\# (\tilde{\nu})\right) + \ell_2 \left((b_{r,\gamma})_\# (\tilde{\nu}), (b_{r,\gamma})_\# (\nu)\right).$$

By definition, we have

$$\ell_2 \left(\nu, \tilde{\nu}\right) = \sqrt{(K+1) \int_0^{\iota_K} \left((1-\gamma)\frac{K}{K+1}x\right)^2 dx}$$

$$= \sqrt{\frac{1}{3K(K+1)(1-\gamma)}}$$

$$\leq \frac{1}{K\sqrt{1-\gamma}}.$$

By the contraction property, we have

$$\ell_2 \left((b_{r,\gamma})_\# (\nu), (b_{r,\gamma})_\# (\tilde{\nu})\right) \leq \sqrt{\gamma}\ell_2 \left(\nu, \tilde{\nu}\right) \leq \frac{\sqrt{\gamma}}{K\sqrt{1-\gamma}}.$$

We only need to bound $\ell_2 \left(\tilde{\nu}, (b_{r,\gamma})_\# (\tilde{\nu})\right)$. We can find that $(b_{r,\gamma})_\# (\tilde{\nu})$ is the continuous uniform distribution on $\left[r, r + \gamma\iota_K + \gamma(1-\gamma)^{-1}\right]$, and the upper bound is less than the upper bound of $\nu$, namely, $r + \gamma\iota_K + \gamma(1-\gamma)^{-1} \leq (1-\gamma)^{-1} + \gamma\iota_K < (1-\gamma)^{-1} + \iota_K$. Hence

$$\ell_2^2 \left(\tilde{\nu}, (b_{r,\gamma})_\# (\tilde{\nu})\right) = \int_0^r \left((1-\gamma)\frac{K}{K+1}x\right)^2 dx + \int_r^{r+\gamma\iota_K+\gamma(1-\gamma)^{-1}} \left[(1-\gamma)\frac{K}{K+1}\left(x - \frac{x-r}{\gamma}\right)\right]^2 dx$$

$$+ \int_{r+\gamma\iota_K+\gamma(1-\gamma)^{-1}}^{(1-\gamma)^{-1}+\iota_K} \left(1 - (1-\gamma)\frac{K}{K+1}x\right)^2 dx$$

$$= \frac{(1-\gamma)^2 K^2 r^3}{3(K+1)^2} + \left(\frac{(1-\gamma)\gamma K^2 r^3}{3(K+1)^2} + \frac{(1-\gamma)\gamma K^2 \left(\frac{K+1}{K} - r\right)^3}{3(K+1)^2}\right) + \frac{(1-\gamma)^2 K^2 \left(\frac{K+1}{K} - r\right)^3}{3(K+1)^2}$$

$$\leq (1-\gamma)^2 + (1-\gamma)\gamma$$

$$= 1 - \gamma.$$

To summarize, we have

$$\ell_2 \left(\nu, (b_{r,\gamma})_\# (\nu)\right) \leq \frac{1}{K\sqrt{1-\gamma}} + \sqrt{1-\gamma} + \frac{\sqrt{\gamma}}{K\sqrt{1-\gamma}} \leq 3\sqrt{1-\gamma},$$

where we used the assumption $K \geq (1-\gamma)^{-1}$. $\qquad\square$

# I Analysis of the Categorical Projected Bellman Matrix

Recall that $\tilde{\boldsymbol{G}}(r) = \boldsymbol{G}(r) - \mathbf{1}_K^\top \otimes \boldsymbol{g}_K(r)$. We extend the definition in Theorem 3.1 and let $g_{j,k}(r) = h\left(((r+\gamma x_j - x_k)/\iota_K)_+\right) = h(r/\iota_K + \gamma j - k)$ for $j, k \in \{0, 1, \cdots, K\}$ where $h(x) = (1 - |x|)_+$.

**Lemma I.1.** *For any $r \in [0,1]$ and any $k \in \{0, 1, \cdots, K\}$, in $\boldsymbol{g}_k(r)$ there is either only one nonzero entry or two adjacent nonzero entries.*

*Proof.* It is clear that $h(x) > 0 \iff -1 < x < 1$. Let $k_j(r) = \min\{k : g_{j,k}(r) > 0\}$, then $k_j(r) = \min\{k : r/\iota_K + \gamma j - k < 1\} = \min\{k : 0 \leq r/\iota_K + \gamma j - k < 1\}$. The existence of $k_j(r)$ is due to

$$r/\iota_K + \gamma j - K \leq 1/\iota_K + \gamma j - K \leq (1-\gamma)K + \gamma K - K = 0 < 1.$$

Let $a_j(r) := r/\iota_K + \gamma j - k_j(r) \in [0, 1)$. Then $g_{j,k_j(r)}(r) = h(a_j(r)) = 1 - a_j(r)$ and $g_{j,k_j(r)+1}(r) = h(a_j(r) - 1) = a_j(r)$ are the only entries that can be nonzeros. $\qquad\square$

The following results are immediate corollaries.

**Corollary I.1.**

$$\sum_{k=0}^{i} g_{j,k}(r) = \begin{cases} 0, & \text{for } 0 \le i < k_j(r), \\ 1 - a_j(r), & \text{for } i = k_j(r), \\ 1, & \text{for } k_j(r) < i \le K. \end{cases}$$

**Corollary I.2.**

$$k_{j+1}(r) = \begin{cases} k_j(r), & \text{if } a_j(r) \le 1 - \gamma, \\ k_j(r) + 1, & \text{if } a_j(r) > 1 - \gamma. \end{cases}$$

*As a result,*

$$a_{j+1}(r) = \begin{cases} a_j(r) + \gamma, & \text{if } a_j(r) \le 1 - \gamma, \\ a_j(r) + \gamma - 1, & \text{if } a_j(r) > 1 - \gamma. \end{cases}$$

**Lemma I.2.** *All entries in* $C\tilde{G}(r)C^{-1}$ *are non-negative.* $\left\| C\tilde{G}(r)C^{-1} \right\|_\infty = \gamma$ *and* $\left\| C\tilde{G}(r)C^{-1} \right\|_1 \le 1.$

*Proof.* By definition the entries of $\tilde{G}(r)$ are

$$(\tilde{G}(r))_{j,i} = g_{j,i}(r) - g_{K,i}(r) \quad \text{for } j, i \in \{0, 1, \cdots, K - 1\}.$$

Using the previous corollaries, through direct calculation we have that if $k_{j+1}(r) = k_j(r)$,

$$(C\tilde{G}(r)C^{-1})_{j,i} = \sum_{k=0}^{i} g_{j,k}(r) - \sum_{k=0}^{i} g_{j+1,k}(r) = \begin{cases} 0, & \text{for } 0 \le i < k_j(r), \\ a_{j+1}(r) - a_j(r), & \text{for } i = k_j(r), \\ 0, & \text{for } k_j(r) < i < K. \end{cases}$$

And if $k_{j+1}(r) = k_j(r) + 1$,

$$(C\tilde{G}(r)C^{-1})_{j,i} = \sum_{k=0}^{i} g_{j,k}(r) - \sum_{k=0}^{i} g_{j+1,k}(r) = \begin{cases} 0, & \text{for } 0 < i < k_j(r), \\ 1 - a_j(r), & \text{for } i = k_j(r), \\ a_{j+1}(r), & \text{for } i = k_{j+1}(r), \\ 0, & \text{for } k_j(r) < i < K. \end{cases}$$

As a result, all entries in $C\tilde{G}(r)C^{-1}$ is non-negative. Moreover, the sum of each column and $\left\| C\tilde{G}(r)C^{-1} \right\|_\infty$ is $\gamma$ since

$$\sum_{i=0}^{K-1} (C\tilde{G}(r)C^{-1})_{j,i} = \begin{cases} a_{j+1}(r) - a_j(r) = \gamma, & \text{if } k_{j+1}(r) = k_j(r), \\ 1 - a_j(r) + a_{j+1}(r) = \gamma, & \text{if } k_{j+1}(r) = k_j(r) + 1. \end{cases}$$

Moreover, the row sum of $C\tilde{G}(r)C^{-1}$ is

$$\sum_{j=0}^{K-1} (C\tilde{G}(r)C^{-1})_{j,i} = \sum_{m=0}^{i} g_{0,m}(r) - \sum_{m=0}^{i} g_{K,m}(r) \le 1 - 0 = 1.$$

Thus, it holds that $\left\| C\tilde{G}(r)C^{-1} \right\|_1 \le 1.$ □

