# OpenReview forum: "A Finite Sample Analysis of Distributional TD Learning with Linear Function Approximation"
_NeurIPS.cc/2025/Conference — NeurIPS 2025 poster_

### Official Review · Reviewer_VGyk · 2025-06-15

**Clarity:** 4
**Significance:** 3
**Originality:** 3
**Rating:** 5
**Confidence:** 4

**Summary:**

This paper investigates the finite-sample statistical rate of distributional TD Learning under linear function approximation. The authors propose a Linear-CTD which achieves finite-sample convergence rates that are independent of the number of categorical supports, a key improvement over vanilla approaches. They establish the first non-asymptotic sample complexity bounds for distributional TD and derive finite-sample rates that match those of classical TD learning, demonstrating that estimating the full return distribution is statistically no hard than estimating its expectation. Theoretical contributions are complemented by empirical results validating the improved performance and robustness of Linear-CTD.

**Questions:**

N/A

**Ethical Concerns:**

["NO or VERY MINOR ethics concerns only"]

**Final Justification:**

I have no remaining questions and would keep my score.

**Paper Formatting Concerns:**

One suggestion for improvement would be to include a table summarizing the main theoretical results and comparing them with prior work, which could enhance clarity and accessibility (the current version is somewhat dense and requires flipping back and forth to see the comparison). Additionally, the authors may consider presenting some of the empirical results in the main text, despite space limitations.

**Quality:**

4

**Strengths And Weaknesses:**

In my opinion, the novelty of this paper lies in its theoretical development, which establishes the first non-asymptotic sample complexity bounds for distributional TD learning under linear function approximation. This fills the gap in the RL. The authors show that the results are consistent with classical TD rates, providing the important insight that learning the full return distribution is no harder than learning its expectation. The paper is well written and clearly organized. In particular, I like Section 5, which outlines the proof strategy, making the non-asymptotic analysis transparent and easy to follow. Theoretical findings are also supported by empirical experiments, which reinforce the effectiveness of the proposed approach. I do not see major weaknesses in the paper, although I may overestimate the novelty and miss some technical issues, as I did not check the proofs line by line.

---

> ### Author Rebuttal · Authors · 2025-07-29
>
> Thank you for your encouraging comments and constructive suggestions.
> We are very glad to know that you find our theoretical results (Linear-CTD proposed in the paper achieves finite-sample convergence rates that are consistent with classical TD rates, and are independent of the number of categorical supports $K$, a key improvement over vanilla approaches) novel, and agree that the key conclusion of our paper (learning the full return distribution is no harder than learning its expectation with function approximation) is an important insight in RL.
> We are also grateful for your positive feedback on our writing (particularly Section 5) and the experimental section.
>
> Regarding your concerns, we provide the following responses:
>
> > 1. One suggestion for improvement would be to include a table summarizing the main theoretical results and comparing them with prior work, which could enhance clarity and accessibility (the current version is somewhat dense and requires flipping back and forth to see the comparison).
>
> Thanks for your suggestion of presenting a table summarizing the main theoretical results and comparing them with prior work, we agree with you that it will enhance clarity and accessibility.
>
> We will add the following table in the revised version.
> In the table, when the task is  policy evaluation, the sample complexity is defined in terms of the $\mu_\pi$-weighted $L^2$ norm as the measure of error; when the task is distributional policy evaluation, the sample complexity is defined in terms of the $\mu_\pi$-weighted $W_1$ metric as the measure of error.
> This allows for a clearer comparison of theoretical results with prior work.
>
> | Paper | Sample Complexity | Method |
> | --- | --- | --- |
> | [1, 2, 3] | $\widetilde{\mathcal{O}}\left(\frac{\|\|\psi^\star\|\|^2\_{\Sigma_{\phi}}+1}{(1-\gamma)^2\lambda\_{\min}}(\frac{1}{\varepsilon^2}+\frac{1}{\lambda\_{\min}})\right)$ | Linear-TD |
> | [4] | $\widetilde{\mathcal{O}}\left(\frac{\|\|\psi^\star\|\|^2\_{\Sigma\_{\phi}}+1}{\varepsilon^2(1-\gamma)^2\lambda\_{\min}}\right)$ | Linear-TD with Variance Reduction |
> | [5] (Section 9.7) | Contraction Analysis | Vanilla Linear-CTD |
> | This Work (Theorem E.2) | $\widetilde{\mathcal{O}}\left(\frac{K^4(\|\|\theta^\star\|\|^2\_{V_1}+1)}{(1-\gamma)^2\lambda\_{\min}}(\frac{1}{\varepsilon^2}+\frac{K^2}{\lambda\_{\min}})\right)$ | Vanilla Linear-CTD |
> | This Work (Theorem 4.1) | $\widetilde{\mathcal{O}}\left(\frac{\|\|\theta^\star\|\|^2\_{V_1}+1}{(1-\gamma)^2\lambda\_{\min}}(\frac{1}{\varepsilon^2}+\frac{1}{\lambda\_{\min}})\right)$ | Linear-CTD |
>
> Here, Vanilla Linear-CTD refers to the stochastic semi-gradient descent (SSGD) algorithm with the probability mass function (PMF) representational (see Eqn.(26) in Appendix D.7).
> The contraction analysis in [5] in the table means that they provided a contraction analysis for the dynamic programming version of Vanilla Linear-CTD algorithm.
>
> > 2. Additionally, the authors may consider presenting some of the empirical results in the main text, despite space limitations.
>
> Thank you for your suggestion. We agree that presenting some experimental results in the main text would be helpful for readers.
> We will move Figures 1 and 2 to the main text in the revised version.
> Figure 1 demonstrates the convergence results of our algorithm.
> By comparing Figure 1 with Figure 2, we can verify the advantage of our Linear-CTD over the baseline algorithm (Vanilla Linear-CTD) as mentioned in Remark 5: when $K$ increases, the step size of the baseline algorithm needs to decay (Lemma E.3).
> In contrast, the step size of our Linear-CTD does not require adjustment as $K$ increases (Lemma 5.3).
>
> [1] Gen Li, Weichen Wu, Yuejie Chi, Cong Ma, Alessandro Rinaldo, and Yuting Wei. High-probability sample complexities for policy evaluation with linear function approximation. IEEE Transactions on Information Theory, 2024.
>
> [2] Sergey Samsonov, Daniil Tiapkin, Alexey Naumov, and Eric Moulines. Improved high-probability bounds for the temporal difference learning algorithm via exponential stability. In The Thirty Seventh Annual Conference on Learning Theory, pages 4511–4547. PMLR, 2024.
>
> [3] Weichen Wu, Gen Li, Yuting Wei, and Alessandro Rinaldo. Statistical Inference for Temporal Difference Learning with Linear Function Approximation. arXiv preprint arXiv:2410.16106, 2024.
>
> [4] Tianjiao Li, Guanghui Lan, and Ashwin Pananjady. Accelerated and instance-optimal policy evaluation with linear function approximation. SIAM Journal on Mathematics of Data Science, 5(1):174–200, 2023.
>
> [5] Marc G. Bellemare, Will Dabney, and Mark Rowland. Distributional Reinforcement Learning. MIT Press, 2023. http://www.distributional-rl.org.

---

### Official Review · Reviewer_j4KH · 2025-06-25

**Clarity:** 3
**Significance:** 2
**Originality:** 2
**Rating:** 4
**Confidence:** 4

**Summary:**

The paper investigates distributional temporal difference learning in the presence of linear-function approximation and proposes the Linear-CTD algorithm, which enjoys the same sample complexity as classical TD, albeit estimating the entire return distribution as opposed to just its expectation. This is done by viewing the categorical Bellman equation as a linear system and by preconditioning the matrix  so that convergence depends no longer on the number of support points  K; rigorous non-asymptotic high-probability results are further obtained by studying exponential stability of random matrix products and applying Polyak-Ruppert tail averaging, with empirical researchers confirming these rates and showcasing gains in performance over previous methods.

**Questions:**

see weakness part

**Ethical Concerns:**

["NO or VERY MINOR ethics concerns only"]

**Final Justification:**

Most of my concerns are addressed.

**Limitations:**

see weakness part

**Quality:**

2

**Strengths And Weaknesses:**

Strengths：1. the first non-asymptotic high-probability bounds for distributional TD with linear function approximation, thus filling in the gaps  2. both the linearization and the analysis of exponential stability have been carried out with great care; also, the bounds are consistent with those known in the Linear-TD setting.

Weaknesses：1. The paper's contributions are heavily reliant on pre-existing results, which dampens potential originality. Most of the basic technical challenges have been considered in prior work, such as those relating to Wasserstein-metric convergence or analysis under non-i.i.d. sampling.

2. However, there is no compelling need shown in this paper on why an in-depth analysis of distributional convergence is necessary.

---

> ### Author Rebuttal · Authors · 2025-07-29
>
> We sincerely appreciate your effort and time in reviewing our paper.
> And we are glad to hear that you find our results are novel and technically sound.
> Below, we hope our responses can address your concerns.
>
> > Weakness 1. The paper's contributions are heavily reliant on pre-existing results, which dampens potential originality. Most of the basic technical challenges have been considered in prior work, such as those relating to Wasserstein-metric convergence or analysis under non-i.i.d. sampling.
>
> While we have drawn upon some pre-existing results, including the Wasserstein-metric convergence of the baseline algorithm and the theoretical framework under non-i.i.d. sampling, we maintain that our work has significant technical originality and contributions.
>
> 1) Regarding the Wasserstein-metric convergence you mentioned, we speculate that you are referring to the contraction analysis for the dynamic programming version of the baseline algorithm Vanilla Linear-CTD provided by Section 9.7 in [1].
> To the best of our knowledge, there is a substantial gap between the contraction analysis of dynamic programming algorithms and the finite-sample analysis of online algorithm (e.g. from iterative policy evaluation (dynamic programming) to TD learning, and from value iteration to Q-Learning).
>
> 2) While we employed the theoretical framework of linear stochastic approximation from [2] (encompassing both i.i.d. and non-i.i.d. scenarios) to analyze the sample complexity, significant efforts were required to derive bounds that are independent of $K$ and match those of classic Linear-TD.
> These include a novel analysis of the upper bound on the spectral norm of matrix $C\tilde G^{-1}(r)C^{-1}$ in Section 5.3, an upper bound on the spectral norm of the expectation of the Kronecker product presented in Lemma 5.4, and the technique of converting the bounding of the norm of $b_t$ into bounding a $\ell_2$ distance between two probability distributions.
> We believe these results are of independent interest and will be beneficial for subsequent research in distributional reinforcement learning.
>
> We also would like to mention our algorithmic contributions, which is fully original.
> To be specific, we proposed an improved version of the linear-categorical TD learning algorithm (Linear-CTD).
> Compared to the baseline algorithm, which we refer to as Vanilla Linear-CTD, our algorithm is a preconditioned variant that is more robust, as its step size is independent of the number of support points $K$ (as noted by Reviewer PXyE), which is supported by extensive numerical experiments.
>
> In summary, we believe that our work demonstrates significant originality and contributions compared to pre-existing results.
> We hope that the response clarifies this point.
>
> > Weakness 2. However, there is no compelling need shown in this paper on why an in-depth analysis of distributional convergence is necessary.
>
> We would believe in-depth theoretical analyses of distribution policy evaluation algorithms is timely and important.
> In the original version, we reviewed existing theoretical work on TD learning [2, 3, 4, 5], distributional TD learning in the tabular setting [6, 7, 8, 9, 10], as well as asymptotic analyses of distributional TD learning under linear function approximation [1].
> Building on these prior studies, we argued that investigating the non-asymptotic convergence rate of distributional TD learning under linear function approximation is a natural and meaningful extension.
>
> In the revised version, we will include the following content in the Introduction section to further clarify the significance of an in-depth analysis of distributional convergence.
>
> 1. In risk-sensitive domains such as finance [11] and healthcare [12], decision-making relies not only on the mean of returns but also on some risk metrics (e.g., variance, quantile, CVaR).
> Distributional convergence results offer theoretical guarantees for the reliable estimation of these risk metrics in practical tasks.
>
> 2. Many risk-sensitive RL algorithms [13, 14] incorporate distributional RL as a core step to estimate specific expected utility functions of return distributions.
> The distributional convergence results can provide finite-sample convergence rates for this critical step, giving theoretical guarantees for these algorithms.
>
> For further discussions on the significance of studying distributional RL algorithms, it could be also referred to Chapter 1 in [1] and Section 2.2 in [15].
>
> [1] Marc G. Bellemare, Will Dabney, and Mark Rowland. Distributional Reinforcement Learning. MIT Press, 2023. http://www.distributional-rl.org.
>
> [2] Sergey Samsonov, Daniil Tiapkin, Alexey Naumov, and Eric Moulines. Improved high-probability bounds for the temporal difference learning algorithm via exponential stability. In The Thirty Seventh Annual Conference on Learning Theory, pages 4511–4547. PMLR, 2024.
>
> [3] Gen Li, Changxiao Cai, Yuxin Chen, Yuting Wei, and Yuejie Chi. Is q-learning minimax optimal? a tight sample complexity analysis. Operations Research,
> 72(1):222–236, 2024a.
>
> [4] Gen Li, Weichen Wu, Yuejie Chi, Cong Ma, Alessandro Rinaldo, and Yuting Wei. High-probability sample complexities for policy evaluation with linear function approximation. IEEE Transactions on Information Theory, 2024.
>
> [5] Weichen Wu, Gen Li, Yuting Wei, and Alessandro Rinaldo. Statistical Inference for Temporal Difference Learning with Linear Function Approximation. arXiv preprint arXiv:2410.16106, 2024.
>
> [6] Mark Rowland, Marc Bellemare, Will Dabney, R´emi Munos, and Yee Whye Teh. An analysis of categorical distributional reinforcement learning. In International Conference on Artificial Intelligence and Statistics, pages 29–37. PMLR, 2018.
>
> [7] Mark Rowland, R´emi Munos, Mohammad Gheshlaghi Azar, Yunhao Tang, Georg Ostrovski, Anna Harutyunyan, Karl Tuyls, Marc G Bellemare, and Will Dabney. An analysis of quantile temporal-difference learning. Journal of Machine Learning Research, 25:1–47, 2024.
>
> [8] Liangyu Zhang, Yang Peng, Jiadong Liang, Wenhao Yang, and Zhihua Zhang.
> Estimation and Inference in Distributional Reinforcement Learning. The
> Annals of Statistics, 2025. Forthcoming.
>
> [9] Mark Rowland, Li Kevin Wenliang, Remi Munos, Clare Lyle, Yunhao Tang, and Will Dabney. Near-minimax-optimal distributional reinforcement learning with a generative model. In The Thirty-eighth Annual Conference on Neural Information Processing Systems, 2024
>
> [10] Yang Peng, Liangyu Zhang, and Zhihua Zhang. Statistical efficiency of distributional temporal difference learning. In The Thirty-eighth Annual Conference on Neural Information Processing Systems, 2024.
>
> [11] Haoran Wang and Xun Yu Zhou. Continuous-time mean–variance portfolio selection: A reinforcement learning framework. Mathematical Finance, 30(4): 1273–1308, 2020.
>
> [12] Lan Wang, Yu Zhou, Rui Song, and Ben Sherwood. Quantile-optimal treatment regimes. Journal of the American Statistical Association, 113(523):1243–1254,2018.
>
> [13] Mehrdad Moghimi and Hyejin Ku. Beyond CVar: Leveraging static spectral risk measures for enhanced decision-making in distributional reinforcement learning. In Forty-second International Conference on Machine Learning, 2025. URL https://openreview.net/forum?id=WeMpvGxXMn.
>
> [14] Bernardo Avila Pires, Mark Rowland, Diana Borsa, Zhaohan Daniel Guo, Khimya Khetarpal, Andre Barreto, David Abel, Remi Munos, and Will Dabney. Optimizing return distributions with distributional dynamic programming. arXiv preprint arXiv:2501.13028, 2025.
>
> [15] Zhengling Qi, Chenjia Bai, Zhaoran Wang, and Lan Wang. Distributional off-policy evaluation in reinforcement learning. Journal of the American Statistical Association, jun 2025. doi: 10.1080/01621459.2025.2506197. Forthcoming.

---

> > ### Comment · Reviewer_j4KH · 2025-08-06
> > **thanks for your response. I will increase the score.**
> >
> > thanks for your response. I will increase the score.

---

> > > ### Author Response · Authors · 2025-08-06
> > >
> > > Thank you for your positive feedback and for increasing the score. We appreciate your time and consideration. And we will work to refine the manuscript further and incorporate your valuable and constructive suggestions in the revision.

---

### Official Review · Reviewer_PXyE · 2025-07-01

**Clarity:** 3
**Significance:** 4
**Originality:** 3
**Rating:** 5
**Confidence:** 4

**Summary:**

This paper studies the sample complexity of temporal difference algorithms based on categorical representations and linear function approximation in the context of distributional reinforcement learning.
In particular, the authors analyze the non-asymptotic convergence rate of the proposed method, Linear-CTD, in terms of $W_{1,\mu_{\pi}}$ and derive tight sample complexity bounds.

**Questions:**

Could the author elaborate on the meaning of Equation (10) and the footnote that follows it?
Specifically, is the term $(k+1)/(K+1)$ used as a technical convenience for representing the discrete uniform distribution in terms of $\mathrm{1}_k^T$?
If so, does it also help improve the tightness of the derived sample complexity bounds?

**Ethical Concerns:**

["NO or VERY MINOR ethics concerns only"]

**Final Justification:**

After reading the rebuttal and participating in the discussion, I am satisfied with the authors’ clarification, particularly regarding the role of normalization and the use of the uniform distribution. I am maintaining my positive score and increasing my confidence from 3 to 4.

**Limitations:**

Line 214 mentions the need for future analysis of categorical algorithms with nonlinearity.

Line 266 suggests that variance-reduction techniques may help overcome the sample size barrier.

**Paper Formatting Concerns:**

No major formatting issues observed.

**Quality:**

4

**Strengths And Weaknesses:**

**Strengths**

- Despite the complexity of the literature, the paper clearly explains the necessary background on linear function approximation and distributional RL using standard notation. The overall structure is well-organized and readable.

- The proposed method, Linear-CTD, is robust in the sense that its step size $\alpha$ does not depend on the number of support points $K$. Although the paper is theory-focused, the authors also provide supporting experiments and include code in the supplementary material.

- By thoroughly comparing with existing analyses on categorical representations, the authors clearly highlight the novelty of Linear-CTD. Although I did not check all proofs in detail, the theoretical results appear consistent with prior work.

**Weaknesses**

- Although some explanation is provided in the appendix, the meaning and implication of Theorem 3.1 may not be immediately clear to readers unfamiliar with this line of work. For better readability, a short paragraph providing intuition or commentary around the theorem in the main text would be helpful.

**Minor Comments**

Since the paper focuses on categorical representations, the authors may consider citing the following two important works in the related literature section:

- Rowland, Mark, et al. "Statistics and samples in distributional reinforcement learning." International Conference on Machine Learning. PMLR, 2019.

- Cho, Taehyun, et al. "Tractable and Provably Efficient Distributional Reinforcement Learning with General Value Function Approximation." arXiv preprint arXiv:2407.21260 (2024).

---

> ### Author Rebuttal · Authors · 2025-07-29
>
> Thank you for your encouraging comments.
> We are very glad to know that you think our paper well-organized and readable, and our Linear-CTD robust and novel compared with baseline method.
> Regarding the weaknesses, minor comments and questions, we provide the following detailed responses:
>
> > Weakness: Although some explanation is provided in the appendix, the meaning and implication of Theorem 3.1 may not be immediately clear to readers unfamiliar with this line of work. For better readability, a short paragraph providing intuition or commentary around the theorem in the main text would be helpful.
>
> Thank you for your feedback. We fully agree that adding more explanations for the linear-categorical projected Bellman equation (Theorem 3.1) will enhance readability.
> In the original version, due to space constraints some critical intermediate results had to be placed in the appendix, such as Proposition D.3, which characterizes the properties of the categorical projected Bellman operator and introduces the notations $g$ and $G$ for the first time.
> In the revised version, we will move this proposition back to the main text and systematically elaborate on the important concepts including $g$ and $G$ which are related to the categorical projected Bellman operator.
> To further help readers understand, we will add the following table in the revised version that compares various concepts between Linear-TD and Linear-CTD.
> We believe this will improve the readability of the paper.
>
> | Concepts | Linear-TD | Linear-CTD |
> | --- | --- | --- |
> | Parametrization | $V_\psi(s)=\phi(s)^\top \psi \quad [\text{Eqn.~(3)}]$ | $F_k(s; \theta)=\phi(s)^\top \theta(k)+\frac{k+1}{K+1} \quad [\text{Eqn.~(10)}]$ |
> | Bellman Operator | $(T^\pi V)(s)=\mathbb{E}[r_0+\gamma V(s_1)\mid s_0=s]$ | $(\mathcal{T}^\pi \eta)(s)=\mathbb{E}[(b_{r_0,\gamma})\_{\\#}\eta(s_1)\mid s_0=s]$ |
> | Projection Operator | $V_{\tilde \psi}=\Pi^\pi\_\phi V, \tilde{\psi}=\Sigma^{-1}\_{\phi}\mathbb{E}_{s}[\phi(s)V(s)]$ | $\eta_{\tilde{\theta}}=\Pi^{\pi}\_K \eta, \tilde{\Theta}=\Sigma^{-1}\_{\phi}\mathbb{E}_{s}[\phi(s)(F\_{\eta}(s)-F\_{\nu}(s))^\top]  \quad [\text{Eqn. (11)}]$ |
> | Projected Bellman Equation | $V_{\psi}=\Pi^\pi_\phi T^\pi V_{\psi} , \quad [\text{See Eqn.~(4)}]$ | $\eta_{ \theta}=\Pi^\pi_{\phi,K} \mathcal{T}^\pi \eta_{\theta}, \quad [\text{See Eqn.~(12)}]$ |
> | Update Rule | $[\text{See Eqn.~(5)}]$ | $[\text{See Eqn.~(13)}]$ |
> | $A_t$ | $\phi(s_t)\phi(s_t)-\gamma\phi(s_t)\phi(s_{t+1})$ | $[I_K\otimes(\phi(s_t)\phi(s_t))]-[(C\tilde G^{-1}(r_t)C^{-1})\otimes(\phi(s_t)\phi(s_{t+1}))]$ |
> | Key Quantity in $A_t$ | $\gamma$ | $C\tilde G^{-1}(r_t)C^{-1} \text{ with spectral norm }\sqrt{\gamma}$ |
> | $b_t$ | $r_t\phi(s_t)$ | $\frac{1}{K+1}[C(\sum_{j=0}^Kg_j(r_t)-1_K)]\otimes \phi(s_t)$ |
> | Key Quantity in $b_t$ | $r_t\leq 1$ | $K^{-3/2}(1-\gamma)^{-1}\Vert C(\sum_{j=0}^Kg_j(r_t)-1_K)\Vert \leq 1$ |
> | Measure of Error | $\Vert V-V^\pi\Vert \_{\mu\_\pi}=(\mathbb{E}_{s}[(V(s)-V^\pi(s))^2])^{1/2}$ | $W\_{1,\mu_\pi}(\eta, \eta^\pi)=(\mathbb{E}_{s}[W_1^2(\eta(s),\eta^\pi(s))])^{1/2}$ |
> | Approximation Error | $\Vert V\_{\psi^\star}-V^\pi\Vert \_{\mu_\pi}\leq (1-\gamma^2)^{-1/2}\Vert \Pi^\pi\_\phi V^\pi-V^\pi\Vert \_{\mu_\pi}$ | $[\text{See Proposition~3.2}]$ |
> | Sample Complexity | $\widetilde{\mathcal{O}}\left(\frac{\Vert \psi^\star\Vert ^2_{\Sigma_{\phi}}+1}{(1-\gamma)^2\lambda_{\min}}\left(\frac{1}{\varepsilon^2}+\frac{1}{\lambda_{\min}}\right)\right)$ | $\widetilde{\mathcal{O}}\left(\frac{\Vert\theta^\star\Vert ^2_{V_1}+1}{(1-\gamma)^2\lambda_{\min}}\left(\frac{1}{\varepsilon^2}+\frac{1}{\lambda_{\min}}\right)\right)\quad [\text{See Theorem~4.1}]$ |
>
> > Minor Comments
>
> Thank you for your reminder.
> The important works [1, 2] are not limited to the classic categorical and quantile representations; instead, they discuss distributional RL algorithms that consider general statistical functional representations.
> We will discuss these works in the revised version.
>
> > Questions: Could the author elaborate on the meaning of Equation (10) and the footnote that follows it? Specifically, is the term $(k+1)/(K+1)$ used as a technical convenience for representing the discrete uniform distribution in terms of $1_K^\top$? If so, does it also help improve the tightness of the derived sample complexity bounds?
>
> Thank you for your valuable questions and suggestions.
> The term $(k+1)/(K+1)$ in Eqn. (10) is indeed the cdf of discrete uniform distribution $\nu$ on $\\{x\_0,\ldots,x\_K\\}$ at $x_k$.
> And as you noted, introducing normalization indeed helps improve the tightness of the derived sample complexity bound and is indispensable.
> In addition, the normalization also offers algorithmic and interpretability benefits.
> We will elaborate on normalization in detail below and add this content in the revised version.
> We believe that this will enhance readers' understanding.
>
> As stated in footnote 2, the $(k+1)/(K+1)$ term corresponds to the cdf of a discrete uniform distribution $\nu$ on $\\{x\_0,\ldots,x\_K\\}$ at $x_k$ for normalization, guaranteeing our estimator has mass $1$.
> First, we explain why normalization is necessary.
> Although we work with signed measures, it remains necessary to ensure the total mass is normalized to $1$.
> The reasons are from both algorithmic and theoretical considerations.
> From an algorithmic perspective, without normalization, the difference between our estimator and the true return distribution would not be a signed measure with mass zero.
> This would make our estimator less interpretable and lead to issues in defining many important quantities.
> For instance, the $W_1$ distance would lose its integral representation $W_1(\nu_1,\nu_2)=\sup_{f:Lip(f)\leq 1}(\nu_1-\nu_2)f$.
> Furthermore, it would fail to generate the same iterative sequence as classic TD (Proposition D.4) and would not be equivalent to tabular CTD in the tabular case.
> In this sense, the algorithm will not be a natural extension of Linear-TD and tabular CTD.
> From a theoretical perspective, tight sample complexity bounds can only be achieved after introducing normalization.
> Without normalization, the $1_K$ term would be absent from Eqn. (18), which would lead to a worse bound of $\|\|b_t\|\|$, and furthermore, the sample complexity of Linear-CTD cannot match that of classic Linear-TD.
>
> Next, we explain the impact of choosing different normalizers (distributions used for normalization) and why we choose the discrete uniform distribution $\nu$ as the normalizer.
> As noted in footnote 2, the definition of space $\mathscr{P}_{\phi,K}^{sign}$ depends on the choice of the normalizer, further affecting the optimal solution and the iterating process of the algorithm, an outcome we aim to avoid.
> Fortunately, such undesirable properties would not appear in the tabular case.
> In the tabular case, it can be easily shown that the Linear-CTD algorithm is equivalent to tabular CTD regardless of the distribution chosen for normalization.
> In the general case, we can introduce an intercept term (a state-independent feature that is always $1$) in the feature $\phi$ to ensure that the choice of the distribution has no impact on the algorithm (because we can always adjust the coefficients of the intercept terms to offset the impact of changing the normalizers).
> Based on the above discussion, any distribution could be used for normalization in practice.
>
> And we choose the uniform distribution as the normalizer only for conceptual and technical simplicity.
> For example, in Section 5.3 Bounding the Norm of $b_t$ paragraph, using the discrete uniform distribution $\nu$ allows us to more conveniently derive results like Lemma G.4, thereby making it easy to get a tight bound for $\|\|b_t\|\|$.
> We hope this explanation is helpful.
>
> [1] Rowland, Mark, et al. Statistics and samples in distributional reinforcement learning. International Conference on Machine Learning. PMLR, 2019.
>
> [2] Cho, Taehyun, et al. Tractable and Provably Efficient Distributional Reinforcement Learning with General Value Function Approximation. arXiv preprint arXiv:2407.21260 (2024).

---

> > ### Comment · Reviewer_PXyE · 2025-08-04
> >
> > Thanks for the thorough and thoughtful response. The clarification on normalization, especially the explanation around the uniform distribution $\nu$ and its role in Footnote 2 , was very helpful. I also found the summary table particularly useful in understanding the analysis, and I would strongly recommend including it in the appendix of the revised version.
> >
> > The author's response addressed the majority of my concerns. I will maintain a positive score and increase my confidence from 3 to 4 accordingly.

---

> > > ### Author Response · Authors · 2025-08-04
> > >
> > > Thank you very much for your thorough review and valuable feedback.
> > > We are glad to hear that you found our clarification on normalization helpful and our summary table particularly useful in understanding the analysis. We will incorporate your suggestions in the revised version.
> > >
> > > Once again, thank you for your positive evaluation and the increase in confidence from 3 to 4.

---

### Official Review · Reviewer_8d43 · 2025-07-03

**Clarity:** 4
**Significance:** 3
**Originality:** 3
**Rating:** 5
**Confidence:** 4

**Summary:**

This work derives finite-sample rates for distributional TD learning algorithm with linear function approximation. Specifically, the work formulates a projected Bellman operator related to the distributional RL setup and builds a stochastic approximation algorithm to find its fixed point. Their main result concerns the sample complexity of  this algorithm in the generative and the Markovian setting. It is shown that the algorithm needs $O(\epsilon^{-2})$ samples to get an $\epsilon$-accurate solution estimate. This sample complexity matches that of classic TD learning.

**Questions:**

1. Why have you added $(k + 1)/(K + 1)$ in (10)? Please add detail on your claim that it is for normalization.

2. What are the limitations of working with signed measures? Suppose we use a notion of ordering between the return distributions of different policies, e.g., first order stochastic dominance. Would this ordering continue to hold when we project them onto the space of signed measures?

3. More generally, policy iteration with function approximation is known to suffer from policy oscillation, divergence and other issues (Bertsekas, 2011). Do you think Policy Iteration with distributional policy evaluation or some variant address those issues?

Bertsekas, D.P., 2011. Approximate policy iteration: A survey and some new methods. Journal of Control Theory and Applications, 9(3), pp.310-335.

**Ethical Concerns:**

["NO or VERY MINOR ethics concerns only"]

**Final Justification:**

The paper has analyzed distributional TD learning with linear function approximation. I believe the work should be of interest to researchers working on disrtributional RL.

Hence, I recommend a clear accept.

**Limitations:**

Yes

**Quality:**

3

**Strengths And Weaknesses:**

**Strengths**:

1. Paper is well written and easy to read.
2. The paper is able to analyze the challenging distributional TD learning algorithm with linear functiona approximation. Distributional RL has been a promising direction in recent literature and this work should enable their

**Weaknesses**:

1. The proposed theory and algorithms apply to the space of signed measures, rather than usual unsigned measures.

---

> ### Author Rebuttal · Authors · 2025-07-29
>
> Thank you for your valuable review and suggestions.
> We are very glad to know that
> you think our paper is well-written, and our results are helpful for subsequent research in distributional RL.
>
> Regarding the weaknesses and questions, we provide the following detailed responses:
>
> > Weakness 1: The proposed theory and algorithms apply to the space of signed measures, rather than usual unsigned measures. And Question 2: What are the limitations of working with signed measures?
>
> We are grateful to you for raising this issue.
> We would like to point out similar issues also appear in works such as [1, 2] and Section 9.6 in [3], and our new numerical experiments indicate that in non-tabular settings, non-probability signed measures generally arise.
> The non-probability signed measure does induce some limitations.
> For instance, this makes some common statistical functionals such as quantiles and CVaR ill-defined, thereby failing to guarantee their convergence.
> We will provide a more detailed discussion in the revised version.
>
> However, according to our Theorem 4.1, for statistical functionals that are Lipschitz continuous with respect to the $W_1$ distance (e.g., expected utility when the utility function is Lipschitz), we can still establish their finite-sample convergence.
> Thus, our finite-sample theory built on signed measure spaces still finds applications in many problems.
> For example, some risk-sensitive RL algorithms, such as [4, 5], use distributional RL algorithms to estimate specific expected utility functions of the distributions, and our results can give a finite sample convergence rate.
>
> To avoid issues caused by non-probability signed measures, one could consider linear function approximation with softmax-parameterized categorical representations, or quantile  representations in Section 9.5 in [3].
> However, these parameterizations leads to nonlinear algorithms, which may prevent us from proving finite-sample convergence comparable to that of Linear-TD.
> Compared to these two nonlinear algorithms, the introduction of signed measures in our Linear-CTD and other works helps preserve the linear structure of the algorithm, enabling us to obtain convergence results that match those of Linear-TD.
> Regarding the finite-sample analysis of the aforementioned nonlinear algorithms, we leave this as future work.
> We hope this explanation is helpful.
>
> > Question 1: Why have you added $(k+1)/(K+1)$ in (10)? Please add detail on your claim that it is for normalization.
>
> Thank you for your valuable questions and suggestions.
> This term serves as a normalizer.
> Below we will provide a detailed explanation and include it in the revised version.
>
> As stated in footnote 2, the $(k+1)/(K+1)$ term corresponds to the cdf of a discrete uniform distribution $\nu$ on $\\{x\_0,\ldots,x\_K\\}$ at $x_k$ for normalization, guaranteeing our estimator has mass $1$.
> First, we explain why normalization is necessary.
> Although we work with signed measures, it remains necessary to ensure the total mass is normalized to $1$.
> The reasons are from both algorithmic and theoretical considerations.
> From an algorithmic perspective, without normalization, the difference between our estimator and the true return distribution would not be a signed measure with mass zero.
> This would make our estimator less interpretable and lead to issues in defining many important quantities.
> For instance, the $W_1$ distance would lose its integral representation $W_1(\nu_1,\nu_2)=\sup_{f:Lip(f)\leq 1}(\nu_1-\nu_2)f$.
> Furthermore, it would fail to generate the same iterative sequence as classic TD (Proposition D.4) and would not be equivalent to tabular CTD in the tabular case.
> In this sense, the algorithm will not be a natural extension of Linear-TD and tabular CTD.
> From a theoretical perspective, tight sample complexity bounds can only be achieved after introducing normalization.
> Without normalization, the $1_K$ term would be absent from Eqn. (18), which would lead to a worse bound of $\|\|b_t\|\|$, and furthermore, the sample complexity of Linear-CTD cannot match that of classic Linear-TD.
>
> Next, we explain the impact of choosing different normalizers (distributions used for normalization) and why we choose the discrete uniform distribution $\nu$ as the normalizer.
> As noted in footnote 2, the definition of space $\mathscr{P}_{\phi,K}^{sign}$ depends on the choice of the normalizer, further affecting the optimal solution and the iterating process of the algorithm, an outcome we aim to avoid.
> Fortunately, such undesirable properties would not appear in the tabular case.
> In the tabular case, it can be easily shown that the Linear-CTD algorithm is equivalent to tabular CTD regardless of the distribution chosen for normalization.
> In the general case, we can introduce an intercept term (a state-independent feature that is always $1$) in the feature $\phi$ to ensure that the choice of the distribution has no impact on the algorithm (because we can always adjust the coefficients of the intercept terms to offset the impact of changing the normalizers).
> Based on the above discussion, any distribution could be used for normalization in practice.
>
> And we choose the uniform distribution as the normalizer only for conceptual and technical simplicity.
> For example, in Section 5.3 Bounding the Norm of $b_t$ paragraph, using the discrete uniform distribution $\nu$ allows us to more conveniently derive results like Lemma G.4, thereby making it easy to get a tight bound for $\|\|b_t\|\|$.
> We hope this explanation is helpful.
>
> > Question 2 Cont.: Suppose we use a notion of ordering between the return distributions of different policies, e.g., first order stochastic dominance. Would this ordering continue to hold when we project them onto the space of signed measures?
>
> Thank you for raising this valuable question.
> Through theoretical analysis, we find that the answer is no.
> However, the reason why the ordering no longer holds generally is not due to the issue of non-probability signed measures, but the linear function approximation.
> Here we give the justification:
> Let $\xi_1,\xi_2\in\mathscr{P}^{\mathcal{S}}$ such that $\xi_1$ stochastic dominant $\xi_2$, that is $F_{\xi_1(s)}(x)\geq F_{\xi_2(s)}(x)$ for any $s\in\mathcal{S}$ and $x\in\mathbb{R}$.
> We denote by $\mathbf{F}\_{\xi_i}(s)=(F\_{\xi_i(s)}(x_k))\_{k\in\{0,1,\ldots,K-1\}}$ the cdf vector of the categorical projection $\Pi\_K \xi\_i(s)\in\mathscr{P}\_K$.
> Then using $\Pi\_K$ is a monotone map (Proposition 5 in [6]), we have $\mathbf{F}\_{\xi\_1}(s)\geq\mathbf{F}\_{\xi\_2}(s)$ element-wisely.
> By Proposition 3.1, the difference of the cdf vectors of $(\Pi^\pi\_{\phi, K}\xi\_1)(s^\prime),(\Pi^\pi\_{\phi, K}\xi\_2)(s^\prime)\in\mathscr{P}\_K$ is given by $\tilde{\mathbf F}\_{\xi\_1}(s^\prime)-\tilde{\mathbf F}\_{\xi\_2}(s^\prime)=\mathbb{E}\_{s\sim\mu\_{\pi}}[(\mathbf{F}\_{\xi\_1}(s)-\mathbf{F}\_{\xi_2}(s))\phi(s)^{\top}]\Sigma\_{\phi}^{-1}\phi(s^\prime)$.
> In the tabular case, it is easy to see that $\tilde{\mathbf F}\_{\xi_1}(s^\prime)\geq\tilde{\mathbf F}\_{\xi_2}(s^\prime)$, but in the linear approximation case, it is not necessarily non-negative because $\phi(s)^{\top}\Sigma\_{\phi}^{-1}\phi(s^\prime)$ can be negative and one can construct a counterexample based on the fact.
> Moreover, if we consider the more general case, namely we apply two different projection operators of different policies to $\xi\_1$ and $\xi\_2$ respectively, the answer is still no.
>
> In the simple classic Linear-TD, the same argument holds:
> Let $V_1, V_2\in\mathbb{R}^{\mathcal{S}}$ such that $V_1\geq V_2$, then we have $(\Pi^\pi\_{\phi}V_1)(s^\prime)-(\Pi^\pi\_{\phi}V_2)(s^\prime)=\mathbb{E}\_{s\sim\mu_{\pi}}[(V_1(s)-V_2(s))\phi(s)^{\top}]\Sigma\_{\phi}^{-1}\phi(s^\prime)$, which is not necessarily non-negative, except in the tabular case.
>
> > Question 3: More generally, policy iteration with function approximation is known to suffer from policy oscillation, divergence and other issues (Bertsekas, 2011). Do you think Policy Iteration with distributional policy evaluation or some variant address those issues?
>
> Thank you for this highly valuable question. In the case of linear approximation, we believe the answer is no. As stated in Proposition D.4 of our paper, under linear function approximation, distributional RL algorithms generate an identical sequence of value function estimates to classic RL algorithms.
> Thus, introducing distribution estimation cannot alleviate these problems.
>
> In the general nonlinear function approximation setting, as demonstrated in many empirically focused studies, distributional RL may empirically mitigate such problems, which is likely due to enhanced representation learning enabled by richer learning objectives.
>
> We hope this response is helpful.
>
> [1] Marc G Bellemare, Nicolas Le Roux, Pablo Samuel Castro, and Subhodeep Moitra. Distributional reinforcement learning with linear function approximation. AISTATS 2019.
>
> [2] Clare Lyle, Marc G Bellemare, and Pablo Samuel Castro. A comparative analysis of expected and distributional reinforcement learning. AAAI 2019.
>
> [3] Marc G. Bellemare, Will Dabney, and Mark Rowland. Distributional Reinforcement Learning. MIT Press, 2023. http://www.distributional-rl.org.
>
> [4] Mehrdad Moghimi and Hyejin Ku. Beyond CVar: Leveraging static spectral risk measures for enhanced decision-making in distributional reinforcement learning. ICML 2025.
>
> [5] Bernardo Avila Pires, Mark Rowland, Diana Borsa, Zhaohan Daniel Guo, Khimya Khetarpal, Andre Barreto, David Abel, Remi Munos, and Will Dabney. Optimizing return distributions with distributional dynamic programming. arXiv preprint arXiv:2501.13028, 2025.
>
> [6] Mark Rowland, Marc Bellemare, Will Dabney, R´emi Munos, and Yee Whye Teh. An analysis of categorical distributional reinforcement learning. AISTATS 2018.

---

> > ### Comment · Reviewer_8d43 · 2025-08-05
> >
> > Thank you for the detailed response. The theoretical results presented in this work are interesting, and I find them valuable for researchers working on distributional RL.
> >
> > That said, your response to the last question raises concerns about the practical utility of distributional RL beyond standard RL. I worry that many of the same issues that affect standard RL—such as high sensitivity to hyperparameter choices—may persist in distributional RL as well.

---

> > > ### Author Response · Authors · 2025-08-06
> > >
> > > Thank you for your positive feedback, especially for finding our results interesting and valuable for research on distributional RL (DistRL).
> > >
> > > The question you raised is valuable, and there is substantial literature focusing on it.
> > > We are more than willing to discuss it with you.
> > > Before that, we need to clarify that this question is beyond the scope of this paper.
> > > Our primary goal is to derive sample complexity bound of estimating return distributions in $W\_1$ distance that matches that of value function under linear function approximation.
> > > By the integral representation $W_1(\mu,\nu)=\sup\_{f:Lip(f)\leq 1} (\mu-\nu)f$, estimating return distributions in $W_1$ is more challenging than estimating value functions.
> > > Our results bridge the gap between the sample complexity bounds of the two tasks.
> > > And our distributional convergence results have rich applications in some practical problems mentioned below.
> > >
> > > We now discuss on the question you raised:
> > >
> > > Although DistRL faces various issues you mentioned as in standard RL.
> > > DistRL remains a valuable extension with many practical utilities beyond standard RL.
> > >
> > > 1. Estimating the return distributions (rather than only the expectation, i.e. value functions) itself has many applications, as the full distribution contains richer information than the expectation.
> > > On the one hand, in some risk-sensitive domains such as finance [1] and healthcare [2], decision-making relies not only on the mean of returns but also on certain risk metrics (e.g., variance, quantile, CVaR). Our distributional convergence results offer theoretical guarantees for the reliable estimation of these risk metrics in these practical tasks.
> > >
> > > On the other hand, many risk-sensitive RL algorithms [3, 4], which focus on optimizing some risk measures of returns beyond the expectation, incorporate DistRL as a core step to estimate specific expected utility functions of return distributions.
> > > And distributional convergence results like ours can provide finite-sample convergence rates for this critical step, thereby offering theoretical guarantees for these algorithms.
> > >
> > > For further discussions on the applications of DistRL, it could be referred to Chapter 1 and  7.6–7.8 in [5] and Section 2.2 in [6].
> > >
> > > 2. Despite sharing some common issues with standard RL, for problems focusing solely on value functions, DistRL still offer advantages:
> > > [7] show that quantile TD (QTD) are more robust than standard TD in estimating the value function.
> > > Specifically, the update variance of QTD is smaller than that of standard TD, as a result, the convergence of QTD only requires the existence of the first moment of reward distribution, whereas standard TD requires the existence of the second moment.
> > >
> > > [8, 9] propose a maximum likelihood estimation-based DistRL algorithm and showed that it achieves tighter bounds than standard RL algorithms in value function estimation.
> > >
> > > Additionally, some well-known deep DistRL algorithms such as C51 [10], IQN [11], and Rainbow [12] have empirically improved performance in standard RL tasks.
> > >
> > > Sections 10.4–10.7 in [5] discuss in detail the reasons behind the strong empirical performance of deep DistRL algorithms.
> > > A possible reason, as discussed in Page 310 in [5] and [13], is that deep DistRL algorithms enhances the representational learning of neural networks possibly through richer learning objectives, which we mentioned in our rebuttal.
> > >
> > > We hope the discussion is helpful. Thanks again for your encouragement and positive feedback.
> > >
> > > [1] Haoran Wang et al. Continuous-time mean–variance portfolio selection: A reinforcement learning framework. Mathematical Finance 2020
> > >
> > > [2] Lan Wang et al. Quantile-optimal treatment regimes. JASA 2018
> > >
> > > [3] Mehrdad Moghimi et al. Leveraging static spectral risk measures for enhanced decision-making in distributional reinforcement learning. ICML 2025
> > >
> > > [4] Bernardo Avila Pires et al. Optimizing return distributions with distributional dynamic programming. arXiv preprint.
> > >
> > > [5] Marc G. Bellemare et al. Distributional Reinforcement Learning. MIT Press, 2023.
> > >
> > > [6] Zhengling Qi et al.. Distributional off-policy evaluation in reinforcement learning. JASA 2025.
> > >
> > > [7] Mark Rowland et al.. The statistical benefits of quantile temporal-difference learning for value estimation. ICML 2023
> > >
> > > [8] Kaiwen Wang et al.. The benefits of being distributional: Small-loss bounds for reinforcement learning. NeurIPS 2023
> > >
> > > [9] Kaiwen Wang et al. More benefits of being distributional: second-order bounds for reinforcement learning. ICML 2024
> > >
> > > [10] Marc G Bellemare et al. A distributional perspective on reinforcement learning. ICML 2017
> > >
> > > [11] Will Dabney et al. Implicit quantile networks for distributional reinforcement learning. ICML 2018
> > >
> > > [12] Matteo Hessel et al. Rainbow: Combining improvements in deep reinforcement learning. AAAI 2018
> > >
> > > [13] Felipe Petroski Such et al. An atari model zoo for analyzing, visualizing, and comparing deep reinforcement learning agents. IJCAI 2019

---

### Decision · Program_Chairs · 2025-09-17

**Decision:**

Accept (poster)

**Comment:**

This paper analyzed the finite sample convergence of distributional TD learning with linear function approximation. A particular useful take-away is that distributional TD is not statistically harder (ie., taking more samples) than standard TD, thus allowing bootstrapping more information for applications where higher order statistics are useful for decision making.

The reviewers appreciate the clear technical contribution of the paper and the reviewer-author discussion has resolved most of the concerns. Therefore, the paper is worthy of acceptance.